# DEEP STOCHASTIC MECHANICS

## ABSTRACT

This paper introduces a novel deep-learning-based approach for numerical simulation of a time-evolving Schrödinger equation inspired by stochastic mechanics and generative diffusion models. Unlike existing approaches, which exhibit computational complexity that scales exponentially in the problem dimension, our method allows us to adapt to the latent low-dimensional structure of the wave function by sampling from the Markovian diffusion. Depending on the latent dimension, our method may have far lower computational complexity in higher dimensions. Moreover, we propose novel equations for stochastic quantum mechanics, resulting in linear computational complexity with respect to the number of dimensions. Numerical simulations verify our theoretical findings and show a significant advantage of our method compared to other deep-learning-based approaches used for quantum mechanics.

## 1 INTRODUCTION

Mathematical models for many problems in nature appear in the form of partial differential equations (PDEs) in high dimensions. Given access to precise solutions of the many-electron time-dependent Schrödinger equation (TDSE), a vast body of scientific problems could be addressed, including in quantum chemistry (Cances et al., 2003; Nakatsuji, 2012), drug discovery (Ganesan et al., 2017; Heifetz, 2020), condensed matter physics (Liu et al., 2013; Boghosian & Taylor IV, 1998), and quantum computing (Grover, 2001; Papageorgiou & Traub, 2013). However, solving high-dimensional PDEs and the Schrödinger equation, in particular, are notoriously difficult problems in scientific computing due to the well-known curse of dimensionality: the computational complexity grows exponentially as a function of the dimensionality of the problem (Bellman, 2010). Traditional numerical solvers have been limited to dealing with problems in rather low dimensions since they rely on a grid.

Deep learning is a promising way to avoid the curse of dimensionality (Poggio et al., 2017; Madala et al., 2023). However, no known deep learning approach avoids it in the context of the TDSE (Manzhos, 2020). Although generic deep learning approaches have been applied to solving the TDSE (Raissi et al., 2019; E & Yu, 2017; Weinan et al., 2021; Han et al., 2018), this paper shows that it is possible to get performance improvements by developing an approach specific to the TDSE by incorporating quantum physical structure into the deep learning algorithm itself.

We propose a method that relies on a stochastic interpretation of quantum mechanics (Nelson, 1966; Guerra, 1995; Nelson, 2005) and is inspired by the success of deep diffusion models that can model complex multi-dimensional distributions effectively (Yang et al., 2022); we call it *Deep Stochastic Mechanics (DSM)*. Our approach is not limited to only the linear Schrödinger equation, but can be adapted to Klein-Gordon, Dirac equations (Serva, 1988; Lindgren & Liukkonen, 2019), and to the non-linear Schrödinger equations of condensed matter physics, e.g., by using mean-field stochastic differential equations (SDEs) (Eriksen, 2020), or McKean-Vlasov SDEs (dos Reis et al., 2022).

### 1.1 PROBLEM FORMULATION

The Schrödinger equation, a governing equation in quantum mechanics, predicts the future behavior of a dynamic system for $0 \leq t \leq T$ and $\forall x \in \mathcal{M}$:

$$i\hbar\partial_t \psi(x,t) = \mathcal{H}\psi(x,t), \tag{1}$$

$$\psi(x,0) = \psi_0(x), \tag{2}$$

where $\psi : \mathcal{M} \times [0, T] \to \mathbb{C}$ is a wave function defined over a manifold $\mathcal{M}$, and $\mathcal{H}$ is a self-adjoint operator acting on a Hilbert space of wave functions. For simplicity of future derivations, we consider a case of a spinless particle in $\mathcal{M} = \mathbb{R}^d$[1] moving in a smooth potential $V : \mathbb{R}^d \times [0, T] \to \mathbb{R}_+$. In this case, $\mathcal{H} = -\frac{\hbar^2}{2} \mathrm{Tr}(m^{-1} \nabla^2) + V$, where $m \in \mathbb{R}^d \otimes \mathbb{R}^d$ is a mass tensor. The probability density of finding a particle at position $x$ is $|\psi(x, t)|^2$. A notation list is given in Appendix A.

Given initial conditions in the form of samples drawn from a density $\psi_0(x)$, we wish to draw samples from $|\psi(x, t)|^2$ for $t \in (0, T]$ using a neural-network-based approach that can adapt to latent low-dimensional structures in the system and sidestep the curse of dimensionality. Rather than explicitly estimating $\psi(x, t)$ and sampling from the corresponding density, we devise a strategy that directly samples from an approximation of $|\psi(x, t)|^2$, concentrating computation in high-density regions. When regions where the density $|\psi(x, t)|^2$ lie in a latent low-dimensional space, our sampling strategy concentrates computation in that space, leading to the favorable scaling properties of our approach.

## 2 RELATED WORK

Physics-Informed Neural Networks (PINNs) (Raissi et al., 2019) are general-purpose tools that have been widely studied for their ability to solve PDEs and can be applied to solve Equation (1). However, this method is prone to the same issues as classical numerical algorithms since it relies on a collection of collocation points uniformly sampled over the domain $\mathcal{M} \subseteq \mathbb{R}^d$. In the remainder of the paper, we refer to this as a 'grid' for simplicity of exposition. Another recent paper by Bruna et al. (2022) introduces Neural Galerkin schemes based on deep learning, which leverage active learning to generate training data samples for numerically solving real-valued PDEs. Unlike collocation-points-based methods, this approach allows theoretically adaptive data collection guided by the dynamics of the equations if we could sample from the wave function effectively.

Another family of approaches, FermiNet (Pfau et al., 2020) or PauliNet (Hermann et al., 2020), reformulates the problem (1) as maximization of an energy functional that depends on the solution of the stationary Schrödinger equation. This approach sidesteps the curse of dimensionality but cannot be adapted to the time-dependent wave function setting considered in this paper.

The only thing that we can experimentally obtain is samples from the quantum mechanics density. So, it makes sense to focus on obtaining samples from the density rather than attempting to solve the Schrödinger equation; these samples can be used to predict the system's behavior without conducting real-world experiments. Based on this observation, there are a variety of quantum Monte Carlo methods (Corney & Drummond, 2004; Barker, 1979; Austin et al., 2012), which rely on estimating expectations of observables rather than the wave function itself, resulting in improved computational efficiency. However, these methods still encounter the curse of dimensionality due to recovering the full-density operator. The density operator in atomic simulations is concentrated on a lower dimensional manifold of such operators (Eriksen, 2020), suggesting that methods that adapt to this manifold can be more effective than high-dimensional grid-based methods. Deep learning has the ability to adapt to this structure.

As noted in Schlick (2010), knowledge of the density is unnecessary for sampling. We need a score function $\nabla \log \rho$ to be able to sample from it. The fast-growing field of generative modeling with diffusion processes demonstrates that for high-dimensional densities with low-dimensional manifold structure, it is incomparably more effective to learn a score function than the density itself (Ho et al., 2020; Yang et al., 2022).

For high-dimensional real-valued PDEs, there exist a variety of classic and deep learning-based approaches that rely on sampling from diffusion processes, e.g., (Cliffe et al., 2011; Warin, 2018; Han et al., 2018; Weinan et al., 2021). Those works rely on the Feynman-Kac formula (Del Moral, 2004) to obtain an estimator for the solution to the PDE. However, for the Schrödinger equation, we need an analytical continuation of the Feynman-Kac formula on an imaginary time axis (Yan, 1994) as it is a complex-valued equation. This requirement limits the applicability of this approach to our setting. BSDE methods studied by Nüsken & Richter (2021b;a) are closely related to our approach but they are developed for the elliptic version of the Hamilton–Jacobi–Bellman (HJB) equation. We consider the hyperbolic HJB setting, for which the existing method cannot be applied.

---

[1]A multi-particle case is covered by considering $d = 3n$, where $n$ – the number of particles.

## 3 CONTRIBUTIONS

We are inspired by works of Nelson (1966; 2005), who has developed a stochastic interpretation of quantum mechanics, so-called stochastic mechanics, based on a Markovian diffusion. Instead of solving the Schrödinger Equation (1), our method aims to learn the stochastic mechanical process's osmotic and current velocities equivalent to classical quantum mechanics. Our formulation differs from the original one (Nelson, 1966; 2005; Guerra, 1995), as we derive equivalent differential equations describing the velocities that do not require the computation of the Laplacian operator. Another difference is that our formulation interpolates anywhere between stochastic mechanics and deterministic Pilot-wave theory (Bohm, 1952). More details are given in Appendix E.4.

We highlight the main contributions of this work as follows:

- We propose to use a stochastic formulation of quantum mechanics (Nelson, 2005; 1966; Guerra, 1995) to create an efficient computational tool for quantum mechanics simulation.

- We also derive equations describing stochastic mechanics that are equivalent to the expressions introduced by Nelson but which are expressed in terms of the gradient of the divergence operator, making them more amenable to neural network-based solvers.

- We empirically estimate the performance of our method in various settings. Our approach shows a superior advantage to PINNs in terms of accuracy. We also conduct an experiment where our method shows linear convergence time in the dimension, operating easily in a higher-dimensional setting.

- We prove theoretically in Section 4.3 that our proposed loss function upper bounds the $L_2$ distance between the approximate process and the 'true' process that samples from the quantum density, which implies that if loss converges to zero then the approximate process strongly converges to the 'true' process.

Table 1 compares properties of methods for solving Equation (1), where $N$ is the number of discretization points in time, $H_d$ is the number of Monte Carlo iterations required by FermiNet to draw a single sample, and $N_f$ is a number of collocation points for PINN. We follow the general recommendation that each spatial dimension's number of points on the grid should be $\sqrt{N}$. Thus, the number of points on a grid is $\mathcal{O}(N^{\frac{d}{2}+1})$. We assume a numerical solver aims for a precision $\varepsilon = \mathcal{O}(\frac{1}{\sqrt{N}})$. Approaches like the PINN or FermiNet require computing the Laplacian, which leads to at least quadratic computational complexity per training iteration. We also note that for our approach $N$ is independent from $d$. In the general case, overall complexity bounds are not known, except for the numerical solver. We can at least lower bound them based on iterations complexity and known bounds for the convergence of non-convex stochastic gradient descent (Fehrman et al., 2019) that scales polynomial with $\varepsilon^{-1}$.

Table 1: Comparison of different approaches for simulating quantum mechanics.

| Method | Domain | Time-evolving | Adaptive | Iteration Complexity | Overall Complexity |
|---|---|---|---|---|---|
| PINN (Raissi et al., 2019) | Compact | ✓ | ✗ | $\mathcal{O}(N_f d^2)$ | at least $\mathcal{O}(N_f d^2 \text{poly}(\varepsilon^{-1}))$ |
| FermiNet (Pfau et al., 2020) | $\mathbb{R}^d$ | ✗ | ✓ | $\mathcal{O}(H_d d^2)$[2] | at least $\mathcal{O}(H_d d^2 \text{poly}(\varepsilon^{-1}))$ |
| Numerical solver | Compact | ✓ | ✗ | N/A | $\mathcal{O}(d\varepsilon^{-d-2})$ |
| **DSM (Ours)** | $\mathbb{R}^d$ | ✓ | ✓ | $\mathcal{O}(Nd)$ | at least $\mathcal{O}(Nd\text{poly}(\varepsilon^{-1}))$ |

## 4 DEEP STOCHASTIC MECHANICS

There is a family of diffusion processes that are equivalent to Equation (1) in a sense that all time-marginals of any such process coincide with $|\psi|^2$; we refer to Appendix E for derivation. Assuming $\psi(x,t) = \sqrt{\rho(x,t)}e^{iS(x,t)}$, we define:

$$v(x,t) = \frac{\hbar}{m}\nabla S(x,t) \quad \text{and} \quad u(x,t) = \frac{\hbar}{2m}\nabla \log \rho(x,t). \tag{3}$$

---

[2] Although the original method is for Bohr-Openheimmer potential that leads to even higher complexity, we do not take this into account here.

Our method relies on the following stochastic process with $\nu \geq 0$ [3], which corresponds to sampling from $\rho = \left|\psi(x,t)\right|^2$ (Nelson, 1966):

$$\mathrm{d}Y(t) = (v(Y(t),t) + \nu u(Y(t),t))\mathrm{d}t + \sqrt{\frac{\nu\hbar}{m}}\mathrm{d}\vec{W}, \tag{4}$$

$$Y(0) \sim \left|\psi_0\right|^2, \tag{5}$$

where $u$ is an osmotic velocity, $v$ is a current velocity and $\vec{W}$ is a standard (forward) Wiener process. Process $Y(t)$ is called the Nelsonian process. Since we don't know the true $u, v$, we instead aim at approximating them with the process defined using neural network approximations $v_\theta, u_\theta$:

$$\mathrm{d}X(t) = (v_\theta(X(t),t) + \nu u_\theta(X(t),t))\mathrm{d}t + \sqrt{\frac{\nu\hbar}{m}}\mathrm{d}\vec{W}, \tag{6}$$

Any numerical integrator can be used to obtain samples from the diffusion process. The simplest one is the Euler-Maryama integrator (Kloeden & Platen, 1992):

$$X_{i+1} = X_i + (v_\theta(X_i,t_i) + \nu u_\theta(X_i,t_i))\epsilon + \mathcal{N}\left(0,\frac{\nu\hbar}{m}\epsilon I_d\right), \tag{7}$$

where $\epsilon > 0$ denotes a step size, $0 \leq i < \frac{T}{\epsilon}$, and $\mathcal{N}(0,I_d)$ is a Gaussian distribution. We consider this integrator in our work. Switching to higher-order integrators, e.g., the Runge-Kutta family of integrators (Kloeden & Platen, 1992), can potentially enhance efficiency and stability when $\epsilon$ is larger.

The diffusion process (4) achieves sampling from $\rho = \left|\psi(x,t)\right|^2$ for each $t \in [0,T]$ for known $u$ and $v$. Assume that $\psi_0(x) = \sqrt{\rho_0(x)}e^{iS_0(x)}$. Our approach relies on the following equations for the velocities:

$$\partial_t v = -\frac{1}{m}\nabla V + \langle u, \nabla \rangle u - \langle v, \nabla \rangle v + \frac{\hbar}{2m}\nabla\langle\nabla, u\rangle, \tag{8a}$$

$$\partial_t u = -\nabla\langle v, u \rangle - \frac{\hbar}{2m}\nabla\langle\nabla, v\rangle, \tag{8b}$$

$$v_0(x) = \frac{\hbar}{m}\nabla S_0(x), \ u_0(x) = \frac{\hbar}{2m}\nabla\log\rho_0(x). \tag{8c}$$

These equations are derived in Appendix E.1 and are equivalent to the Schrödinger equation. As mentioned, our equations differ from the canonical ones developed in Nelson (1966); Guerra (1995). In particular, the original formulation in (27), which we call the *Nelsonian version*, includes the Laplacian of $u$; in contrast, *our version* in (8a) uses the gradient of the divergence operator. These versions are equivalent in our setting, but our version has significant computational advantages, as we describe later in Proposition 4.1.

## 4.1 LEARNING DRIFTS

This section describes how we learn the velocities $u_\theta(X,t)$ and $v_\theta(X,t)$, parameterized by neural networks with parameters $\theta$. We propose to use a combination of three losses: two of them come from the Navier-Stokes-like equations (8a), (8b), and the third one enforces the initial conditions (8c). We define non-linear differential operators that appear in Equation (8a), (8b):

$$\mathcal{D}_u[v,u,x,t] = -\nabla\langle v(x,t), u(x,t)\rangle - \frac{\hbar}{2m}\nabla\langle\nabla, v(x,t)\rangle, \tag{9}$$

$$\mathcal{D}_v[v,u,x,t] = \frac{1}{m}\nabla V(x,t) + \frac{1}{2}\nabla\|u(x,t)\|^2 - \frac{1}{2}\nabla\|v(x,t)\|^2 + \frac{\hbar}{2m}\nabla\langle\nabla, u(x,t)\rangle \tag{10}$$

We aim to minimize the following losses:

$$L_1(v_\theta, u_\theta) = \int_0^T \mathbb{E}^X\left\|\partial_t u_\theta(X(t),t) - \mathcal{D}_u[v_\theta, u_\theta, X(t), t]\right\|^2\mathrm{d}t, \tag{11}$$

---

[3]$\nu = 0$ is allowed if and only if $\psi_0$ is sufficiently regular, e.g., $|\psi_0|^2 > 0$ everywhere.

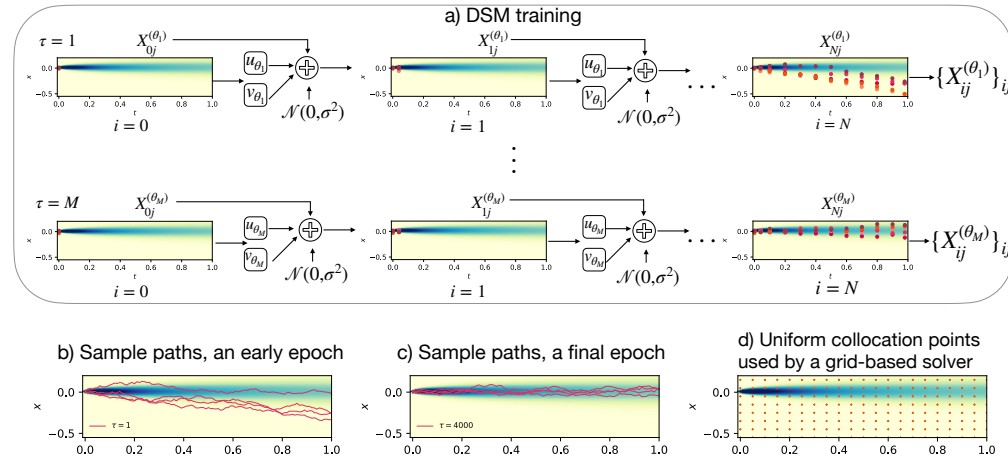

Figure 1: An illustration of our approach. Blue regions in the plots correspond to higher-density regions. (a) DSM training scheme: at every epoch $\tau$, we generate $B$ full trajectories $\{X_{ij}\}_{ij}$, $i = 0, ..., N$, $j = 1, ..., B$. Then we update the weights of our NNs. (b) An illustration of sampled trajectories at the early epoch. (c) An illustration of sampled trajectories at the final epoch. (d) Collocation points for a grid-based solver where it should predict values of $\psi(x, t)$.

$$L_2(v_\theta, u_\theta) = \int_0^T \mathbb{E}^X \left\| \partial_t v_\theta(X(t), t) - \mathcal{D}_v[v_\theta, u_\theta, X(t), t] \right\|^2 \mathrm{d}t, \tag{12}$$

$$L_3(v_\theta, u_\theta) = \mathbb{E}^X \| u_\theta(X(0), 0) - u_0(X(0)) \|^2 \tag{13}$$

$$L_4(v_\theta, u_\theta) = \mathbb{E}^X \| v_\theta(X(0), 0) - v_0(X(0)) \|^2 \tag{14}$$

where $u_0, v_0$ are defined in Equation (8c). Finally, we define a combined loss using weighted sum with $w_i > 0$:

$$\mathcal{L}(\theta) = \sum_{i=1}^4 w_i L_i(v_\theta, u_\theta). \tag{15}$$

The basic idea of our approach is to, for each iteration $\tau$, to sample new trajectories using Equation (7) with $\nu = 1$. These trajectories are then used to compute stochastic estimates of the loss (15), and then we back-propagate gradients of the loss to update $\theta$. We re-use recently generated trajectories to reduce computational overhead as SDE integration cannot be paralleled. The training procedure is summarized in Algorithm 1 and Figure 1; a more detailed version is presented in Appendix B. We

---

**Algorithm 1** Training algorithm pseudocode

---

**Input** $\psi_0$ – initial wave-function, $M$ – epoch number, $B$ – batch size, other parameters (optimizer parameters, physical constants, Euler-Maryama parameters; see Appendix B)
Initialize NNs $u_{\theta_0}, v_{\theta_0}$
**for** each iteration $0 \le \tau < M$ **do**
    Sample $B$ trajectories using $u_{\theta_\tau}, v_{\theta_\tau}$ via Equation (7) with $\nu = 1$
    Estimate loss $\mathcal{L}(v_{\theta_\tau}, u_{\theta_\tau})$ from Equation (15) over the sampled trajectories
    Back-propagate gradients to get $\nabla_\theta \mathcal{L}(v_{\theta_\tau}, u_{\theta_\tau})$
    Adam optimizer step to get $\theta_{\tau+1}$
**end for**
**output** $u_{\theta_M}, v_{\theta_M}$

---

use trained $u_{\theta_M}, v_{\theta_M}$ to simulate the forward diffusion for $\nu \ge 0$ given $X_0 \sim \mathcal{N}(0, I_d)$:

$$X_{i+1} = X_i + (v_{\theta_M}(X_i, t_i) + \nu u_{\theta_M}(X_i, t_i))\epsilon + \mathcal{N}\left(0, \frac{\hbar}{m} \nu \epsilon I_d\right). \tag{16}$$

In Appendix G, we describe a wide variety of possible ways to apply our approach for estimating an arbitrary quantum observable , singular initial conditions like $\psi_0 = \delta_{x_0}$, singular potentials, correct estimations of observable that involve measurement process, recovering the wave function from $u, v$.

Although PINNs can be used to solve the equations (8a), (8b), that approach would suffer from having fixed sampled density (see Section 5). While ideologically, we, similarly to PINNs, attempt to minimize residuals of PDEs (8a), (8b), we do so on the distribution generated by sampled trajectories $X(t)$, which in turn depends on current neural approximations $v_\theta, u_\theta$. This allows our method to focus only on high-density regions and alleviates the inherent curse of dimensionality that comes from reliance on a grid.

## 4.2 ALGORITHMIC COMPLEXITY

All known deep learning approaches for quantum mechanics suffer from the need to compute the Laplacian (that requires $\mathcal{O}(d^2)$ operations, see (Paszke et al., 2017)), which is the major bottleneck for scaling them to many-particle systems. The following proposition is proved in Appendix D.4:

**Proposition 4.1.** *The algorithmic complexity w.r.t. $d$ of computing $\nabla \langle \nabla, \cdot \rangle$ and equation (9) is $O(d)$.*

So, our formulation of stochastic mechanics with novel equations (8) is much more amenable to automatic differentiation tools than if we developed a neural diffusion approach based on the Nelsonian version that would require $O(d^2)$ operations. The same trick cannot be applied for the classical form of the Schrödinger equation as it relies on the fact that $u, v$ are the full gradients, which is not the case for the wave function $\psi$ itself. That means that the algorithmic complexity of PINN for the Shrödinger equation is $O(d^2)$. It is possible to make PINN work at $\mathcal{O}(d)$ by using stochastic estimators of the trace (Hutchinson, 1989). However, it will introduce a noise of an amplitude $\mathcal{O}(\sqrt{d})$, which will require setting a larger batch size (as $\mathcal{O}(d)$) to offset the noise.

## 4.3 THEORETICAL GUARANTEES

To further justify the effectiveness of our loss function, we prove the following in Appendix F:

**Theorem 4.2.** *(Strong Convergence Bound) We have the following bound between processes $Y$ (the Nelsonian process that samples from $|\psi|^2$) and $X$ (the neural approximation with $v_\theta, u_\theta$):*

$$\sup_{t \leq T} \mathbb{E}\|X(t) - Y(t)\|^2 \leq C_T \mathcal{L}(v_\theta, u_\theta), \tag{17}$$

*where constant $C_T$ is defined explicitly in F.13.*

This theorem means optimizing the loss leads to strong convergence of neural process $X$ to the Nelsonian process $Y$, and that the loss value directly translates into an improvement of $L_2$ error between processes. Constant $C$ depends on horizon $T$ and Lipshitz constants of $u, v, u_\theta, v_\theta$. It also hints that we have a 'low-dimensional' structure when the Lipshitz constants of $u, v, u_\theta, v_\theta$ are $\ll d$, which is the case of low-energy regimes (as large Lipshitz smoothness constant implies large value of the Laplacian and, hence, energy) and with proper selection of neural architecture (Aziznejad et al., 2020).

## 5 EXPERIMENTS

**Experimental setup**  As a baseline, we use an analytical solution (if it is known) or a numerical solution. We compare our method's (DSM) performance with PINNs when possible. Further details on architecture, training procedures, hyperparameters and additional experiments for our approach and PINNs, and numerical solvers can be found in Appendix C and Appendix D. The code of our experiments can be found on GitHub [4].

**Evaluation metrics**  We estimate errors between the true values of the mean and the variance of $X_t$ as the relative $L_2$-norm, namely $\mathcal{E}_m(X)$ and $\mathcal{E}_v(X)$. The standard deviation (confidence intervals) of the observables are indicated in the results. True $v$ and $u$ values are estimated numerically with

---

[4] https://github.com/anon14112358/deep_stochastic_mechanics

the finite difference method. Our trained $u_\theta$ and $v_\theta$ should output these values. We measure errors $\mathcal{E}(u)$ and $\mathcal{E}(v)$ as the $L_2$-norm between the true and predicted values in $L_2(\mathbb{R}^d \times [0, T], \mu)$ with $\mu(\mathrm{d}x, \mathrm{d}t) = |\psi(x, t)|^2 \mathrm{d}x\mathrm{d}t$.

## 5.1 HARMONIC OSCILLATOR

We consider a harmonic oscillator model with $x \in \mathbb{R}^1$, $V(x) = \frac{1}{2}m\omega^2(x - 0.1)^2$, $t \in [0, 1]$ and where $m = 1$ and $\omega = 1$. The initial wave function is given as $\psi(x, 0) \propto e^{-x^2/(4\sigma^2)}$. Then $u_0(x) = -\frac{\hbar x}{2m\sigma^2}$, $v_0(x) \equiv 0$. $X(0)$ comes from $X(0) \sim \mathcal{N}(0, \sigma^2)$, where $\sigma^2 = 0.1$.

We use the numerical solution as the ground truth. Our approach is compared with a PINN. The PINN input data consists of $N_0 = 1000$ data points sampled for estimating $\psi(x, 0)$, $N_b = 300$ data points for enforcing the boundary conditions (we assume zero boundary conditions), and $N_f = 60000$ collocation points to enforce the corresponding equation inside the solution domain, all points sampled uniformly for $x \in [-2, 2]$ and $t \in [0, 1]$.

Figure 2 (a) summarizes the results of our experiment. The left panel of the figure illustrates the evolution of the density $|\psi(x, t)|^2$ over time for different methods. It is evident that our approach accurately captures the density evolution, while the PINN model initially aligns with the ground truth but deviates from it over time. Sampling collocation points uniformly when density is concentrated in a small region explains why PINN struggles to learn the dynamics of Equation (1); we illustrate this effect in Figure 1 (d). The right panel demonstrates observables of the system, the averaged mean of $X_t$ and the averaged variance of $X_t$. Our approach consistently follows the corresponding distribution of $X_t$. On the contrary, the predictions of the PINN model only match the distribution at the initial time steps but fail to accurately represent it as time elapses. Table 2 shows the error rates for our method and PINNs. In particular, our method performs better in terms of all error rates than the PINN. These findings emphasize the better performance of the proposed method in capturing the dynamics of the Schrödinger equation compared to the PINN model.

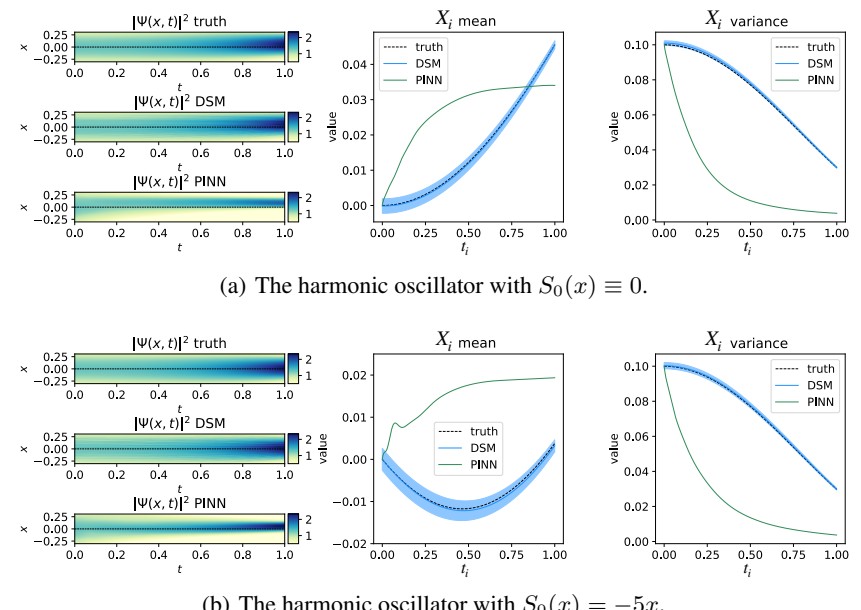

(a) The harmonic oscillator with $S_0(x) \equiv 0$.

(b) The harmonic oscillator with $S_0(x) = -5x$.

Figure 2: The results for 1d harmonic oscillator. DSM corresponds to our method.

We also consider a non-zero initial phase $S_0(x) = -5x$. It corresponds to the initial impulse of a particle. Then $v_0(x) \equiv -\frac{5\hbar}{m}$. The PINN inputs are $N_0 = 3000$, $N_b = 300$ data points, and $N_f = 80000$ collocation points. Figure 2 (b) and Table 2 present the results of our experiment. Our method consistently follows the corresponding ground truth while the PINN model fails to do so. It indicates the ability of our method to accurately model the behavior of the quantum system. In

Table 2: The results for different harmonic oscillator settings. In the 3d setting, the reported errors are averaged across the dimensions. The **best** result is in bold.

| Problem | Model | $\mathcal{E}_m(X_i)\downarrow$ | $\mathcal{E}_v(X_i)\downarrow$ | $\mathcal{E}(v)\downarrow$ | $\mathcal{E}(u)\downarrow$ |
|---|---|---|---|---|---|
| 1d, $S_0(x)\equiv 0$ | PINN | 0.698 | 0.701 | 25.861 | 3.621 |
| | DSM | $\mathbf{0.077 \pm 0.052}$ | $\mathbf{0.011 \pm 0.006}$ | **0.00011** | $\mathbf{2.811 \times 10^{-5}}$ |
| | Gaussian sampling | $0.294 \pm 0.152$ | $0.488 \pm 0.018$ | 3.198 | 1.185 |
| 1d, $S_0(x) = -5x$ | PINN | 2.819 | 0.674 | 281.852 | 68.708 |
| | DSM | $\mathbf{0.223 \pm 0.207}$ | $\mathbf{0.009 \pm 0.008}$ | $\mathbf{1.645 \times 10^{-5}}$ | $\mathbf{2.168 \times 10^{-5}}$ |
| | Gaussian sampling | $0.836 \pm 0.296$ | $0.086 \pm 0.007$ | 77.578 | 24.152 |
| 3d, $S_0(x)\equiv 0$ | DSM (Nelsonian) | $0.100 \pm 0.061$ | $0.012 \pm 0.009$ | $1.200\times 10^{-4}$ | $\mathbf{3.324 \times 10^{-5}}$ |
| | DSM (Grad. Divergence) | $\mathbf{0.073 \pm 0.048}$ | $\mathbf{0.011 \pm 0.008}$ | $\mathbf{4.482 \times 10^{-5}}$ | $4.333 \times 10^{-5}$ |
| | Gaussian sampling | $0.459 \pm 0.126$ | $5.101 \pm 0.201$ | 13.453 | 5.063 |
| Interacting system | PINN | 0.100 | 2.350 | 29.572 | 2.657 |
| | DSM | $\mathbf{0.091 \pm 0.050}$ | $\mathbf{0.102 \pm 0.020}$ | $\mathbf{6.511 \times 10^{-5}}$ | $\mathbf{5.139 \times 10^{-5}}$ |

addition, we consider an oscillator model with three non-interacting particles, which can be seen as 3d system. The results are given in Table 2 and Appendix D.2.

## 5.2 INTERACTING SYSTEM

Next, we consider a system of two interacting bosons in a harmonic trap with a soft contact term $V(x_1, x_2) = \frac{1}{2}m\omega^2(x_1^2 + x_2^2) + \frac{g}{2}\frac{1}{\sqrt{2\pi\sigma^2}}e^{-(x_1-x_2)^2/(2\sigma^2)}$ and initial condition $\psi_0 \propto e^{-m\omega^2 x^2/(2\hbar)}$. We use $\omega = 1$, $T = 1$, $\sigma^2 = 0.1$, and $N = 1000$. The term $g$ controls interaction strength. When $g = 0$, there is no interaction, and $\psi_0$ is the groundstate of the corresponding Hamiltonian $\mathcal{H}$. We use $g = 1$ in our simulations. Figure 3 shows simulation results: our method follows the corresponding ground truth while PINN fails over time. As $t$ increases, the variance of $X_i$ for PINN either decreases or remains relatively constant, contrasting with the dynamics that exhibit more divergent behavior. We hypothesize that such discrepancy in the performance of PINN, particularly in matching statistics, is due to the design choice. Specifically, the output predictions, $\psi(x_i, t)$, made by PINNs are not constrained to adhere to physical meaningfulness, meaning $\int_{\mathbb{R}^d} |\psi(x,t)|^2 \mathrm{d}x$ does not always equal 1, making uncontrolled statistics.

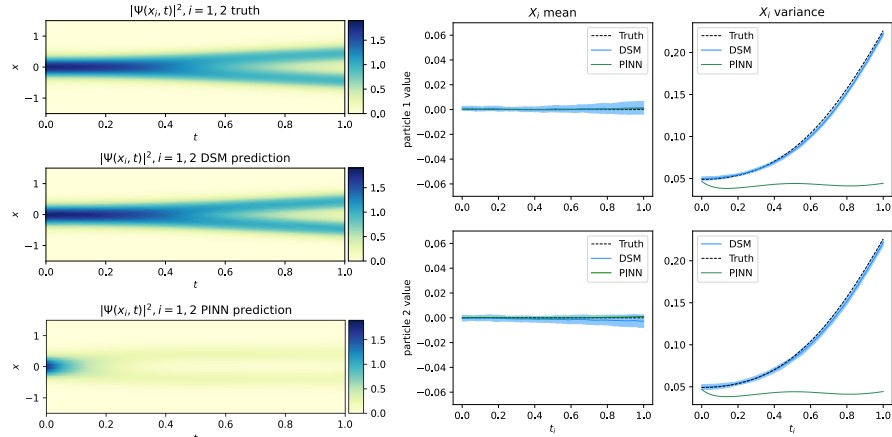

Figure 3: The results for two interacting particles. DSM corresponds to our method.

## 5.3 NAIVE SAMPLING

To further evaluate our approach, we consider the following sampling scheme: it is possible to replace all measures in the expectations from Equation (15) with a Gaussian noise $\mathcal{N}(0, 1)$. Minimising

this loss perfectly would imply that the PDE is satisfied for all values $x, t$. Table 2 shows worse quantitative results compared to our approach. More detailed results, including the singular initial condition and 3d harmonic oscillator setting, are given in Appendix D.3.

## 5.4 ALGORITHMIC COMPLEXITY

We measure training time per epoch for two versions of the DSM algorithm for $d = 1, 3, 5, 7, 9$: the Nelsonian one and our version. The experiments are conducted using the harmonic oscillator model with $S_0(x) \equiv 0$. The results are averaged across 30 runs. Figure 4 on the left shows the results. It demonstrates quadratic time per iteration scaling for the Nelsonian version, while the time grows linearly for our version. The memory complexity results are given in Appendix D.4.

Figure 4 on the right illustrates the total training time versus the problem dimension. We train our models until the training loss reaches a threshold of $2.5 \times 10^{-5}$. We observe that train time grows linearly with $d$. The performance errors are presented in Appendix D.4. These empirical findings demonstrate the computational efficiency of our algorithm. In contrast, traditional numerical solvers would suffer from exponential growth in data when tackling this task.

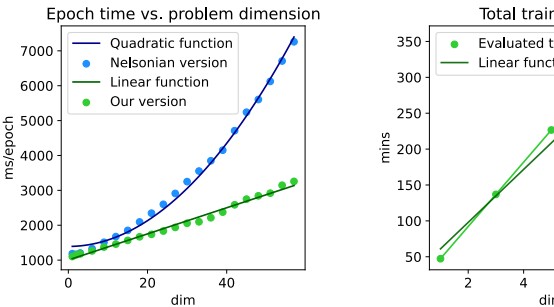

Figure 4: Empirical complexity evaluation of our method.

## 6 DISCUSSION AND LIMITATIONS

**Limitations** This paper considers the simplest case of the linear spinless Schrödinger equation on a flat manifold $\mathbb{R}^d$ with a smooth potential. For many practical setups, such as quantum chemistry, quantum computing or condensed matter physics, our approach should be modified, e.g., by adding a spin component or by considering some approximation and, therefore, requires additional validations that are beyond of the scope of this work. We have shown evidence of adaptation of our method to one kind of low-dimensional structure, but this paper does not explore a broader range of systems with low latent dimension.

**Broader impacts** It is hypothesized that simulations of quantum systems cannot be done effectively on classic computers, otherwise known as the problem of P $\neq$ BQP (Bernstein & Vazirani, 1997). If that is true, then no algorithm should scale as a polynomial of the dimension for *all* problems. In our work, we propose the algorithm that can simulate *some* systems effectively in linear time. It may be possible to learn polynomial-time approximations to Shor's algorithm (Shor, 1997) on a classic computer using some modification of the proposed deep learning approach. While this possibility is highly unlikely, the risk that comes with it (Bernstein & Lange, 2017) should not be ignored.

**Conclusion and future work** We develop the new algorithm for simulating quantum mechanics that addresses the curse of dimensionality by leveraging the latent low-dimensional structure of the system. This approach is based on a modification of the stochastic mechanics theory that establishes a correspondence between the Schrödinger equation and a diffusion process. We learn the drifts of this diffusion process using deep learning to sample from the corresponding quantum density. We believe that our approach has the potential to bring to quantum mechanics simulation the same progress that deep learning has enabled in artificial intelligence. We provide future work discussion in Appendix I.

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
