# A  NOTATION

- $\langle a, b \rangle = \sum_{i=1}^{d} a_i b_i$ for $a, b \in \mathbb{R}^d$ – a scalar product.
- $\|a\| = \sqrt{\langle a, a \rangle}$ for $a \in \mathbb{R}^d$ – a norm.
- $\text{Tr}(A) = \sum_{i=1}^{d} a_{ii}$ for a matrix $A = \left[a_{ij}\right]_{i=1, j=1}^{d,d}$.
- $A(t), B(t), C(t), \ldots$ – stochastic processes indexed by time $t \geq 0$.
- $A_i, B_i, C_i, \ldots$ – approximations to those processes.
- $a, b, c$ – other variables.
- $\mathbf{A}, \mathbf{B}, \mathbf{C}, \ldots$ – quantum observables, e.g., $\mathbf{X}(t)$ – result of quantum measurement of the coordinate of the particle at moment $t$.
- $\rho_A(x, t)$ – density probability of a process $A(t)$ at time $t$.
- $\psi(x, t)$ – a wave function.
- $\psi_0 = \psi(x, 0)$ – an initial wave function.
- $\rho(x, t) = \left|\psi(x, t)\right|^2$ – a quantum density.
- $\rho_0(x) = \rho(x, 0)$ – an initial probability distribution.
- $\psi(x, t) = \sqrt{\rho(x, t)} e^{iS(x,t)}$ where $S(x, t)$ – a single-valued representative of the phase of the wave function.
- $\nabla = \left(\frac{\partial}{\partial x_1} \cdot, \ldots, \frac{\partial}{\partial x_d} \cdot\right)$ – a gradient operator. If $f : \mathbb{R}^d \to \mathbb{R}^m$, then $\nabla f(x) \in \mathbb{R}^{d \times m}$ is the Jacobian of $f$, in the case of $m = 1$ we call it a gradient of $f$.
- $\nabla^2 = \left[\frac{\partial^2}{\partial x_i \partial x_j}\right]_{i=1, j=1}^{d,d}$ – the Hessian operator.
- $\nabla^2 \cdot A = \left[\frac{\partial^2}{\partial x_i \partial x_j} a_{ij}\right]_{i=1, j=1}^{d,d}$ for $A = \left[a_{ij}(x)\right]_{i=1, j=1}^{d,d}$.
- $\langle \nabla, \cdot \rangle$ – a divergence operator, e.g., for $f : \mathbb{R}^d \to \mathbb{R}^d$ we have $\langle \nabla, f(x) \rangle = \sum_{i=1}^{d} \frac{\partial}{\partial x_i} f_i(x)$.
- $\Delta = \text{Tr}(\nabla^2)$ – the Laplace operator.
- $m$ – a mass tensor (or a scalar mass).
- $\hbar$ – the reduced Planck's constant.
- $\partial_y = \frac{\partial}{\partial y}$ – a short-hand notation for a partial derivative operator.
- $\left[A, B\right] = AB - BA$ – a commutator of two operators. If one of the arguments is a scalar function, we consider a scalar function as a point-wise multiplication operator.
- $|z| = \sqrt{x^2 + y^2}$ for a complex number $z = x + iy \in \mathcal{C}, x, y \in \mathbb{R}$.
- $\mathcal{N}(\mu, C)$ – a Gaussian distribution with mean $\mu \in \mathbb{R}^d$ and covariance $C \in \mathbb{R}^{d \times d}$.
- $A \sim \rho$ means that $A$ is a random variable with distribution $\rho$. We do not differentiate between "sample from" and "distributed as", but it is evident from context when we consider samples from distribution versus when we say that something has such distribution.
- $\delta_x$ – delta-distribution concentrated at $x$. It is a generalized function corresponding to the "density" of a distribution with a singular support $\{x\}$.

# B  DSM ALGORITHM

We present detailed versions of our method: Algorithm 2 for batch generation and Algorithm 3 for training. During inference, distributions of $X_i$ converge to $\rho = |\psi|^2$ and thus we obtain the desired outcome. Furthermore, solving (8a) on points generated by the current best approximations of $u, v$, the method exhibits self-adaptation behavior: the method obtains its current belief where $X(t)$ is concentrated, updates its belief and iterates accordingly. With each iteration of the inference, the method focuses more on high-concentration regions of $\rho$.

---

**Algorithm 2** GenerateBatch($u, v, \rho_0, \nu, T, B, N$) – sample trajectories

---

**Physical hyperparams:** $T$ – time horizon, $\psi_0$ – initial wave-function.
**Hyperparams:** $\nu \geq 0$ – diffusion constant, $B \geq 1$ – batch size, $N \geq 1$ – time grid size.
$t_i = iT/N$ for $0 \leq i \leq N$
sample $X_{0j} \sim |\psi_0|^2$ for $1 \leq jB$
**for** $1 \leq i \leq N$ **do**
    sample $\xi_j \sim \mathcal{N}(0, I_d)$ for $1 \leq j \leq B$
    $X_{ij} = X_{(i-1)j} + \frac{T}{N}\left(v_\theta(X_{(i-1)j}, t_{i-1}) + \nu u_\theta(X_{(i-1)j}, t_{i-1})\right) + \sqrt{\frac{\nu\hbar T}{mN}}\xi_j$ for $1 \leq j \leq B$
**end for**
**output** $\left\{\left\{X_{ij}\right\}_{j=1}^{B}\right\}_{i=0}^{N}$

---

**Algorithm 3** A training algorithm

---

**Physical hyperparams:** $m > 0$ – mass, $\hbar > 0$ – reduced Planck constant, $T$ – a time horizon, $\psi_0 : \mathbb{R}^d \to \mathbb{C}$ – an initial wave function, $V : \mathbb{R}^d \times [0, T] \to \mathbb{R}$ – potential.
**Hyperparams:** $\eta > 0$ – learning rate for backprop, $\nu > 0$ – diffusion constant, $B \geq 1$ – batch size, $M \geq 1$ – optimization steps, $N \geq 1$ – time grid size, $w_u, w_v, w_0 > 0$ – weights of losses.
**Instructions:**
$t_i = iT/N$ for $0 \leq i \leq N$
**for** $1 \leq \tau \leq M$ **do**
    $X = \text{GenerateBatch}(u_{\theta_{\tau-1}}, v_{\theta_{\tau-1}}, \psi_0, \nu, T, B, N)$
    define $L_\tau^u(\theta) = \frac{1}{(N+1)B}\sum_{i=0}^{N}\sum_{j=1}^{B}\left\|\partial_t u_\theta(X_{ij}, t_i) - \mathcal{D}_u[u_\theta, v_\theta, X_{ij}, t_i]\right\|^2$
    define $L_\tau^v(\theta) = \frac{1}{(N+1)B}\sum_{i=0}^{N}\sum_{j=1}^{B}\left\|\partial_t v_\theta(X_{ij}, t_i) - \mathcal{D}_v[u_\theta, v_\theta, X_{ij}, t_i]\right\|^2$
    define $L_\tau^0(\theta) = \frac{1}{B}\sum_{j=1}^{B}\left(\left\|u_\theta(X_{0j}, t_0) - u_0(X_{0j})\right\|^2 + \left\|v_\theta(X_{0j}, t_0) - v_0(X_{0j}, t_0)\right\|^2\right)$
    define $\mathcal{L}_\tau(\theta) = w_u L_\tau^u(\theta) + w_v L_\tau^v(\theta) + w_0 L_\tau^0(\theta)$
    $\theta_\tau = \text{OptimizationStep}(\theta_{\tau-1}, \nabla_\theta \mathcal{L}_\tau(\theta_{\tau-1}), \eta)$
**end for**
**output** $u_{\theta_M}, v_{\theta_M}$

---

## C   EXPERIMENTAL SETUP DETAILS

In our experiments, we set $m = 1$, $\hbar = 10^{-2}$[5], $\sigma^2 = 10^{-1}$. For the harmonic oscillator model, $N = 1000$ and the batch size $B = 100$; for the singular initial condition problem, $N = 100$ and $B = 100$. For evaluation, our method samples 10 000 points per time step, and the observables are estimated from these samples; we run the model this way ten times.

### C.1   A NUMERICAL SOLUTION

**1d harmonic oscillator with $S_0(x) \equiv 0$:**   To evaluate our method's performance, we use a numerical solver that integrates the corresponding differential equation given the initial condition. We use SciPy library (Virtanen et al., 2020). The solution domain is $x \in [-2, 2]$ and $t \in [0, 1]$, where $x$ is split into 566 points and $t$ into 1001 time steps. This solution can be repeated $d$ times for the $d$-dimensional harmonic oscillator problem.

**1d harmonic oscillator with $S_0(x) = -5x$:**   We use the same numerical solver as for the $S_0(x) \equiv 0$ case. The solution domain is $x \in [-2, 2]$ and $t \in [0, 1]$, where $x$ is split into 2829 points and $t$ is split into 1001 time steps.

### C.2   ARCHITECTURE AND TRAINING DETAILS

A basic NN architecture for our approach and the PINN is a feed-forward NN with one hidden layer with hyperbolic tangent activation functions. We represent the velocities $u$ and $v$ using the basic NN architecture with 200 neurons in the case of the singular initial condition. The training process takes about 7 mins. For $d = 1$ harmonic oscillator with zero initial phase problem, there are 200 neurons for our method and 400 for the PINN; for $d = 3$ and more dimensions, we use 400 neurons. This rule holds for the experiments measuring total training time in Section 5.4. In a $d = 1$ harmonic oscillator with a non-zero initial phase problem, we use 300 hidden neurons in our models. In the experiments devoted to measuring time per epoch (from Section 5.4), the number of hidden neurons is fixed to 200 for all dimensions. We use the Adam optimizer (Kingma & Ba, 2014) with a learning rate $10^{-4}$. In our experiments, we set $w_u = 1, w_v = 0.8, w_0 = 1$. For PINN evaluation, the test sets are the same as the grid for the numerical solver. In our experiments, we usually use a single NVIDIA A40 GPU. For the results reported in Section 5.4, we use an NVIDIA A100 GPU.

### C.3   ON OPTIMIZATION

We use Adam optimizer (Kingma & Ba, 2014) in our experiments. Since the operators (9) are not linear, we may not be able to claim convergence to the global optima of such methods as SGD or Adam in the Neural Tangent Kernel (NTK) (Jacot et al., 2018) limit. Such proof exists for PINNs in Wang et al. (2022) due to the linearity of the Schrödinger equation (1). It is possible that non-linearity in the loss (15) requires non-convex methods to achieve theoretical guarantees on convergence to the global optima (Raginsky et al., 2017; Muzellec et al., 2020). Further research into NTK and non-linear PDEs is needed (Wang et al., 2022).

The only noise source in our loss (15) comes from trajectory sampling. This fact contrasts sharply with generative diffusion models relying on score matching (Yang et al., 2022). In these models, the loss has $\mathcal{O}(\epsilon^{-1})$ variance as it implicitly attempts to numerically estimate the stochastic differential $\frac{X(t+\epsilon)-X(t)}{\epsilon}$ which leads to $\frac{1}{\sqrt{\epsilon}}$ contribution from increments of the Wiener process. In our loss, the stochastic differentials are evaluated analytically in Equation (9) avoiding such contributions; for details, see Nelson (1966; 2005). This leads $\mathcal{O}(1)$ variance of the gradient and, thus, allows us to achieve fast convergence with smaller batches.

---

[5]The value of the reduced Plank constant depends on the metric system that we use and, thus, for our evaluations we are free to choose any value.

# D EXPERIMENTS

## D.1 SINGULAR INITIAL CONDITIONS

As a proof of concept, we consider a case of one particle $x \in \mathbb{R}^1$ with $V(x) \equiv 0$ and $\psi_0 = \delta_0$, $t \in [0, 1]$. Since $\delta$-function is a generalized function, we must take a $\delta$-sequence for the training. The most straightforward approach is to take $\widetilde{\psi_0} = \frac{1}{(2\pi\alpha)^{\frac{1}{4}}} e^{-\frac{x^2}{4\alpha}}$ with $\alpha \to 0_+$. In our experiments we take $\alpha = \frac{\hbar^2}{m^2}$, yielding $v_0(x) \equiv 0$ and $u_0(x) = -\frac{\hbar x}{2m\alpha}$. Since $\psi_0$ is singular, we must set $\nu = 1$ during sampling. The analytical solution is given as $\psi(x, t) = \frac{1}{(2\pi t)^{\frac{1}{4}}} e^{-\frac{x^2}{4t}}$. So, we expect the standard deviation of $X(t)$ to grow as $\sqrt{t}$, and the mean value of $X(t)$ to be zero.

We do not compare our approach with PINNs since it is a simple proof of concept, and the analytical solution is known. Figure 5 summarizes the results of our experiment. Specifically, the left panel of the figure shows the magnitude of the density obtained with our approach alongside the true density. The right panel of Figure 5 shows statistics of $X_t$, such as mean and variance, and the corresponding error bars. The resulting prediction errors are calculated against the truth data for this problem and are measured at $0.008 \pm 0.007$ in the $L_2$-norm for the averaged mean and $0.011 \pm 0.007$ in the relative $L_2$-norm for the averaged variance of $X_t$. Our approach can accurately capture the behavior of the Schrödinger equation in the singular initial condition case.

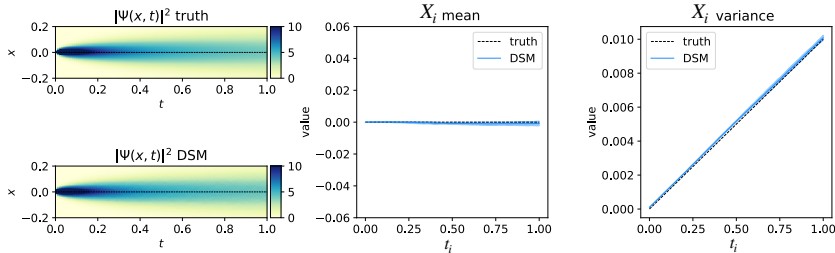

Figure 5: Results for the singular initial condition case. DSM corresponds to our method.

## D.2 3D HARMONIC OSCILLATOR

We further explore our approach by considering the harmonic oscillator model with $S_0(x) \equiv 0$ with three non-interacting particles. This setting can be viewed as a 3d problem, where the solution is a 1d solution repeated three times. Due to computational resource limitations, we are unable to execute the PINN model. The number of collocation points should grow exponentially with the problem dimension so that the PINN model converges. We have about 512 GB of memory but cannot store $60000^3$ points. We conduct experiments comparing two versions of the proposed algorithm: the Nelsonian one and our version. Table 2 provides the quantitative results of these experiments. Our version demonstrates slightly better performance compared to the Nelsonian version, although the difference is not statistically significant. Empirically, our version requires more steps to converge compared to the Nelsonian version: 7000 vs. 9000 epochs correspondingly. However, the training time of the Nelsonian approach is about 20 mins longer than our approach's time.

Figure 6 demonstrates the obtained statistics with the proposed algorithm's two versions (Nelsonian and Gradient Divergence) for every dimension. Figure 7 compares the density function for every dimension for these two versions. Table 3 summarizes the error rates per dimension. The results suggest no significant difference in the performance of these two versions of our algorithm. As mentioned in the main text, the Gradient Divergence version tends to require more steps to converge, but it has linear time complexity in contrast to the quadratic complexity of the Nelsonian version.

## D.3 NAIVE SAMPLING

Figure 8 shows performance of Gaussian sampling approach applied to the harmonic oscillator and the singular initial condition setting. Table 4 compares results of all methods. Our approach converges

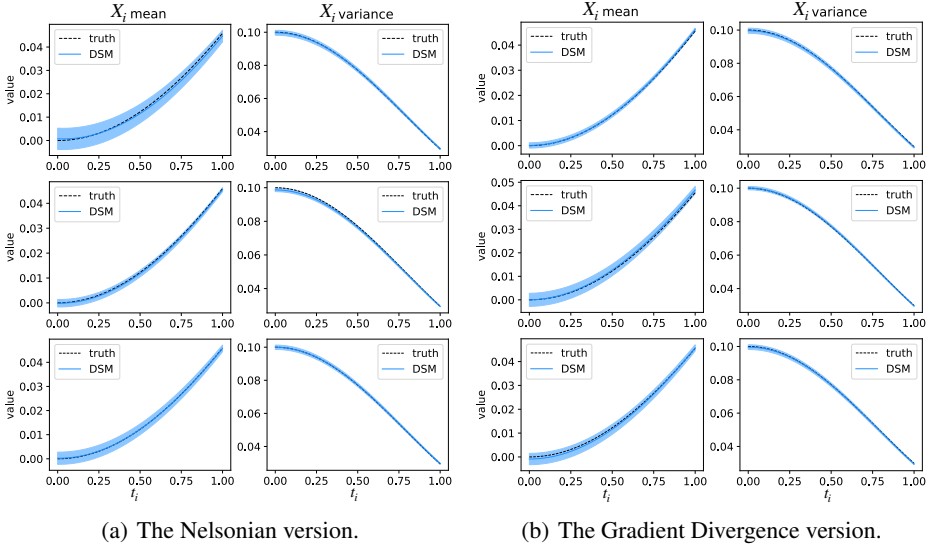

(a) The Nelsonian version.

(b) The Gradient Divergence version.

Figure 6: The obtained statistics for 3d harmonic oscillator using two versions of the proposed approach.

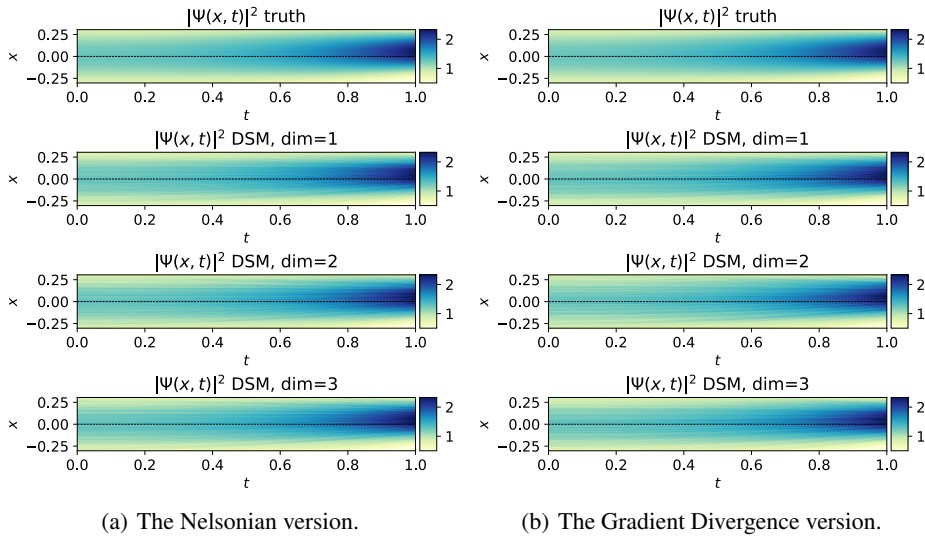

(a) The Nelsonian version.

(b) The Gradient Divergence version.

Figure 7: The density function for 3d harmonic oscillator using two versions of the proposed approach.

to the ground truth while naive sampling does not. Figure 8 illustrates performance of Gaussian sampling.

### D.4 COMPUTATIONAL COMPLEXITY

*Proposition 4.1.* Computing a forward pass of $u_\theta, v_\theta$ scales as $\mathcal{O}(d)$ by their design. What we need is to prove that the loss function (15) can be computed in $\mathcal{O}(d)$. we have two kind of operators appear there $\langle \nabla \cdot, \cdot \rangle$ and $\nabla \langle \nabla, \cdot \rangle$.

The first one is pure Jacobian-vector product, thus, there is an algorithm to estimate it with linear complexity (assuming the forward pass has linear complexity), see (Griewank & Walther, 2008).

Table 3: The results for 3d harmonic oscillator with $S_0(x) \equiv 0$ using two versions of the proposed approach: the Nelsonian one uses the Laplacian operator in the training loss, the Gradient Divergence version is our modification that replaces Laplacian with gradient of divergence.

| Model | $\mathcal{E}_m(X_i^{(1)})\downarrow$ | $\mathcal{E}_m(X_i^{(2)})\downarrow$ | $\mathcal{E}_m(X_i^{(3)})\downarrow$ | $\mathcal{E}_m(X_i)\downarrow$ |
|---|---|---|---|---|
| DSM (Nelsonian) | $0.170 \pm 0.081$ | $0.056 \pm 0.030$ | $\mathbf{0.073 \pm 0.072}$ | $0.100 \pm 0.061$ |
| DSM (Gradient Divergence) | $\mathbf{0.038 \pm 0.023}$ | $\mathbf{0.100 \pm 0.060}$ | $0.082 \pm 0.060$ | $\mathbf{0.073 \pm 0.048}$ |

| Model | $\mathcal{E}_v(X_i^{(1)})\downarrow$ | $\mathcal{E}_v(X_i^{(2)})\downarrow$ | $\mathcal{E}_v(X_i^{(3)})\downarrow$ | $\mathcal{E}_v(X_i)\downarrow$ |
|---|---|---|---|---|
| DSM (Nelsonian) | $\mathbf{0.012 \pm 0.009}$ | $0.012 \pm 0.009$ | $0.011 \pm 0.008$ | $0.012 \pm 0.009$ |
| DSM (Gradient Divergence) | $\mathbf{0.012 \pm 0.010}$ | $\mathbf{0.009 \pm 0.005}$ | $\mathbf{0.011 \pm 0.010}$ | $\mathbf{0.011 \pm 0.008}$ |

| Model | $\mathcal{E}(v^{(1)})\downarrow$ | $\mathcal{E}(v^{(2)})\downarrow$ | $\mathcal{E}(v^{(3)})\downarrow$ | $\mathcal{E}(v))\downarrow$ |
|---|---|---|---|---|
| DSM (Nelsonian) | $0.00013$ | $0.00012$ | $0.00012$ | $0.00012$ |
| DSM (Gradient Divergence) | $\mathbf{4.346 \times 10^{-5}}$ | $\mathbf{4.401 \times 10^{-5}}$ | $\mathbf{4.700 \times 10^{-5}}$ | $\mathbf{4.482 \times 10^{-5}}$ |

| Model | $\mathcal{E}(u^{(1)})\downarrow$ | $\mathcal{E}(v^{(2)})\downarrow$ | $\mathcal{E}(v^{(3)})\downarrow$ | $\mathcal{E}(v)\downarrow$ |
|---|---|---|---|---|
| DSM (Nelsonian) | $\mathbf{4.441 \times 10^{-5}}$ | $\mathbf{2.721 \times 10^{-5}}$ | $2.810 \times 10^{-5}$ | $\mathbf{3.324 \times 10^{-5}}$ |
| DSM (Gradient Divergence) | $6.648 \times 10^{-5}$ | $4.405 \times 10^{-5}$ | $\mathbf{1.915 \times 10^{-5}}$ | $4.333 \times 10^{-5}$ |

Table 4: Error rates for different problem settings using two sampling schemes: our (DSM) and Gaussian sampling. Gaussian sampling replaces all measures in the expectations with Gaussian noise in Equation (15). The **best** result is in bold. These results demonstrate that our approach work better than the naïve sampling scheme.

| Problem | Model | $\mathcal{E}_m(X_i)\downarrow$ | $\mathcal{E}_v(X_i)\downarrow$ | $\mathcal{E}(v)\downarrow$ | $\mathcal{E}(u)\downarrow$ |
|---|---|---|---|---|---|
| Singular IC | Gaussian sampling | $0.043 \pm 0.042$ | $0.146 \pm 0.013$ | $1.262$ | $0.035$ |
| | DSM | $\mathbf{0.008 \pm 0.007}$ | $\mathbf{0.011 \pm 0.007}$ | $\mathbf{0.524}$ | $\mathbf{0.008}$ |
| Harm osc 1d, $S_0(x) \equiv 0$ | Gaussian sampling | $0.294 \pm 0.152$ | $0.488 \pm 0.018$ | $3.19762$ | $1.18540$ |
| | DSM | $\mathbf{0.077 \pm 0.052}$ | $\mathbf{0.011 \pm 0.006}$ | $\mathbf{0.00011}$ | $\mathbf{2.811 \times 10^{-5}}$ |
| Harm osc 1d, $S_0(x) = -5x$ | Gaussian sampling | $0.836 \pm 0.296$ | $0.086 \pm 0.007$ | $77.57819$ | $24.15156$ |
| | DSM | $\mathbf{0.223 \pm 0.207}$ | $\mathbf{0.009 \pm 0.008}$ | $\mathbf{1.645 \times 10^{-5}}$ | $\mathbf{2.168 \times 10^{-5}}$ |
| Harm osc 3d, $S_0(x) \equiv 0$ | Gaussian sampling | $0.459 \pm 0.126$ | $5.101 \pm 0.201$ | $13.453$ | $5.063$ |
| | DSM | $\mathbf{0.073 \pm 0.048}$ | $\mathbf{0.011 \pm 0.008}$ | $\mathbf{4.482 \times 10^{-5}}$ | $\mathbf{4.333 \times 10^{-5}}$ |

For the second type, we observe that $\nabla\langle\nabla, \cdot\rangle = \langle\nabla\langle\nabla\cdot, \mathbb{1}_d\rangle, \mathbb{1}_d\rangle$. Thus, this is a composition of two Jacobian-vector products and by applying the same result twice we obtain that its complexity scales as a constant multiply of the forward pass complexity. $\qquad\square$

We empirically estimate memory allocation on a GPU (NVIDIA A100) when training two versions of the proposed algorithm. In addition, we estimate the number of epochs until the training loss function is less than $10^{-2}$ for different problem dimensions. The results are visualized in Figure 9(a) proves the memory usage of the Gradient Divergence version grows linearly with the dimension while it grows quadratically in the Nelsonian version. We also empirically access the convergence speed of two versions of our approach. Figure 9(b) shows how many epochs are needed to make the training loss less than $1 \times 10^{-2}$. Usually, the Gradient Divergence version requires slightly more epochs to converge to this threshold than the Nelsonian one. The number of epochs is averaged across five runs. In both experiments, the setup is the same as we describe in Section 5.4.

Also, we provide more details on the experiment measuring the total training time per dimensions $d = 1, 3, 5, 7, 9$. This experiment is described in Section 5.4, and the training time grows linearly with the problem dimension. Table 5 presents the error rates and train time. The results show that the proposed approach can perform well for every dimension while the train time scales linearly with the problem dimension.

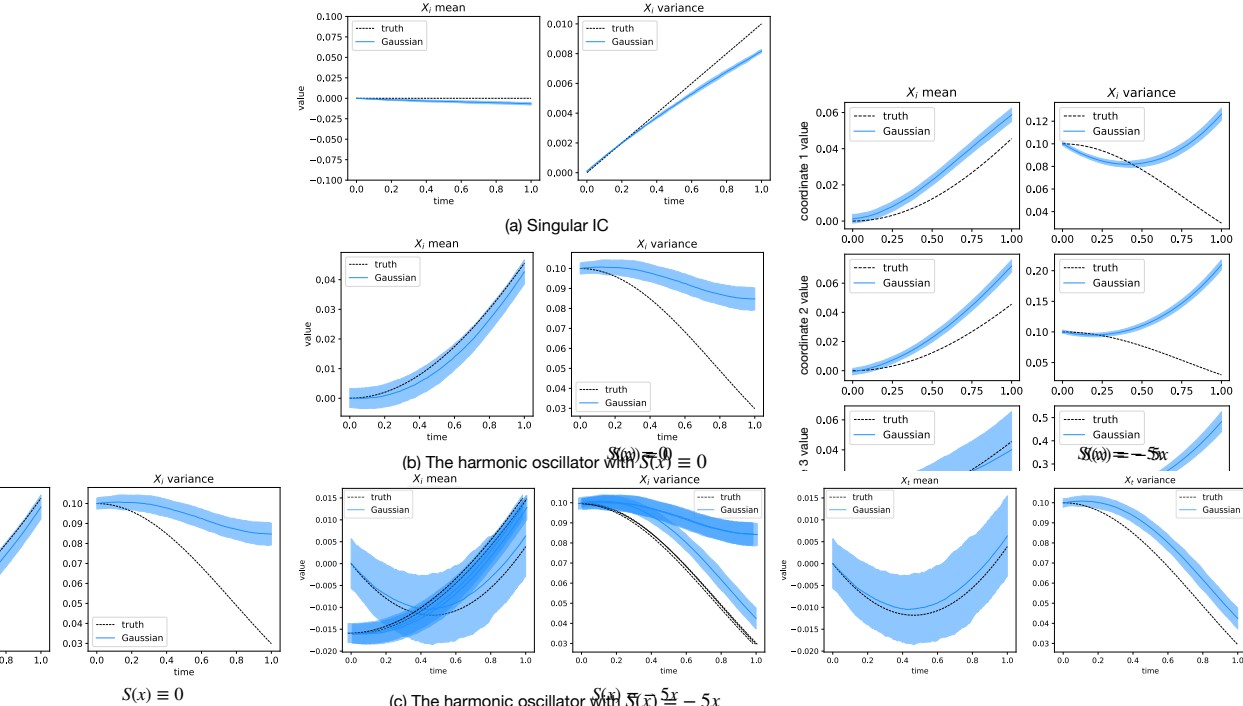

(a) Singular IC

(b) The harmonic oscillator with $S(x) \equiv 0$

(c) The harmonic oscillator with $S(\bar{x}) \stackrel{5x}{=} - 5x$

Figure 8: An illustration of produced trajectories using the naïve Gaussian sampling scheme as a comparison with the proposed approach. The obtained trajectories do not match the solution, while the results in our paper suggest that the proposed DSM approach converges better. Compare with Figures 5, 2, 6.

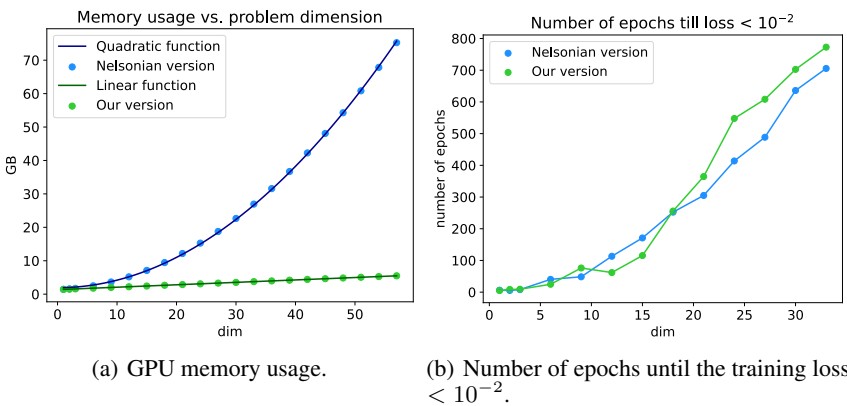

(a) GPU memory usage.

(b) Number of epochs until the training loss $< 10^{-2}$.

Figure 9: Empirical complexity evaluation of two versions of the proposed method: memory usage and the number of epochs until the loss is less than the threshold.

### D.5 INTERACTING SYSTEM

As a numerical solver, we use the qmsolve library [6]. The solution domain is $x \in [-1.5, 1.5]$ and $t \in [0, 1]$, where $x$ is split into 100 points and $t$ into 1001 time steps. The PINN inputs are $N_0 = 10000$, $N_b = 1000$ data points, and $N_f = 1000000$ collocation points. Figure 10 shows performance of both methods for the interacting particles problem.

---

[6]https://github.com/quantum-visualizations/qmsolve

Table 5: Training time and test errors for the harmonic oscillator model for different $d$.

| $d$ | $\mathcal{E}_m(X_i)\downarrow$ | $\mathcal{E}_v(X_i)\downarrow$ | $\mathcal{E}(v)\downarrow$ | $\mathcal{E}(u)\downarrow$ | Train time |
|---|---|---|---|---|---|
| 1 | $0.074 \pm 0.052$ | $0.009 \pm 0.007$ | 0.00012 | 2.809e-05 | 46m 20s |
| 3 | $0.073 \pm 0.048$ | $0.010 \pm 0.008$ | 4.479e-05 | 3.946e-05 | 2h 18m |
| 5 | $0.081 \pm 0.057$ | $0.009 \pm 0.008$ | 4.956e-05 | 4.000e-05 | 3h 10m |
| 7 | $0.085 \pm 0.060$ | $0.011 \pm 0.009$ | 5.877e-05 | 4.971e-05 | 3h 40m |
| 9 | $0.096 \pm 0.081$ | $0.011 \pm 0.009$ | 7.011e-05 | 6.123e-05 | 4h 46m |

**Permutation invariance.** As the system is symmetric (we have two bosons), we enforce symmetry for both DSM and PINNs. In particular, the neural network inputs $x$ are sorted, ensuring that the models are permutation invariant. On one hand, such approach helps with physical property of system being satisfied. One the other hand, sorting $x$ increases computational time, and it might be preferable to avoid in higher dimensions. For the two interacting particles system, the performance difference between a regular and permutation invariant architectures is not so significant, though.

**Architecture of the neural network.** Instead of a multi-layer perceptron as used in Raissi et al. (2019), we followed the design choice of Jiang & Willett (2022) to use residual connection blocks. In our experiments, we used the Tanh as the activation function, set the hidden dimension to be 300, and used the same architecture for both DSM and PINN. Empirically, we found out that this design choice would lead to faster convergence in terms of the training time.

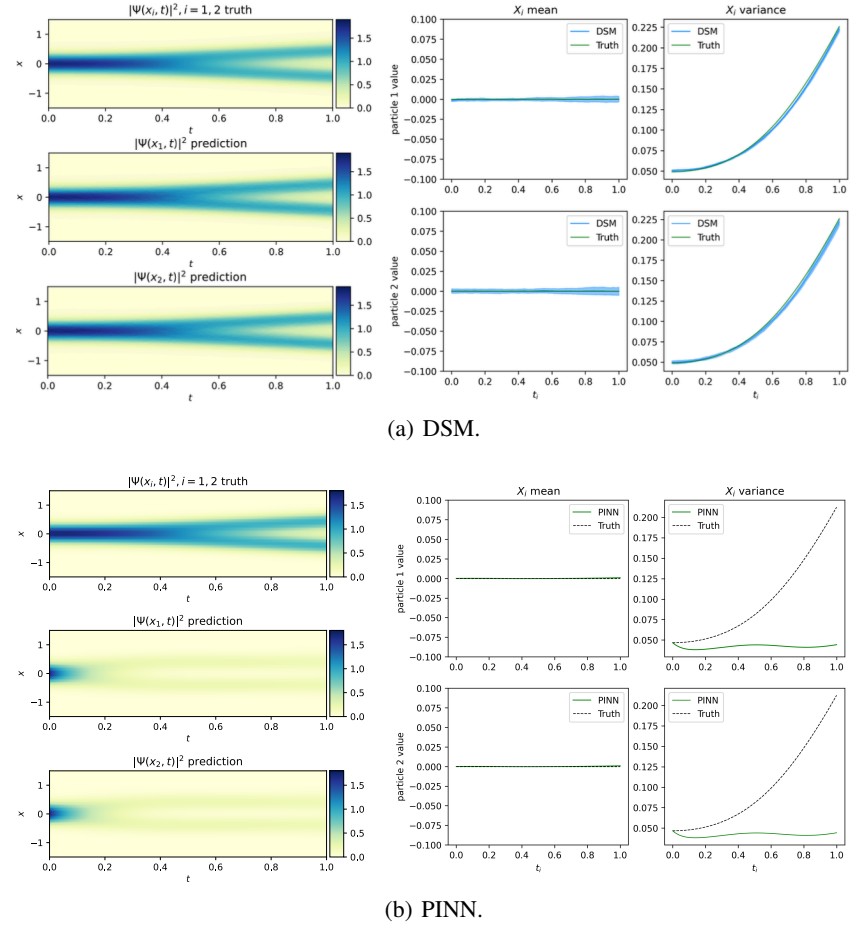

(a) DSM.

(b) PINN.

Figure 10: DSM and PINN results for two interacting particles.

### D.5.1 SCALING EXPERIMENTS

This section investigates scaling capabilities of our DSM approach in the case of interacting bosons in the harmonic oscillator. We compare performance of our algorithm with a numerical solver based on the Crank–Nicolson method method (we modified qmsolve library to work for $d > 2$). Table 7 shows training time, time per epoch and memory usage for our method. Table 6 reports time and memory usage of the Crank–Nicolson method solver.

**Memory**: In particular, DSM memory usage and time per epoch grow linearly in $d$ (according to our theory and evident in our numerical results) in contrast to the Crank-Nikolson solver, whose memory usage grows exponentially since discretization matrices are of size $N^d \times N^d$. We cannot run the Crank-Nikolson method for $d > 4$ with the amount of memory available in our computer system. The results show that our method is far more memory efficient for larger $d$.

**Compute time**: While our DSM total compute times (including training time) are longer than Crank-Nikolson compute times for small $d$, the trend as $d$ increases suggests the computational efficiency of our DSM method scales much better with d than the Crank-Nikolson method.

Figure 11 shows generated density functions. Note that while we don't have the baseline for $d = 5$, we believe DSM predictions are still reasonable. We assume that the simulation results are reliable for $N = 60$ in $d = 4$.

Table 6: Time (in seconds) to get a solution and memory usage (in Gb) of the Crank-Nicolson method for different problem dimensions.

|  | $d = 2$ | $d = 3$ | $d = 4$ |
| --- | --- | --- | --- |
| Time | 0.75 | 35.61 | 2363 |
| Memory usage | 7.4 | 10.6 | 214 |

Table 7: Training time (in minutes), time per epoch (in seconds/epoch) and memory usage (in Gb) of our method for different problem dimensions.

|  | $d = 2$ | $d = 3$ | $d = 4$ | $d = 5$ |
| --- | --- | --- | --- | --- |
| Training time | 29.5 | 60.3 | 97.5 | 154 |
| Time per epoch | 0.52 | 1.09 | 1.16 | 1.24 |
| Memory usage | 17.0 | 22.5 | 28.0 | 33.5 |

As for the DSM implementation details, we fix hyperparameters and only change $d$: for example the neural networks size is 500, batch size is 100. We train our method until the average training loss becomes lower than a particular threshold (0.007). These numbers are reported for a GPU A40. The Crank-Nikolson method is run on CPU.

## E STOCHASTIC MECHANICS

First, we will show how the equations of stochastic mechanics are derived from the Schrödinger one. For full derivation and proof of equivalence, see Nelson (1966).

### E.1 STOCHASTIC MECHANICS EQUATIONS

Let's consider the polar decomposition once again $\psi = \sqrt{\rho}e^{iS}$. Observe for $\partial \in \{\partial_t, \partial_{x_i}\}$ that:

$$\partial\psi = (\partial\sqrt{\rho})e^{iS} + (i\partial S)\psi = \frac{\partial\rho}{2\sqrt{\rho}}e^{iS} + (i\partial S)\psi = \frac{1}{2}\frac{\partial\rho}{\rho}\sqrt{\rho}e^{iS} + (i\partial S)\psi = \left(\frac{1}{2}\partial\log\rho + i\partial S\right)\psi,$$

$$\partial^2\psi = \partial\left(\left(\frac{1}{2}\partial\log\rho + i\partial S\right)\psi\right) = \left(\frac{1}{2}\partial^2\log\rho + i\partial^2 S + \left(\frac{1}{2}\partial\log\rho + i\partial S\right)^2\right)\psi$$

Substituting it into the Schrödinger equation:

$$i\hbar\left(\frac{1}{2}\partial_t\log\rho + i\partial_t S\right)\psi = -\frac{\hbar^2}{2m}\left(\frac{1}{2}\Delta\log\rho + i\Delta S + \left\|\frac{1}{2}\nabla\log\rho + i\nabla S\right\|^2\right)\psi + V\psi \tag{18}$$

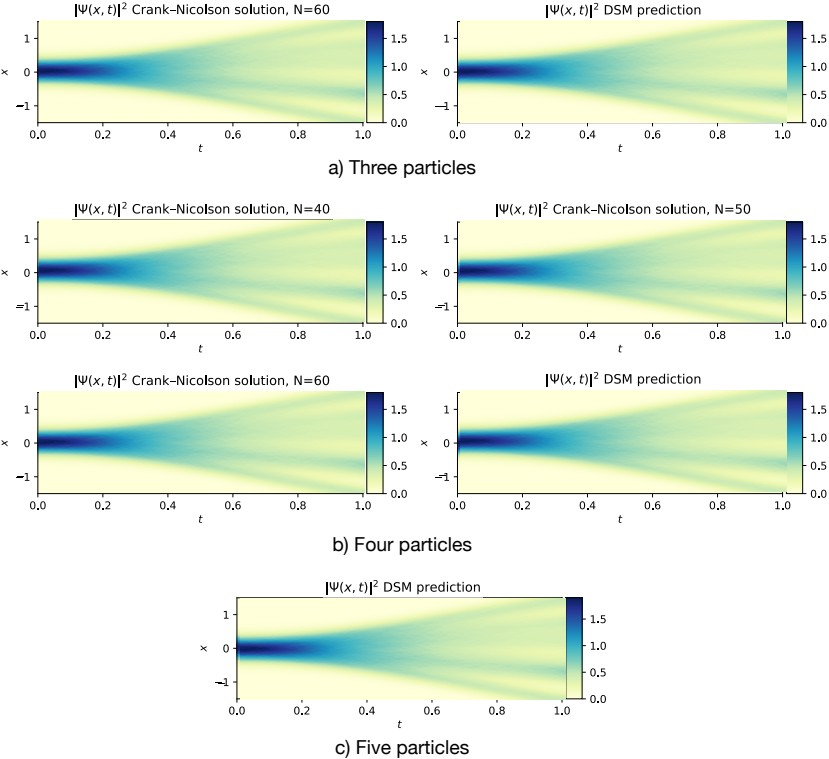

Figure 11: Examples of the obtained density for different number of interacting particles $d$. For five particles, our computer system does not allow to run the Crank-Nicolson solver.

Dividing by $\psi$[7] and separating real and imaginary parts, we obtain:

$$-\hbar\partial_t S = -\frac{\hbar^2}{2m}\left(\frac{1}{2}\Delta\log\rho + \frac{1}{4}\|\log\rho\|^2 - \|\nabla S\|^2\right) + V, \tag{19}$$

$$\frac{\hbar}{2}\partial_t\log\rho = -\frac{\hbar^2}{2m}\left(\Delta S + \langle\log\rho, \nabla S\rangle\right). \tag{20}$$

Noting that $\Delta = \langle\nabla, \nabla\cdot\rangle$ and substituting $v = \frac{\hbar}{m}\nabla S, u = \frac{\hbar}{2m}\log\rho$ to simplify we obtain:

$$m\frac{\hbar}{m}\partial_t S = \frac{\hbar}{2m}\langle\nabla, u\rangle + \frac{1}{2}\|u\|^2 - \frac{1}{2}\|v\|^2 - V, \tag{21}$$

$$\frac{\hbar}{2m}\partial_t\log\rho = -\frac{\hbar}{2m}\langle\nabla, v\rangle - \langle u, v\rangle. \tag{22}$$

Finally, by taking $\nabla$ from both parts, noting that $\left[\nabla, \partial_t\right] = 0$ for scalar functions and again substituting $u, v$, we arrive at:

$$\partial_t v = -\frac{1}{m}\nabla V + \langle u, \nabla\rangle u - \langle v, \nabla\rangle v + \frac{\hbar}{2m}\nabla\langle\nabla, u\rangle \tag{23}$$

$$\partial_t u = -\nabla\langle v, u\rangle - \frac{\hbar}{2m}\nabla\langle\nabla, v\rangle. \tag{24}$$

---

[7]Here, we assume $\psi \neq 0$. Even though it may seem a restriction, in our method, we will solve equations only on $X(t)$ which satisfy $\mathbb{P}\left(\psi(X(t), t) = 0\right) = 0$, thus, we are allowed to assume this safely without loss of generality. The same cannot be said if we considered the PINN over a grid to solve our equations.

To get the initial conditions on the velocities of the process $v_0 = v(x, 0)$ and $u_0 = u(x, 0)$, we can refer to the equations that we used in the derivation

$$v(x, t) = \frac{\hbar}{m} \nabla S(x, t), \tag{25}$$

$$u(x, t) = \frac{\hbar}{2m} \nabla \log \rho(x, t) \tag{26}$$

Substituting $t = 0$ we can get our initial conditions on $v_0(x) = \frac{\hbar}{m} \nabla S(x, 0)$, $u_0(x) = \nu \nabla \log \rho_0(x)$ where $\rho_0(x) = \rho(x, 0)$.

For more detailed derivation and proof of equivalence of those two equations to the Schrödinger one, see Nelson (1966; 2005); Guerra (1995). Moreover, this equivalence holds for manifolds $\mathcal{M}$ with trivial second cohomology group as noted in Alvarez (1986); Prieto & Vitolo (2014); Wallstrom (1989).

### E.2 Novel Equations of Stochastic Mechanics

We note that our equations differ from Nelson (1966); Guerra (1995). In Nelson (1966), we see

$$\partial_t v = -\frac{1}{m} \nabla V + \langle u, \nabla \rangle u - \langle v, \nabla \rangle v + \frac{\hbar}{2m} \Delta u, \tag{27}$$

$$\partial_t u = -\nabla \langle v, u \rangle - \frac{\hbar}{2m} \nabla \langle \nabla, v \rangle; \tag{28}$$

and in Guerra (1995), we see

$$\partial_t v = -\frac{1}{m} \nabla V + \langle u, \nabla \rangle u - \langle v, \nabla \rangle v + \frac{\hbar}{2m} \Delta u, \tag{29}$$

$$\partial_t u = -\nabla \langle v, u \rangle - \frac{\hbar}{2m} \Delta v. \tag{30}$$

The discrepancy seems to occur because the work by Nelson (2005) covers the case of the multi-valued $S$ and thus does not assume that $[\Delta, \nabla] = 0$ to transform $\nabla \langle \nabla, \nabla S \rangle$ into $\Delta(\nabla S)$ to make the equations work for the case of a non-trivial cohomology group of $\mathcal{M}$. So, Nelson (2005) does not transform $\nabla \langle \nabla, \nabla S \rangle$ into $\Delta(\nabla S)$, but Guerra (1995) uses $\Delta(\nabla S)$. Computing $\Delta$ with autograd tools requires $\mathcal{O}(d^2)$ operations as it requires computing the full Hessian $\nabla^2$. Instead, we treat $\log \rho$ as it can be multi-valued not because we want more generality but because we want to have $\mathcal{O}(d)$ computational time in the dimension as computing $\nabla \langle \nabla, \cdot \rangle$ is much easier with autograd tools. Generally, we cannot swap $\Delta$ with $\nabla \langle \nabla, \cdot \rangle$ unless the solutions of the equation can be represented as full gradients of some function, which is the case for stochastic mechanical equations but not for the Shrödinger one. For some reason, it is not discussed in Nelson (2005; 1966); Guerra (1995), but it is the reason why the equations in Nelson (1966); Guerra (1995) are different.

We derive equations different from both works and provide insights into why there are four different equivalent sets of equations (by changing $\Delta$ with $\nabla \langle \nabla, \cdot \rangle$ in both equations independently). From a numerical perspective, it is more beneficial to avoid Laplacian calculations. However, we notice that inference using equations from Nelson (1966) converges faster by iterations to the true $u, v$ compared to our version. It comes at the cost of a severe slowdown in each iteration for $d \gg 1$, which diminishes the benefit since the overall training time to get comparable results decreases significantly.

### E.3 Diffusion Processes of Stochastic Mechanics

Let's consider arbitrary Ito diffusion process:

$$dX(t) = b(X(t), t)dt + \sigma(X(t), t)d\vec{W}, \tag{31}$$

$$X(0) \sim \rho_0, \tag{32}$$

where $W(t) \in \mathbb{R}^d$ is the standard Wiener process, $b : \mathbb{R}^d \times [0, T] \to \mathbb{R}^d$ is the drift function, and $\sigma : \mathbb{R}^d \times [0, T] \to \mathbb{R}^{d \times d}$ is a symmetric positive definite matrix-valued function called a diffusion coefficient. Essentially, $X(t)$ samples from $\rho_X = \text{Law}(X(t))$ for each $t \in [0, T]$. Thus, we may wonder how to define $b$ and $\sigma$ to ensure $\rho_X = |\psi|^2$.

There is the forward Kolmogorov equation for the density $\rho_X$ associated with this diffusion process:

$$\partial_t \rho_X = \langle \nabla, b\rho_X \rangle + \frac{1}{2}\text{Tr}\big(\nabla^2 \cdot (\sigma\sigma^T \rho_X)\big). \tag{33}$$

Moreover, the diffusion process is time reversible. This leads to the backward Kolmogorov equation:

$$\partial_t \rho_X = \langle \nabla, b^*\rho_X \rangle - \frac{1}{2}\text{Tr}\big(\nabla^2 \cdot (\sigma\sigma^T \rho_X)\big), \tag{34}$$

where $b_i^* = b_i - \rho_X^{-1}\langle \nabla, \sigma\sigma^T e_i \rho_X \rangle$ with $e_{ij} = \delta_{ij}$ for $j \in \{1, \ldots, d\}$. Summing up those two equations, we obtain the following:

$$\partial_t \rho_X = \langle \nabla, v\rho_X \rangle, \tag{35}$$

where $v = \dfrac{b + b^*}{2}$ is so called probability current. This is the continuity equation for the Ito diffusion process from Equation (31). We refer to Anderson (1982) for details. We note that the same Equation (35) can be obtained with arbitrary non-singular $\sigma(x, t)$ as long as $v = v(x, t)$ remains fixed.

**Proposition E.1.** *Consider arbitrary $\nu > 0$, denote $\rho = |\psi|^2$ and consider decomposition $\psi = \sqrt{\rho}e^{iS}$. Then the following process $X(t)$:*

$$\mathrm{d}X(t) = \big(\nabla S(X(t), t) + \frac{\nu\hbar}{2m}\nabla \log \rho(X(t), t)\big)\mathrm{d}t + \sqrt{\frac{\nu\hbar}{m}}\mathrm{d}\vec{W}, \tag{36}$$

$$X(0) \sim |\psi_0|^2, \tag{37}$$

*satisfies* $\text{Law}(X(t)) = |\psi|^2$ *for any* $t > 0$.

*Proof.* We want to show that by choosing appropriately $b, b_*$, we can ensure that $\rho_X = |\psi|^2$. Let's consider the Schrödinger equation once again:

$$i\hbar\partial_t \psi = (-\frac{\hbar^2}{2m}\Delta + V)\psi, \tag{38}$$

$$\psi(\cdot, 0) = \psi_0 \tag{39}$$

where $\Delta = \text{Tr}(\nabla^2) = \sum_{i=1}^d \frac{\partial^2}{\partial x_i^2}$ is the Laplace operator. The second cohomology is trivial in this case. So, we can assume that $\psi = \sqrt{\rho}e^{iS}$ with $S(x, t)$ is a single-valued function.

By defining the drift $v = \dfrac{\hbar}{m}\nabla S$, we can derive quantum mechanics continuity equation on density $\rho$:

$$\partial_t \rho = \langle \nabla, v\rho \rangle, \tag{40}$$

$$\rho(\cdot, 0) = |\psi_0|^2. \tag{41}$$

This immediately tells us what should be initial distribution $\rho_0$ and $\frac{b+b^*}{2}$ for the Ito diffusion process (31).

For now, the only missing parts for obtaining the diffusion process from the quantum mechanics continuity equation are to identify the term $\frac{b-b^*}{2}$ and the diffusion coefficient $\sigma$. Both of them should be related as $(b - b^*)_i = \rho^{-1}\langle \nabla, \sigma\sigma^T e_i \rho \rangle$. Thus, we can pick $\sigma \propto I_d$ to simplify the equations. Nevertheless, our results can be extended to any non-trivial diffusion coefficient. Therefore, by defining $u(x, t) = \dfrac{\hbar}{2m}\nabla \log \rho(x, t)$ and using arbitrary $\nu > 0$ we derive

$$\partial_t \rho = \langle \nabla, (v + \nu u)\rho \rangle + \frac{\nu\hbar}{2m}\Delta\rho. \tag{42}$$

Thus, we can sample from $\rho_X(x, t) \equiv \rho(x, t)$ using the diffusion process with $b(x, t) = v(x, t) + \nu u(x, t)$ and $\sigma(x, t) \equiv \frac{\nu\hbar}{m}I_d$:

$$\mathrm{d}X(t) = (v(X(t), t) + \nu u(X(t), t))\mathrm{d}t + \sqrt{\frac{\nu\hbar}{m}}\mathrm{d}\vec{W}, \tag{43}$$

$$X(0) \sim |\psi_0|^2. \tag{44}$$

$\square$

To obtain numerical samples from the diffusion, one can use any numerical integrator, for example, the Euler-Maryama integrator (Kloeden & Platen, 1992):

$$X_{i+1} = X_i + (v(X_i, t_i) + \nu u(X_i, t_i))\epsilon + \sqrt{\frac{\nu \hbar}{m}} \epsilon \mathcal{N}(0, I_d), \tag{45}$$

$$X_0 \sim |\psi_0|^2, \tag{46}$$

where $\epsilon > 0$ is a step size, $0 \le i < \frac{T}{\epsilon}$. We consider this type of integrator in our work. However, integrators of higher order, e.g., Runge-Kutta family of integrators (Kloeden & Platen, 1992), can achieve the same integration error with larger $\epsilon > 0$; this approach is out of the scope of our work.

### E.4 INTERPOLATION BETWEEN BOHMIAN AND NELSONIAN PICTURES

We also differ from Nelson (1966) since we define $u$ without $\nu$. We bring it into the picture separately as a multiplicative factor:

$$dX(t) = (v(X(t), t) + \nu u(X(t), t))dt + \sqrt{\frac{\nu \hbar}{m}} d\overrightarrow{W}, \tag{47}$$

$$X(0) \sim |\psi_0|^2 \tag{48}$$

This trick allows us to recover Nelson's diffusion when $\nu = 1$:

$$dX(t) = (v(X(t), t) + u(X(t), t))dt + \sqrt{\frac{\hbar}{m}} d\overrightarrow{W}, \tag{49}$$

$$X(0) \sim |\psi_0|^2 \tag{50}$$

For cases of $|\psi_0|^2 > 0$ everywhere, e.g., if the initial conditions are gaussian but not singular like $\delta_{x_0}$, we can actually set $\nu = 0$ to obtain deterministic flow:

$$dX(t) = v(X(t), t)dt, \tag{51}$$

$$X(0) \sim |\psi_0|^2 \tag{52}$$

This is the guiding equation in Bohr's pilot-wave theory (Bohm, 1952). The major drawback of using Bohr's interpretation is that $\rho_X$ may not equal $\rho = |\psi|^2$, a phenomenon known as quantum non-equilibrium (Colin & Struyve, 2010). Though, under certain mild conditions (Boffi & Vanden-Eijnden, 2023) (one of which is $|\psi_0|^2 > 0$ everywhere) time marginals of such deterministic process $X(t)$ will satisfy $\text{Law}(X(t)) = \rho$ for each $t \in [0, T]$. As with the SDE case, it is unlikely that those trajectories are "true" trajectories. It only matters that their time marginals coincide with true quantum mechanical densities.

## F ON STRONG CONVERGENCE

Let's consider a standard Wiener processes $\overrightarrow{W}^X, \overrightarrow{W}^Y$ in $\mathbb{R}^d$ and define $\overrightarrow{\mathcal{F}}_t$ as a filtration generated by $\left\{ \left( \overrightarrow{W}^X(t'), \overrightarrow{W}^Y(t) \right) : t' \le t \right\}$. Let $\overleftarrow{\mathcal{F}}_t$ be a filtration generated by all events $\left\{ \left( \overrightarrow{W}^X(t'), \overrightarrow{W}^Y(t) \right) : t' \ge t \right\}$.

Assume that $u, v, \widetilde{u}, \widetilde{v} \in C^{2,1}(\mathbb{R}^d \times [0, T]; \mathbb{R}^d) \cap C_b^{1,0}(\mathbb{R}^d \times [0, T]; \mathbb{R}^d)$, where $C_b^{p,k}$ is a class of continuously differentiable functions with uniformly bounded $p$-th derivative in a coordinate $x$ and $k$-th continuously differentiable in $t$, $C^{p,k}$ analogously but without requiring bounded derivative. For $f : \mathbb{R}^d \times [0, T] \to \mathbb{R}^k$ define $\|f\|_\infty = \operatorname{ess\,sup}_{t \in [0,T], x \in \mathbb{R}^d} \|f(x, t)\|$ and $\|\nabla f\|_\infty = \operatorname{ess\,sup}_{t \in [0,T], x \in \mathbb{R}^d} \|\nabla f(x, t)\|_{op}$ where $\|\cdot\|_{op}$ denotes operator norm.

$$dX(t) = \left( \widetilde{v}(X(t), t) + \widetilde{u}(X(t), t) \right)dt + \sqrt{\frac{\hbar}{m}} d\overrightarrow{W}^X(t), \tag{53}$$

$$dY(t) = \left( v(Y(t), t) + u(Y(t), t) \right)dt + \sqrt{\frac{\hbar}{m}} d\overrightarrow{W}^Y(t), \tag{54}$$

$$X(0) \sim |\psi_0|^2, \tag{55}$$

$$Y(0) = X(0), \tag{56}$$

where $u, v$ are true solutions to equations (27). We have that $p_Y(\cdot, t) = |\psi(\cdot, t)|^2 \ \forall t$ where $p_Y$ is density of the process $Y(t)$. We have not specified yet quadratic covariation of those two processes $\frac{\mathrm{d}[\overrightarrow{W^X}, \overrightarrow{W^Y}]_t}{\mathrm{d}t} = \lim_{\mathrm{d}t \to 0_+} \mathbb{E}\left(\frac{\left(\overrightarrow{W^X}(t+\mathrm{d}t) - \overrightarrow{W^X}(t)\right)\left(\overrightarrow{W^Y}(t+\mathrm{d}t) - \overrightarrow{W^Y}(t)\right)}{\mathrm{d}t}\Big|\overrightarrow{\mathcal{F}_t}\right)$. We will though specify it as $\mathrm{d}[\overrightarrow{W^X}, \overrightarrow{W^Y}]_t = I_d \mathrm{d}t$ as we will see it will allow to cancel some terms appearing in the equations. As for now we will derive all results in most general setting.

Let's define our loss functions:

$$L_1(\widetilde{v}, \widetilde{u}) = \int_0^T \mathbb{E}^X \|\partial_t \widetilde{u}(X(t), t) - \mathcal{D}_u[\widetilde{v}, \widetilde{u}, x, t]\|^2 \mathrm{d}t, \tag{57}$$

$$L_2(\widetilde{v}, \widetilde{u}) = \int_0^T \mathbb{E}^X \|\partial_t \widetilde{v}(X(t), t) - \mathcal{D}_v[\widetilde{v}, \widetilde{u}, X(t), t]\|^2 \mathrm{d}t, \tag{58}$$

$$L_3(\widetilde{u}, \widetilde{v}) = \mathbb{E}^X \|\widetilde{u}(X(0), 0) - u(X(0), 0)\|^2 \tag{59}$$

$$L_4(\widetilde{u}, \widetilde{v}) = \mathbb{E}^X \|\widetilde{v}(X(0), 0) - v(X(0), 0)\|^2 \tag{60}$$

Our goal is to show that for some constants $w_i > 0$, there is natural bound $\sup_{0 \le t \le T} \mathbb{E}\|X(t) - Y(t)\|^2 \le \sum w_i L_i(\widetilde{v}, \widetilde{u})$.

### F.1 STOCHASTIC PROCESSES

Consider a general Itô SDE defined using a drift process $F(t)$ and a covariance process $G(t)$, both predictable with respect to forward and backward flirtations $\overleftarrow{\mathcal{F}_t}$ and $\overrightarrow{\mathcal{F}_t}$:

$$\mathrm{d}Z(t) = F(t)\mathrm{d}t + G(t)\mathrm{d}\overrightarrow{W}, \tag{61}$$
$$Z(0) \sim \rho_0.$$

Moreover, assume $[Z(t), Z(t)]_t = \mathbb{E}\int_0^t G^T G(t)\mathrm{d}t < \infty$, $\mathbb{E}\int_0^t \|F(t)\|^2 \mathrm{d}t < \infty$. We denote by $\mathbb{P}_t^Z = \mathbb{P}(Z(t) \in \cdot)$ a law of the process $Z(t)$. Let's define a (extended) forward generate of the process as the linear operator satisfying

$$\overrightarrow{M^f}(t) = f(Z(t), t) - f(Z(0), 0) - \int_0^t \overrightarrow{\mathcal{L}^X} f(Z(t), t) \text{ is } \overrightarrow{\mathcal{F}_t}\text{-martingale}. \tag{62}$$

Such an operator is uniquely defined and is called a forward generator associated with the process $Z_t$. Similarly, we define a (extended) backward generator $\overleftarrow{\mathcal{L}^X}$ as linear operator satisfying:

$$\overleftarrow{M^f}(t) = f(Z(t), t) - f(Z(0), 0) - \int_0^t \overleftarrow{\mathcal{L}^X} f(Z(t), t) \text{ is } \overleftarrow{\mathcal{F}_t}\text{-martingale} \tag{63}$$

For more information on properties of generators we refer to (Baldi & Baldi, 2017).

**Lemma F.1.** *(Itô Lemma, Baldi & Baldi (2017))*

$$\overrightarrow{\mathcal{L}^Z} f(x, t) = \partial_t f(x, t) + \langle \nabla f(x, t), F(t) \rangle + \frac{\hbar}{2m} \mathrm{Tr}\left(G^T(t) \nabla^2 f(x, s) G(t)\right). \tag{64}$$

**Lemma F.2.** *Let $p_Z(x, t) = \frac{\mathrm{d}\mathbb{P}_t^Z}{\mathrm{d}x}$ be the density of the process with respect to standard Lebesgue measure on $\mathbb{R}^d$. Then*

$$\overleftarrow{\mathcal{L}^Z} f(x, t) = \partial_t f(x, t) + \langle \nabla f(x, t), F(t) - \frac{\hbar}{m} \nabla \log p_Z(x, t) \rangle - \frac{1}{2} \mathrm{Tr}\left(G^T(t) \nabla^2 f(x, s) G(t)\right). \tag{65}$$

*Proof.* We have the following operator identities:

$$\overleftarrow{\mathcal{L}^Z} = \left(\overrightarrow{\mathcal{L}^Z}\right)^* = p_Z^{-1}\left(\overrightarrow{\mathcal{L}^X}\right)^\dagger p_Z$$

where $\mathcal{A}^*$ is adjoint operator in $L_2(\mathbb{R}^d \times [0, T], \mathbb{P}^Z \otimes \mathrm{d}t)$ and $\mathcal{A}^\dagger$ is adjoint in $L_2(\mathbb{R}^d \times [0, T], \mathrm{d}x \otimes \mathrm{d}t)$. Using Itô lemma F.1 and grouping all terms yields the statement. $\square$

**Lemma F.3.** *The following identity holds for any process $Z(t)$:*

$$\overrightarrow{\mathcal{L}^Z}\overleftarrow{\mathcal{L}^Z}x = \overleftarrow{\mathcal{L}^Z}\overrightarrow{\mathcal{L}^Z}x. \tag{66}$$

*Proof.* One needs to recognize that (35) is the difference between two types of generators, we automatically have the following identity that holds for any process $Z$. $\square$

**Lemma F.4.** *(Nelson Lemma, Nelson (2020))*

$$\mathbb{E}^Z\Big(f(Z(t),t)g(Z(t),t) - f(Z(0),t)g(Z(0),t)\Big) \tag{67}$$

$$= \mathbb{E}^Z\int_0^t\Big(\overrightarrow{\mathcal{L}^Z}f(Z(s),t)g(Z(s),t) + f(Z(s),t)\overleftarrow{\mathcal{L}^Z}g(Z(s),s)\Big)\mathrm{d}s \tag{68}$$

**Lemma F.5.** *It holds that:*

$$\mathbb{E}^Z\Big(\|Z(t)\|^2 - \|Z(0)\|^2\Big) \tag{69}$$

$$= \int_0^t\mathbb{E}^Z\Big(2\langle\overleftarrow{\mathcal{L}^Z}Z(0), Z(s)\rangle + 2\int_0^s\langle\overleftarrow{\mathcal{L}^Z}\overrightarrow{\mathcal{L}^Z}Z(z), Z(s)\rangle\mathrm{d}z\Big)\mathrm{d}s + \big[Z(t), Z(t)\big]_t \tag{70}$$

*Proof.* By using Itô Lemma F.1 for $f(x) = \|x\|^2$ and noting that $\overrightarrow{\mathcal{L}^Z}Z(t) = F(t)$ we immediately obtain:

$$\mathbb{E}^Z(\|Z(t)\|^2 - \|Z(0)\|^2) = \int_0^t\mathbb{E}\Big(2\langle\overrightarrow{\mathcal{L}^Z}Z(s), Z(s)\rangle + \mathrm{Tr}\big(G^TG(t)\big)\Big)\mathrm{d}s$$

Let's deal with the term $\int_0^t\langle\overrightarrow{\mathcal{L}^Z}Z(s), Z(s)\rangle\mathrm{d}s$. We have the following observation: $\overrightarrow{M^F}(z) = \overleftarrow{\mathcal{L}^Z}Z(s) - \overleftarrow{\mathcal{L}^Z}Z(0) - \int_0^s\overleftarrow{\mathcal{L}^Z}\overrightarrow{\mathcal{L}^Z}Z(z)\mathrm{d}z$ is $\overleftarrow{\mathcal{F}}_s$-martingale, thus

$$\int_0^t\langle\overrightarrow{\mathcal{L}^Z}Z(s), Z(s)\rangle\mathrm{d}s = \int_0^t\langle\overleftarrow{\mathcal{L}^Z}Z(0) + \int_0^s\big(\overleftarrow{\mathcal{L}^Z}\overrightarrow{\mathcal{L}^Z}Z(z) + \overrightarrow{M^F}(z)\big)\mathrm{d}z, Z(s)\rangle\mathrm{d}s,$$

The process $\overrightarrow{A}(s', s) = \int_{s'}^s\langle\overleftarrow{M^F}(z), Z(s)\rangle\mathrm{d}z$ is again $\overleftarrow{\mathcal{F}}_{s'}$-martingale for $s' \leq s$, which implies that $\mathbb{E}^Z\overrightarrow{A}(0, s) = 0$. Noting that $\mathbb{E}^Z\int_0^t\mathrm{Tr}\big(G^T(t)G(t)\big)\mathrm{d}t = \big[Z(t), Z(t)\big]_t$ yields the lemma. $\square$

## F.2 ADJOINT PROCESSES

Consider process $X'(t)$ defined through time-reversed SDE:

$$\mathrm{d}X'(t) = (\widetilde{v}(X'(t),t) + \widetilde{u}(X'(t),t))\mathrm{d}t + \sqrt{\frac{\hbar}{2m}}\overleftarrow{\mathrm{d}W^X}(t). \tag{71}$$

We call such process as adjoint to the process $X$. Lemma F.3 can be generalized to the pair of adjoint processes $(X, X')$ in the following way and will be instrumental in proving our results.

**Lemma F.6.** *For any pair of processes $X(t), X'(t)$ such that the forward drift of $X$ is of form $\widetilde{v} + \widetilde{u}$ and backward drift of $X'$ is $\widetilde{v} - \widetilde{u}$:*

$$\overrightarrow{\mathcal{L}^X}\overleftarrow{\mathcal{L}^{X'}}x - \overleftarrow{\mathcal{L}^{X'}}\overrightarrow{\mathcal{L}^X}x = \overleftarrow{\mathcal{L}^{X'}}\overleftarrow{\mathcal{L}^{X'}}x - \overrightarrow{\mathcal{L}^X}\overrightarrow{\mathcal{L}^X}x. \tag{72}$$

*with both sides being equal to $0$ if and only if $X'$ is time reversal of $X$.*

*Proof.* Manual substitution of explicit forms of generators and drifts yields equation (8b) for both cases. This equation is zero only if $\widetilde{u} = \frac{\hbar}{2m}\nabla\log p_X$ $\square$

**Lemma F.7.** *The following bound holds:*

$$\Big\|\big(\overrightarrow{\mathcal{L}^X} + \overleftarrow{\mathcal{L}^X}\big)(\widetilde{u} - \frac{\hbar}{2m}\nabla\log p_X)\Big\| \leq \Big\|\overrightarrow{\mathcal{L}^X}\overleftarrow{\mathcal{L}^{X'}}x - \overleftarrow{\mathcal{L}^{X'}}\overrightarrow{\mathcal{L}^X}x\Big\| + 2\|\nabla\widetilde{v}\|_\infty\|\widetilde{u} - \frac{\hbar}{2m}\nabla\log p_X\|. \tag{73}$$

*Proof.* First, using Lemma F.6 we obtain:

$$\overrightarrow{\mathcal{L}^X}\overleftarrow{\mathcal{L}^X}x - \overleftarrow{\mathcal{L}^X}\overrightarrow{\mathcal{L}^X}x = 0 \tag{74}$$

$$\iff \overrightarrow{\mathcal{L}^X}\big(\widetilde{v} + \widetilde{u} - \frac{\hbar}{m}\nabla\log p_X\big) - \overleftarrow{\mathcal{L}^X}\big(\widetilde{v} + \widetilde{u}\big) = 0 \tag{75}$$

$$\iff \overrightarrow{\mathcal{L}^X}\big((\widetilde{v} - \widetilde{u}) + (2\widetilde{u} - \frac{\hbar}{m}\nabla\log p_X)\big) - \overleftarrow{\mathcal{L}^X}\big(\widetilde{v} + \widetilde{u}\big) = 0 \tag{76}$$

$$\iff \overrightarrow{\mathcal{L}^X}\big((\widetilde{v} - \widetilde{u}) + (2\widetilde{u} - \frac{\hbar}{m}\nabla\log p_X)\big) - \overleftarrow{\mathcal{L}^{X'}}\big(\widetilde{v} + \widetilde{u}\big) + \big(\overleftarrow{\mathcal{L}^{X'}}\big(\widetilde{v} + \widetilde{u}\big) - \overleftarrow{\mathcal{L}^X}\big(\widetilde{v} + \widetilde{u}\big)\big) = 0 \tag{77}$$

$$\iff \overrightarrow{\mathcal{L}^X}\big(2\widetilde{u} - \frac{\hbar}{m}\nabla\log p_X\big) + \overrightarrow{\mathcal{L}^X}\big(\widetilde{v} - \widetilde{u}\big) - \overleftarrow{\mathcal{L}^{X'}}\big(\widetilde{v} + \widetilde{u}\big) + \big(\overleftarrow{\mathcal{L}^{X'}}\big(\widetilde{v} + \widetilde{u}\big) - \overleftarrow{\mathcal{L}^X}\big(\widetilde{v} + \widetilde{u}\big)\big) = 0. \tag{78}$$

Then, we note that:

$$\overleftarrow{\mathcal{L}^{X'}}\big(\widetilde{v} + \widetilde{u}\big) - \overleftarrow{\mathcal{L}^X}\big(\widetilde{v} + \widetilde{u}\big) = \langle\frac{\hbar}{m}\nabla\log p_X - 2\widetilde{u}, \nabla(\widetilde{v} + \widetilde{u})\rangle. \tag{79}$$

This leads us to the following identity:

$$\overrightarrow{\mathcal{L}^X}\big(2\widetilde{u} - \frac{\hbar}{m}\nabla\log p_X\big) + \overrightarrow{\mathcal{L}^X}\big(\widetilde{v} - \widetilde{u}\big) - \overleftarrow{\mathcal{L}^{X'}}\big(\widetilde{v} + \widetilde{u}\big) + \langle\frac{\hbar}{m}\nabla\log p_X - 2\widetilde{u}, \nabla(\widetilde{v} + \widetilde{u})\rangle = 0$$

$$\iff \overrightarrow{\mathcal{L}^X}\big(2\widetilde{u} - \frac{\hbar}{m}\nabla\log p_X\big) + \overrightarrow{\mathcal{L}^X}\overleftarrow{\mathcal{L}^{X'}}x - \overleftarrow{\mathcal{L}^{X'}}\overrightarrow{\mathcal{L}^X}x + \langle\frac{\hbar}{m}\nabla\log p_X - 2\widetilde{u}, \nabla(\widetilde{v} + \widetilde{u})\rangle = 0.$$

Again by using Lemma F.6 to time-reversal $X'$ we obtain:

$$\overleftarrow{\mathcal{L}^X}\overleftarrow{\mathcal{L}^X}x - \overrightarrow{\mathcal{L}^X}\overrightarrow{\mathcal{L}^X}x = 0 \tag{80}$$

$$\iff \overleftarrow{\mathcal{L}^X}\big(\widetilde{v} + \widetilde{u} - \frac{\hbar}{m}\nabla\log p_X\big) - \overrightarrow{\mathcal{L}^X}\big(\widetilde{v} + \widetilde{u}\big) = 0 \tag{81}$$

$$\iff \overleftarrow{\mathcal{L}^X}\big((\widetilde{v} - \widetilde{u}) + (2\widetilde{u} - \frac{\hbar}{m}\nabla\log p_X)\big) - \overrightarrow{\mathcal{L}^X}\big(\widetilde{v} + \widetilde{u}\big) = 0 \tag{82}$$

$$\iff \overleftarrow{\mathcal{L}^{X'}}\big(\widetilde{v} - \widetilde{u}\big) + \overleftarrow{\mathcal{L}^X}\big(2\widetilde{u} - \frac{\hbar}{m}\nabla\log p_X\big) - \overrightarrow{\mathcal{L}^X}\big(\widetilde{v} + \widetilde{u}\big) + \big(\overleftarrow{\mathcal{L}^X}\big(\widetilde{v} - \widetilde{u}\big) - \overleftarrow{\mathcal{L}^{X'}}\big(\widetilde{v} - \widetilde{u}\big)\big) = 0 \tag{83}$$

$$\iff \overleftarrow{\mathcal{L}^X}\big(2\widetilde{u} - \frac{\hbar}{m}\nabla\log p_X\big) + \overleftarrow{\mathcal{L}^{X'}}\big(\widetilde{v} - \widetilde{u}\big) - \overrightarrow{\mathcal{L}^X}\big(\widetilde{v} + \widetilde{u}\big) - \langle\frac{\hbar}{m}\nabla\log p_X - 2\widetilde{u}, \nabla(\widetilde{v} - \widetilde{u})\rangle = 0 \tag{84}$$

$$\iff \overleftarrow{\mathcal{L}^X}\big(2\widetilde{u} - \frac{\hbar}{m}\nabla\log p_X\big) + \overleftarrow{\mathcal{L}^{X'}}\overleftarrow{\mathcal{L}^{X'}}x - \overrightarrow{\mathcal{L}^X}\overrightarrow{\mathcal{L}^X}x - \langle\frac{\hbar}{m}\nabla\log p_X - 2\widetilde{u}, \nabla(\widetilde{v} - \widetilde{u})\rangle = 0. \tag{85}$$

By using Lemma F.6 we thus derive:

$$\overleftarrow{\mathcal{L}^X}\big(2\widetilde{u} - \frac{\hbar}{m}\nabla\log p_X\big) + \overrightarrow{\mathcal{L}^X}\overleftarrow{\mathcal{L}^{X'}}x - \overleftarrow{\mathcal{L}^{X'}}\overrightarrow{\mathcal{L}^X}x - \langle\frac{\hbar}{m}\nabla\log p_X - 2\widetilde{u}, \nabla(\widetilde{v} - \widetilde{u})\rangle = 0. \tag{86}$$

Summing up both identities, therefore, yields:

$$\big(\overleftarrow{\mathcal{L}^X} + \overrightarrow{\mathcal{L}^X}\big)\big(\widetilde{u} - \frac{\hbar}{2m}\nabla\log p_X\big) + \overrightarrow{\mathcal{L}^X}\overleftarrow{\mathcal{L}^{X'}}x - \overleftarrow{\mathcal{L}^{X'}}\overrightarrow{\mathcal{L}^X}x + 2\langle\widetilde{u} - \frac{\hbar}{2m}\nabla\log p_X, \nabla\widetilde{v}\rangle = 0. \tag{87}$$

$$\square$$

**Theorem F.8.** *The following bound holds:*

$$\sup_{0\leq t\leq T}\mathbb{E}^X\big\|\widetilde{u}(X(t),t) - \frac{\hbar}{2m}\nabla\log p_X(X(t),t)\big\|^2 \leq e^{\big(\frac{1}{2} + 4\|\nabla\widetilde{v}\|_\infty\big)T}\big(L_3(\widetilde{v},\widetilde{u}) + L_2(\widetilde{v},\widetilde{u})\big). \tag{88}$$

*Proof.* We consider process $Z(t) = \widetilde{u}u(X(t), t) - \frac{\hbar}{2m}\nabla \log p_X(X(t), t)$. From Nelson's lemma F.4, we have the following identity:

$$\mathbb{E}^X \|\widetilde{u}(X(t), t) - \frac{\hbar}{2m}\nabla \log p_X(X(t), t)\|^2 - \mathbb{E}^X \|\widetilde{u}(X(0), 0) - \frac{\hbar}{2m}\nabla \log p_X(X(0), 0)\|^2 \quad (89)$$

$$= \mathbb{E}^X \int_0^t \langle u(X(s), s) - \frac{\hbar}{2m}\nabla \log p_X(X(s), s), \quad (90)$$

$$(\overrightarrow{\mathcal{L}^X} + \overleftarrow{\mathcal{L}^X})(u(X(s), s) - \frac{\hbar}{2m}\nabla \log p_X(X(s), s))\rangle \mathrm{d}s. \quad (91)$$

Note that $u \equiv \frac{\hbar}{2m}\nabla \log p_X(X(t), t)$. Thus, $\mathbb{E}^X \|\widetilde{u}(X(0), 0) - \frac{\hbar}{2m}\nabla \log p_X(X(0), 0)\|^2 = L_3(\widetilde{v}, \widetilde{u})$. Using inequality $\langle a, b\rangle \leq \frac{1}{2}(\|a\|^2 + \|b\|^2)$ we obtain:

$$\mathbb{E}^X \|u(X(t), t) - \frac{\hbar}{2m}\nabla \log p_X(X(t), t)\|^2 - L_3(\widetilde{v}, \widetilde{u}) \quad (92)$$

$$\leq \int_0^t \left(\frac{1}{2}\mathbb{E}^X \|u(X(s), s) - \frac{\hbar}{2m}\nabla \log p_X(X(s), s)\|^2 \quad (93)\right.$$

$$\left. + \frac{1}{2}\mathbb{E}^X \left\|(\overrightarrow{\mathcal{L}^X} + \overleftarrow{\mathcal{L}^X})(u(X(s), s) - \frac{\hbar}{2m}\nabla \log p_X(X(s), s))\right\|^2\right)\mathrm{d}s \quad (94)$$

Using Lemma F.7, we obtain:

$$\mathbb{E}^X \|u(X(t), t) - \frac{\hbar}{2m}\nabla \log p_X(X(t), t)\|^2 - L_3(\widetilde{v}, \widetilde{u}) \quad (95)$$

$$\leq \int_0^t \left(\frac{1}{2}\mathbb{E}^X \|u(X(s), s) - \frac{\hbar}{2m}\nabla \log p_X(X(s), s)\|^2 \quad (96)\right.$$

$$\left. + \left\|\overrightarrow{\mathcal{L}^X}\overleftarrow{\mathcal{L}^{X'}}x - \overleftarrow{\mathcal{L}^{X'}}\overrightarrow{\mathcal{L}^X}x\right\|^2 + 4\|\nabla\widetilde{v}\|_\infty^2\|\widetilde{u} - \frac{\hbar}{2m}\nabla \log p_X\|^2\right)\mathrm{d}s \quad (97)$$

Observe that $\int_0^t \mathbb{E}^X \left\|\overrightarrow{\mathcal{L}^X}\overleftarrow{\mathcal{L}^{X'}}x - \overleftarrow{\mathcal{L}^{X'}}\overrightarrow{\mathcal{L}^X}x\right\|^2 \mathrm{d}t \leq L_2(\widetilde{v}, \widetilde{u})$, in fact, at $t = T$ it is equality as this is the definition of the loss $L_2$. Thus, we have:

$$\mathbb{E}^X \|u(X(t), t) - \frac{\hbar}{2m}\nabla \log p_X(X(t), t)\|^2 \quad (98)$$

$$\leq L_3(\widetilde{v}, \widetilde{u}) + L_2(\widetilde{v}, \widetilde{u}) + \int_0^t \left(\frac{1}{2} + 4\|\nabla\widetilde{v}\|_\infty\right)\mathbb{E}^X \|u(X(s), s) - \frac{\hbar}{2m}\nabla \log p_X(X(s), s)\|^2 \mathrm{d}s. \quad (99)$$

Using integral Grönwall's inequality (Gronwall, 1919) yields the bound: $\mathbb{E}^X \|u(X(t), t) - \frac{\hbar}{2m}\nabla \log p_X(X(t), t)\|^2 \leq e^{\left(\frac{1}{2} + 4\|\nabla\widetilde{v}\|_\infty\right)t}\left(L_3(\widetilde{v}, \widetilde{u}) + L_2(\widetilde{v}, \widetilde{u})\right)$. $\square$

## F.3 NELSONIAN PROCESSES

Considering those two operators, we can rewrite the equations (27) alternatively as:

$$\frac{1}{2}\left(\overrightarrow{\mathcal{L}^Y}\overleftarrow{\mathcal{L}^Y}x + \overleftarrow{\mathcal{L}^Y}\overrightarrow{\mathcal{L}^Y}x\right) = -\frac{1}{m}\nabla V(x), \quad (100)$$

$$\frac{1}{2}\left(\overrightarrow{\mathcal{L}^Y}\overleftarrow{\mathcal{L}^Y}x - \overleftarrow{\mathcal{L}^Y}\overrightarrow{\mathcal{L}^Y}x\right) = 0. \quad (101)$$

This leads us to the identity:

$$\overrightarrow{\mathcal{L}^Y}\overleftarrow{\mathcal{L}^Y}x = -\frac{1}{m}\nabla V(x). \quad (102)$$

**Lemma F.9.** *We have the following bound:*

$$\int_0^t \mathbb{E}^X \left\|\overrightarrow{\mathcal{L}^{X'}}\overleftarrow{\mathcal{L}^X}X(t) + \frac{1}{m}\nabla V(X(t))\right\|^2 \mathrm{d}t \leq 2L_1(\widetilde{v}, \widetilde{u}) + 2L_2(\widetilde{v}, \widetilde{u}).$$

*Proof.* Consider rewriting losses as:

$$L_1(\widetilde{v}, \widetilde{u}) = \int_0^t \mathbb{E}_{t \sim U[0,T]} \mathbb{E}^X \left\| \frac{1}{2} \left( \overrightarrow{\mathcal{L}^X} \overleftarrow{\mathcal{L}^{X'}} X(t) + \overrightarrow{\mathcal{L}^X} \overleftarrow{\mathcal{L}^{X'}} X(t) \right) + \frac{1}{m} \nabla V(X(t)) \right\|^2 dt, \quad (103)$$

$$L_2(\widetilde{v}, \widetilde{u}) = \frac{1}{4} \int_0^t \mathbb{E}_{t \sim U[0,T]} \mathbb{E}^X \left\| \overrightarrow{\mathcal{L}^X} \overleftarrow{\mathcal{L}^{X'}} X(t) - \overrightarrow{\mathcal{L}^{X'}} \overleftarrow{\mathcal{L}^X} X(t) \right\|^2 dt. \quad (104)$$

Using the triangle inequality yields the statement. $\qquad \square$

**Lemma F.10.** *We have the following bound:*

$$\int_0^t \mathbb{E}^X \left\| \overleftarrow{\mathcal{L}^X} \overrightarrow{\mathcal{L}^X} X(t) + \frac{1}{m} \nabla V(X(t)) \right\|^2 dt$$

$$\leq 2T \left( \|\nabla \widetilde{u}\|_\infty + \|\nabla \widetilde{v}\|_\infty \right)^2 e^{\left( \frac{1}{2} + 4\|\nabla \widetilde{v}\|_\infty \right) T} \left( L_3(\widetilde{v}, \widetilde{u}) + L_2(\widetilde{v}, \widetilde{u}) \right) + 4L_1(\widetilde{v}, \widetilde{u}) + 4L_2(\widetilde{v}, \widetilde{u}).$$

*Proof.* From equation 79 we have:

$$\overleftarrow{\mathcal{L}^X} \overrightarrow{\mathcal{L}^X} X(t) = \overleftarrow{\mathcal{L}^{X'}} \overrightarrow{\mathcal{L}^X} X(t) + \langle \frac{\hbar}{m} \nabla \log p_X - 2\widetilde{u}, \nabla(\widetilde{v} + \widetilde{u}) \rangle. \quad (105)$$

Noting that $\langle \frac{\hbar}{m} \nabla \log p_X - 2\widetilde{u}, \nabla(\widetilde{v} + \widetilde{u}) \rangle \leq \left( \|\nabla \widetilde{u}\|_\infty + \|\nabla \widetilde{v}\|_\infty \right) \left\| \frac{\hbar}{m} \nabla \log p_X - 2\widetilde{u} \right\|$ and using triangle inequality we obtain the bound:

$$\int_0^t \mathbb{E}^X \left\| \overleftarrow{\mathcal{L}^X} \overrightarrow{\mathcal{L}^X} X(t) + \frac{1}{m} \nabla V(X(t)) \right\|^2 dt \quad (106)$$

$$\leq 2 \left( \|\widetilde{u}\|_\infty + \|\widetilde{v}\|_\infty \right)^2 \int_0^t \mathbb{E}^X \left\| u(X(t), t) - \frac{\hbar}{2m} \log p_X(X(t), t) \right\|^2 dt + 4L_1(\widetilde{v}, \widetilde{u}) + 4L_2(\widetilde{v}, \widetilde{u}). \quad (107)$$

Using Theorem F.8 concludes the proof. $\qquad \square$

**Lemma F.11.** *Denote $Z(t) = (X(t), Y(t))$ as compound process. For functions $h(x, y, t) = f(x, t) + g(y, t)$ we have the following identity:*

$$\overrightarrow{\mathcal{L}^Z} h = \overrightarrow{\mathcal{L}^X} f + \overrightarrow{\mathcal{L}^Y} g \quad (108)$$

*Proof.* A generator is a linear operator by very definition. Thus, it remains to prove only

$$\overrightarrow{\mathcal{L}^Z} f = \overrightarrow{\mathcal{L}^X} f \quad (109)$$

Since the definition of $\overrightarrow{\mathcal{F}_t}$ already contains all past events for both processes $X(t), Y(t)$, we see that this is a tautology. $\qquad \square$

As a direct application of this Lemma, we obtain the following Corollary (by applying it twice):

**Corollary F.12.** *We have the following identity:*

$$\overleftarrow{\mathcal{L}^Z} \overrightarrow{\mathcal{L}^Z} \left( X(t) - Y(t) \right) = \overleftarrow{\mathcal{L}^X} \overrightarrow{\mathcal{L}^X} X(t) - \overleftarrow{\mathcal{L}^Y} \overrightarrow{\mathcal{L}^Y} Y(t).$$

**Theorem F.13.** *(Strong Convergence) Let the loss be defined as $\mathcal{L}(\widetilde{v}, \widetilde{u}) = \sum_{i=1}^4 w_i L_i(\widetilde{v}, \widetilde{u})$ for some arbitrary constants $w_i > 0$. Then we have the following bound between processes $X$ and $Y$:*

$$\sup_{t \leq T} \mathbb{E}\|X(t) - Y(t)\|^2 \leq C_T \mathcal{L}(\widetilde{v}, \widetilde{u}) \quad (110)$$

*where $C_T = \max_i \frac{w_i'}{w_i}$, $w_1' = 4e^{T(T+1)}$, $w_2' = e^{T(T+1)} \left( 2T \left( \|\nabla \widetilde{u}\|_\infty + \|\nabla \widetilde{v}\|_\infty \right)^2 e^{\left( \frac{1}{2} + 4\|\nabla \widetilde{v}\|_\infty \right) T} + 4 \right)$, $w_3' = 2T e^{T(T+1)} \left( 1 + \left( \|\nabla \widetilde{u}\|_\infty + \|\nabla \widetilde{v}\|_\infty \right)^2 e^{\left( \frac{1}{2} + 4\|\nabla \widetilde{v}\|_\infty \right) T} \right)$, $w_4' = 2T e^{T(T+1)}$.*

*Proof.* We are going to prove the bound:

$$\sup_{t \le T} \mathbb{E}\|X(t) - Y(t)\|^2 \le \sum_{i=1}^{4} w_i' L_i(\widetilde{v}, \widetilde{u}) \tag{111}$$

for constants that we obtain from the Lemmas above. Then we will use the following trick to get the bound with arbitrary weights:

$$\sum_{i=1}^{4} w_i' L_i(\widetilde{v}, \widetilde{u}) \le \sum_{i=1}^{4} \frac{w_i'}{w_i} w_i L_i(\widetilde{v}, \widetilde{u}) \le \big( \max_i \frac{w_i'}{w_i} \big) \sum_{i=1}^{4} w_i L_i(\widetilde{v}, \widetilde{u}) = C_T \mathcal{L}(\widetilde{v}, \widetilde{u}) \tag{112}$$

First, we apply Lemma F.5 to $Z = X - Y$ by noting that $\big[ X(t) - Y(t), X(t) - Y(t) \big]_t \equiv 0$ and $\|X(0) - Y(0)\|^2 = 0$ almost surely:

$$\mathbb{E}^Z \|X(t) - Y(t)\|^2 \tag{113}$$

$$= \int_0^t \mathbb{E}^Z \Big( 2 \langle \overleftarrow{\mathcal{L}^Z}(X(0) - Y(0)), X(s) - Y(s) \rangle \tag{114}$$

$$+ 2 \int_0^s \langle \overleftarrow{\mathcal{L}^Z} \overrightarrow{\mathcal{L}^Z}(X(z) - Y(z)), X(s) - Y(s) \rangle \mathrm{d}z \Big) \mathrm{d}s \tag{115}$$

$$\le \int_0^t \mathbb{E}^Z \Big( \big\| \overleftarrow{\mathcal{L}^Z}(X(0) - Y(0)) \big\|^2 + \|X(s) - Y(s)\|^2 \tag{116}$$

$$+ \int_0^s \Big( \big\| \overleftarrow{\mathcal{L}^Z} \overrightarrow{\mathcal{L}^Z}(X(z) - Y(z)) \big\|^2 + \|X(s) - Y(s)\|^2 \mathrm{d}z \Big) \Big) \mathrm{d}s \tag{117}$$

$$\le \int_0^t \mathbb{E}^Z \Big( \big\| \overleftarrow{\mathcal{L}^Z}(X(0) - Y(0)) \big\|^2 + (1 + T)\|X(s) - Y(s)\|^2 \tag{118}$$

$$+ \int_0^s \big\| \overleftarrow{\mathcal{L}^Z} \overrightarrow{\mathcal{L}^Z}(X(z) - Y(z)) \big\|^2 \mathrm{d}z \Big) \mathrm{d}s. \tag{119}$$

Then, using Corollary F.12, equation 102 and then Lemma F.10 we obtain that

$$\int_0^s \big\| \overleftarrow{\mathcal{L}^Z} \overrightarrow{\mathcal{L}^Z}(X(z) - Y(z)) \big\|^2 \mathrm{d}z = \int_0^s \big\| \overleftarrow{\mathcal{L}^Z} \overrightarrow{\mathcal{L}^Z} X(z) + \frac{1}{m} \nabla V(X(z)) \big\|^2 \mathrm{d}z \tag{120}$$

$$\le 2T \big( \|\nabla \widetilde{u}\|_\infty + \|\nabla \widetilde{v}\|_\infty \big)^2 e^{\big( \frac{1}{2} + 4\|\nabla \widetilde{v}\|_\infty \big) T} \big( L_3(\widetilde{v}, \widetilde{u}) + L_2(\widetilde{v}, \widetilde{u}) \big) + 4 L_1(\widetilde{v}, \widetilde{u}) + 4 L_2(\widetilde{v}, \widetilde{u}). \tag{121}$$

To deal with the remaining term involving $X(0) - Y(0)$ we observe that:

$$\int_0^t \mathbb{E}^Z \Big( \big\| \overleftarrow{\mathcal{L}^Z}(X(0) - Y(0)) \big\|^2 \le 2T L_3(\widetilde{v}, \widetilde{u}) + 2T L_4(\widetilde{v}, \widetilde{u}), \tag{122}$$

where we used triangle inequality. Combining obtained bounds yields:

$$\mathbb{E}^Z \|X(t) - Y(t)\|^2 \tag{123}$$

$$\le \int_0^t (1 + T)\|X(s) - Y(s)\|^2 \mathrm{d}s \tag{124}$$

$$+ 2T L_3(\widetilde{v}, \widetilde{u}) + 2T L_4(\widetilde{v}, \widetilde{u}) \tag{125}$$

$$+ 2T \big( \|\nabla \widetilde{u}\|_\infty + \|\nabla \widetilde{v}\|_\infty \big)^2 e^{\big( \frac{1}{2} + 4\|\nabla \widetilde{v}\|_\infty \big) T} \big( L_3(\widetilde{v}, \widetilde{u}) + L_2(\widetilde{v}, \widetilde{u}) \big) \tag{126}$$

$$+ 4 L_1(\widetilde{v}, \widetilde{u}) + 4 L_2(\widetilde{v}, \widetilde{u}) \tag{127}$$

$$= \int_0^t (1 + T)\|X(s) - Y(s)\|^2 \mathrm{d}s \tag{128}$$

$$+ 4 L_1(\widetilde{v}, \widetilde{u}) + \Big( 2T \big( \|\nabla \widetilde{u}\|_\infty + \|\nabla \widetilde{v}\|_\infty \big)^2 e^{\big( \frac{1}{2} + 4\|\nabla \widetilde{v}\|_\infty \big) T} + 4 \Big) L_2(\widetilde{v}, \widetilde{u}) \tag{129}$$

$$+ 2T \Big( 1 + \big( \|\nabla \widetilde{u}\|_\infty + \|\nabla \widetilde{v}\|_\infty \big)^2 e^{\big( \frac{1}{2} + 4\|\nabla \widetilde{v}\|_\infty \big) T} \Big) L_3(\widetilde{v}, \widetilde{u}) + 2T L_4(\widetilde{v}, \widetilde{u}). \tag{130}$$

Finally, using integral Grönwall's inequality Gronwall (1919), we have:

$$\mathbb{E}^Z \|X(t) - Y(t)\|^2 \tag{131}$$

$$\leq 4e^{T(T+1)} L_1(\widetilde{v}, \widetilde{u}) + e^{T(T+1)} \left( 2T \left( \|\nabla \widetilde{u}\|_\infty + \|\nabla \widetilde{v}\|_\infty \right)^2 e^{\left( \frac{1}{2} + 4\|\nabla \widetilde{v}\|_\infty \right)T} + 4 \right) L_2(\widetilde{v}, \widetilde{u}) \tag{132}$$

$$+ 2Te^{T(T+1)} \left( 1 + \left( \|\nabla \widetilde{u}\|_\infty + \|\nabla \widetilde{v}\|_\infty \right)^2 e^{\left( \frac{1}{2} + 4\|\nabla \widetilde{v}\|_\infty \right)T} \right) L_3(\widetilde{v}, \widetilde{u}) + 2Te^{T(T+1)} L_4(\widetilde{v}, \widetilde{u}) \tag{133}$$

$$\square$$

# G  APPLICATIONS

## G.1  BOUNDED DOMAIN $\mathcal{M}$

Our approach assumes that the manifold $\mathcal{M}$ is flat or curved. For bounded domains $\mathcal{M}$, e.g., like it is assumed in PINN or any other grid-based methods, our approach can be applied if we embed $\mathcal{M} \subset \mathbb{R}^d$ and define a new family of smooth non-singular potentials $V_\alpha$ on entire $\mathbb{R}^d$ such that $V_\alpha \to V$ when restricted to $\mathcal{M}$ and $V_\alpha \to +\infty$ on $\partial(\mathcal{M}, \mathbb{R}^d)$ (boundary of the manifold in embedded space) as $\alpha \to 0_+$.

## G.2  SINGULAR INITIAL CONDITIONS

It is possible to apply Algorithm 1 to $\psi_0 = \delta_{x_0} e^{iS_0(x)}$ for some $x_0 \in \mathcal{M}$. We need to augment the initial conditions with a parameter $\alpha > 0$ as $\psi_0 = \sqrt{\frac{1}{\sqrt{2\pi\alpha^2}} e^{-\frac{(x-x_0)^2}{2\alpha^2}}}$ for small enough $\alpha > 0$. In that case, $u_0(x) = -\frac{\hbar}{2m} \frac{(x-x_0)}{\alpha}$. We must be careful with choosing $\alpha$ to avoid numerical instability. It makes sense to try $\alpha \propto \frac{\hbar^2}{m^2}$ as $\frac{X(0)-x_0}{\alpha} = \mathcal{O}(\sqrt{\alpha})$. We evaluated such a setup in Appendix D.1.

## G.3  SINGULAR POTENTIAL

We must augment the potential to apply our method for simulations of the atomic nucleus with Bohr-Oppenheimer approximation (Woolley & Sutcliffe, 1977). A potential arising in this case has components of form $\frac{a_{ij}}{\|x_i - x_j\|}$. Basically, it has singularities when $x_i = x_j$. In case when $x_j$ is fixed, our manifold is $\mathcal{M} \backslash \{x_j\}$, which has a non-trivial cohomology group.

When such potential arises we suggest to augment the potential $V_\alpha$ (e.g., replace all $\frac{a_{ij}}{\|x_i - x_j\|}$ with $\frac{a_{ij}}{\sqrt{\|x_i - x_j\|^2 + \alpha}}$) so that $V_\alpha$ is smooth and non-singular everywhere on $\mathcal{M}$. In that case we have that $V_\alpha \to V$ as $\alpha \to 0$. With the augmented potential $V_\alpha$, we can apply stochastic mechanics to obtain an equivalent to quantum mechanics theory. Of course, augmentation will produce bias, but it will be asymptotically negligent as $\alpha \to 0$.

## G.4  MEASUREMENT

Even though we have entire trajectories and know positions for each moment, we should carefully interpret them. This is because they are not the result of the measurement process. Instead, they represent hidden variables (and $u, v$ represent global hidden variables – what saves us from the Bells inequalities as stochastic mechanics is non-local (Nelson, 1966)).

For a fixed $t \in [0, T]$, the distribution of $X(t)$ coincides with the distribution $\mathbf{X}(t)$ for $\mathbf{X}$ being position operator in quantum mechanics. Unfortunately, a compound distribution $(X(t), X(t'))$ for $t \neq t'$ may not correspond to the compound distribution of $(\mathbf{X}(t), \mathbf{X}(t'))$; for details see Nelson (2005). This is because each $\mathbf{X}(t)$ is a result of the *measurement process*, which causes the wave function to collapse (Derakhshani & Bacciagaluppi, 2022).

Trajectories $X_i$ are as if we could measure $\mathbf{X}(t)$ without causing the collapse of the wave function. To use this approach for predicting some experimental results involving multiple measurements, we

need to re-run our method after each measurement process with the measured state as the new initial condition. This issue is not novel for stochastic mechanics. There is the same problem in classical quantum mechanics.

This "contradiction" is resolved once we realize that $\mathbf{X}(t)$ involves measurement, and thus, if we want to calculate correlations of $(\mathbf{X}(t), \mathbf{X}(t'))$ for $t < t'$ we need to do the following:

- Run Algorithm 1 with $\psi_0, V(x,t)$ and $T = t$ to get $\widetilde{u}, \widetilde{v}$.
- Run Algorithm 2 with $\widetilde{u}, \widetilde{v}$, $\psi_0$ to get $\{X_{Nj}\}_{j=1}^B$ – $B$ last steps from trajectories $X_i$ of length $N$.
- For each $X_{Nj}$ in the batch we need to run Algorithm 1 with $\psi_0 = \delta_{X_{Nj}}, V'(x,t') = V(x, t' + t)$ (assuming that $u_0 = 0, v_0 = 0$) and $T = t' - t$ to get $\widetilde{u}_j, \widetilde{v}_j$.
- For each $X_{Nj}$ run Algorithm 2 with batch size $B = 1$, $\psi_0 = \delta_{X_{Nj}}, \widetilde{u}_j, \widetilde{v}_j$ to get $X'_{Nj}$.
- Output pairs $\left\{ (X_{N,j}, X'_{N,j}) \right\}_{j=1}^B$.

Then the distribution of $(X_{N,j}, X'_{N,j})$ will correspond to the distribution of $(\mathbf{X}(t), \mathbf{X}(t'))$. This is well described and proven in Derakhshani & Bacciagaluppi (2022). Therefore, it is possible to simulate the right correlations in time using our approach, though, it may require learning $2(B + 1)$ models. The promising direction of future research is to consider $X_0$ as a feature for the third step here and, thus, learn only $2 + 2$ models.

### G.5 OBSERVABLES

To estimate any scalar observable of form $\mathbf{Y}(t) = y(\mathbf{X}(t))$ in classic quantum mechanics one needs to calculate:

$$\langle \mathbf{Y} \rangle_t = \int_{\mathcal{M}} \overline{\psi(x,t)} y(x) \psi(x,t) \mathrm{d}x.$$

In our setup, we can calculate this using the samples $X_{\left[\frac{Nt}{T}\right]} \approx X(t) \sim \left| \psi(\cdot, t) \right|^2$:

$$\langle \mathbf{Y} \rangle_t \approx \frac{1}{B} \sum_{j}^{B} y(X_{\left[\frac{Nt}{T}\right]j}),$$

where $B \geq 1$ is the batch size, $N$ is the time discretization size. The estimation error has magnitude $\mathcal{O}(\frac{1}{\sqrt{B}} + \epsilon + \varepsilon)$, where $\epsilon = \frac{T}{N}$ and $\varepsilon$ is the $L_2$ error of recovering true $u, v$. In our paper, we have not bounded $\varepsilon$ but provide estimates for it in our experiments against the finite difference solution.[8]

### G.6 WAVE-FUNCTION

Recovering the wave function from $u, v$ is possible using a relatively slow procedure. Our experiments do not cover this because our approach's main idea is to avoid calculating wave function. But for the record, it is possible. Assume we solved equations for $u, v$. We can get the phase and density by integrating Equation (21):

$$S(x,t) = S(x,0) + \int_0^t \Big( \frac{1}{2m} \langle \nabla, u(x,t) \rangle + \frac{1}{2\hbar} \|u(x,t)\|^2 - \frac{1}{2\hbar} \|v(x,t)\|^2 - \frac{1}{\hbar} V(x,t) \Big) \mathrm{d}t,$$
(134)

$$\rho(x,t) = \rho_0(x) \exp \Big( \int_0^t \big( -\langle \nabla, v(x,t) \rangle - \frac{2m}{\hbar} \langle u(x,t), v(x,t) \rangle \big) \Big) \mathrm{d}t$$
(135)

This allows us to define $\psi = \sqrt{\rho(x,t)} e^{iS(x,t)}$, which satisfies the Schrödinger equation (1). Suppose we want to estimate it over a grid with $N$ time intervals and $\lceil \sqrt{N} \rceil$ intervals for each coordinate (a typical recommendation for Equation (1) is to have a grid satisfying $\mathrm{d}x^2 \approx \mathrm{d}t$). It leads to a sample complexity of $\mathcal{O}(N^{\frac{d}{2}+1})$, which is as slow as other grid-based methods for quantum mechanics. The error in that case will also be $\mathcal{O}(\sqrt{\epsilon} + \varepsilon)$ (Smith & Smith, 1985).

---

[8]If we are able to reach $\mathcal{L}(\theta) = 0$ then essentially $\varepsilon = 0$. We leave bounding $\varepsilon$ by $\mathcal{L}(\theta_\tau)$ for future work.

# H ON CRITICISM OF STOCHASTIC MECHANICS

Three major concerns arise regarding stochastic mechanics developed by Nelson (1966); Guerra (1995):

- The proof of the equivalence of stochastic mechanics to classic quantum mechanics relies on an implicit assumption of the phase $S(x,t)$ being single-valued (Wallstrom, 1989).
- If there is an underlying stochastic process of quantum mechanics, it should be non-Markovian (Nelson, 2005).
- For a quantum observable, e.g., a position operator $\mathbf{X}(t)$, a compound distribution of positions at two different timestamps $t, t'$ does not match distribution of $(\mathbf{X}(t), \mathbf{X}(t'))$ (Nelson, 2005).

Appendix G.4 discusses why a mismatch of the distributions is not a problem and how we can adopt stochastic mechanics with our approach to get correct compound distributions by incorporating the measurement process into the stochastic mechanical picture.

## H.1 ON "INEQUIVALENCE" TO SCHRÖDINGER EQUATION

This problem is explored in the paper by Wallstrom (1989). Firstly, the authors argue that proofs of the equivalency in Nelson (1966); Guerra (1995) are based on the assumption that the wave function phase $S$ is single-valued. In the general case of a multi-valued phase, the wave functions are identified with sections of complex line bundles over $\mathcal{M}$. In the case of a trivial line bundle, the space of sections can be formed from single-valued functions, see Alvarez (1986). The equivalence class of line bundles over a manifold $\mathcal{M}$ is called Picard group, and for smooth manifolds, $\mathcal{M}$ is isomorphic to $H^2(\mathcal{M}, \mathbb{Z})$, so-called second cohomology group over $\mathbb{Z}$, see Prieto & Vitolo (2014) for details. Elements in this group give rise to non-equivalent quantizations with irremovable gauge symmetry phase factor.

Therefore, *in this paper, we assume that $H^2(\mathcal{M}, \mathbb{Z}) = 0$*, which allows us to eliminate all criticism about non-equivalence. Under this assumption, stochastic mechanics is *equivalent* indeed. This condition holds when $\mathcal{M} = \mathbb{R}^d$. Though, if a potential $V$ has singularities, e.g., $\frac{a}{\|x-x_*\|}$, then we should exclude $x_*$ from $\mathbb{R}^d$ which leads to $\mathcal{M} = \mathbb{R}^d \backslash \{x_*\}$ and this manifold satisfies $H^2(\mathcal{M}, \mathbb{Z}) \cong \mathbb{Z}$ (May, 1999), which essentially leads to "counterexample" provided in Wallstrom (1989). We suggest a solution to this issue in Appendix G.2.

## H.2 ON "SUPERLUMINAL" PROPAGATION OF SIGNALS

We want to clarify why this work should not be judged from perspectives of *physical realism*, *correspondence to reality* and *interpretations* of quantum mechanics. This tool gives the exact predictions as classical quantum mechanics at a moment of measurement. Thus, we do not care about a superluminal change in the drifts of entangled particles and other problems of the Markovian version of stochastic mechanics.

## H.3 NON-MARKOVIANITY

Nelson believes that an underlying stochastic process of reality should be non-Markovian to avoid issues with the Markovian processes like superluminal propagation of signals (Nelson, 2005). Even if such a process were proposed in the future, it would not affect our approach. In stochastic calculus, there is a beautiful theorem from Gyöngy (1986):

**Theorem H.1.** *Assume $X(t), F(t), G(t)$ are adapted to Wiener process $W(t)$ and satisfy:*

$$\mathrm{d}X(t) = F(t)\mathrm{d}t + G(t)\mathrm{d}\overrightarrow{W}.$$

*Then there exist a Markovian process $Y(t)$ satisfying*

$$\mathrm{d}Y(t) = f(Y(t), t)\mathrm{d}t + g(Y(t), t)\mathrm{d}\overrightarrow{W}$$

*where $f(x,t) = \mathbb{E}(F(t)\|X(t) = x), g(x,t) = \sqrt{\mathbb{E}(G(t)G(t)^T\|X(t) = x)}$ and such that $\forall t$ holds* $\mathrm{Law}(X(t)) = \mathrm{Law}(Y(t))$.

This theorem tells us that we already know how to build a process $Y(t)$ without knowing $X(t)$; it is stochastic mechanics by Nelson (Nelson, 1966; Guerra, 1995) that we know. From a numerical perspective, we better stick with $Y(t)$ as it is easier to simulate, and as we explained, we do not care about correspondence to reality as long as it gives the same final results.

### H.4 GROUND STATE

Unfortunately, our approach is unsuited for the ground state estimation or any other stationary state. FermiNet (Pfau et al., 2020) does a fantastic job already. The main focus of our work is time evolution. It is possible to estimate some observable $\mathbf{Y}$ for the ground state if its energy level is unique and significantly lower than others. In that case, the following value approximately equals the group state observable for $T \gg 1$:

$$\langle \mathbf{Y} \rangle_{ground} \approx \frac{1}{T} \int_0^T \langle \mathbf{Y} \rangle_t \mathrm{d}t \approx \frac{1}{NB} \sum_{i=1}^N \sum_{j=1}^B y(X_{ij})$$

This works only if the ground state is unique, and the initial conditions satisfy $\int_{\mathcal{M}} \overline{\psi_0} \psi_{ground} \mathrm{d}x \neq 0$, and its energy is well separated from other energy levels. In that scenario, oscillations will cancel each other out.

## I FUTURE WORK

This section discusses possible directions for future research. Our method is a promising direction for fast quantum mechanics simulations, but we consider the most straightforward setup in our work. Possible future advances include:

- In our work, we consider the simplest integrator of SDE (Euler-Maryama), which may require setting $N \gg 1$ to achieve the desired accuracy. However, a higher-order integrator (Smith & Smith, 1985) or an adaptive integrator (Ilie et al., 2015) should achieve the desired accuracy with much lower $N$.

- It should be possible to extend our approach to a wide variety of other quantum mechanical equations, including Dirac and Klein-Gordon equations used to account for special relativity (Serva, 1988; Blanchard et al., 2005), a non-linear Schrödinger Equation (1) equation used in condensed matter physics (Serkin & Hasegawa, 2000) by using McKean-Vlasov SDEs and the mean-field limit (Buckdahn et al., 2017a;b; dos Reis et al., 2022), and the Shrödinger equation with a spin component (Dankel, 1970; De Angelis et al., 1991).

- We consider a rather simple, fully connected architecture of neural networks with $\tanh$ activation and three layers. It might be more beneficial to consider specialized architectures for quantum mechanical simulations, e.g., Pfau et al. (2020).

- Many practical tasks require knowledge of the error magnitude. Thus, providing explicit bounds on $\varepsilon$ in terms of $\mathcal{L}(\theta_M)$ is critical.