# OpenReview forum: "Deep Stochastic Mechanics"
_ICLR.cc/2024/Conference — Submitted to ICLR 2024_

### Official Review · Reviewer_APZm · 2023-10-13

**Soundness:** 2 fair
**Presentation:** 1 poor
**Contribution:** 3 good
**Rating:** 6
**Confidence:** 4

**Summary:**

This work introduces an idea based on stochastic mechanics to provide approximate solutions to the time-dependent Schroedinger's equation, and neural parameterizations of various quantities involved.

**Strengths:**

The idea is very innovative and I especially liked the use of Nelsonian dynamics.

**Weaknesses:**

The approach is limited to very simple/small systems (harmonic oscillators and interacting distinguishable particles of only a handful of particles).

The discussion is also quite limited, and bypasses a vast amount of literature in the field, including time-dependent variational wave functions based on neural network parameterizations.

For example:

"Another family of approaches, FermiNet (Pfau et al., 2020) or PauliNet (Hermann et al., 2020),
reformulates the problem (1) as maximization of an energy functional that depends on the solution of
the stationary Schrodinger equation. This approach sidesteps the curse of dimensionality but cannot ¨
be adapted to the time-dependent wave function setting considered in this paper."

The approach that reformulates (1) as a static, variational problem does not date to 2020 and it is as old as quantum mechanics itself.
Also, neural variational parameterizations of the wave function are routinely used to solve the time-dependent Schroedinger's equation. These approaches are based on Dirac and Frenkel's time-dependent variational principle, and have been used in several works, starting e.g. from https://www.science.org/doi/10.1126/science.aag2302, https://journals.aps.org/prl/abstract/10.1103/PhysRevLett.125.100503 and many more.

**Questions:**

1) The approach is shown for very small systems, and in cases that are solved easily with many other approaches (including, for example, using a discrete basis and using the time-dependent variational principle with a neural network wave function). Do the authors have a sense of the scaling in terms of number of particles/ can they show a case that goes beyond what can be simulated exactly?

2) A grid-based discretization seems to be used in the paper, at least in Figure 1 d. Can the authors clarify how does the discretization enter their algorithm?

---

> ### Author Response · Authors · 2023-11-18
> **Authors reply 1/2**
>
> Thank you for your review. Let us address your questions:
> > **W1** *The approach is limited to very simple/small systems…*
>
> Even in the “simple” settings considered in our paper, existing algorithms (e.g. Crank-Nicolson) encounter significant limitations and scaling issues both in terms of computation time and memory requirements. The core contribution of our submission is demonstrating an approach that sidesteps these challenges. We detail in our future work section our interest in taking this work and expanding it to more complex settings, but this initial work was an essential first step and an important contribution in its own right.
>
> > **W2** *The discussion is also quite limited, and bypasses a vast amount of literature in the field, including time-dependent variational wave functions based on neural network parameterizations. For example:  "Another family of approaches, FermiNet (Pfau et al., 2020) or PauliNet (Hermann et al., 2020), reformulates the problem (1) as maximization of an energy functional that depends on the solution of the stationary Schrodinger equation. This approach sidesteps the curse of dimensionality but cannot ¨ be adapted to the time-dependent wave function setting considered in this paper."
> The approach that reformulates (1) as a static, variational problem does not date to 2020 and it is as old as quantum mechanics itself. Also, neural variational parameterizations of the wave function are routinely used to solve the time-dependent Schroedinger's equation. These approaches are based on Dirac and Frenkel's time-dependent variational principle, and have been used in several works, starting e.g. from https://www.science.org/doi/10.1126/science.aag2302, https://journals.aps.org/prl/abstract/10.1103/PhysRevLett.125.100503 and many more.*
>
> Thank you for your suggestion! We are happy to cite and include more discussion of the two mentioned papers (Carleo et al., Schmitt et al.). We agree that the variational approach and its time-dependent variant are old and well-studied methods. Both Carleo et al. and Schmitt et al. are neural network quantum state methods that are applicable to discrete quantum systems, e.g. spin systems, whereas our method is for continuous quantum systems. We are not aware of a time-dependent variant of a continuous neural network quantum state. In addition, unlike our approach, the neural network quantum states discussed in these two papers require expensive MCMC sampling in order to generate samples, which must be done for each training step and each time step in the time evolution.
>
> > **Q1** *The approach is shown for very small systems, and in cases that are solved easily with many other approaches (including, for example, using a discrete basis and using the time-dependent variational principle with a neural network wave function).  Do the authors have a sense of the scaling in terms of number of particles/ can they show a case that goes beyond what can be simulated exactly? *
>
> This question is related to **W1**. In our paper, there is an experiment in Section 5.4 with many non-interacting particles. In particular, our approach shows linear convergence time for a specific case of the harmonic oscillator (Figure 4). In contrast, simulation of this setting (without prior knowledge) using traditional solvers would require a grid of size $N^d$. Our approach bypasses this issue, and our empirical findings support it.
>
> There are experiments in the original submission that do correspond to interacting particles. **For this rebuttal, we ran new experiments for larger d to show scaling properties more clearly.** The following Tables illustrate time and memory usage for the Crank–Nicolson (C-N) solver for d=2, 3, 4 and training time/time per epoch and memory usage of our DSM method for d=2, 3, 4, and 5. Also, Figure 11 in **the revised supplementary material D.5.1** shows predicted densities for different dimensions. Note that while we don’t have the baseline for d=5, we believe DSM predictions are still reasonable.

---

> > ### Author Response · Authors · 2023-11-18
> > **Authors reply 2/2**
> >
> > Table 1. Time to get a solution and memory usage of the Crank-Nicolson method for different problem dimensions.
> > |      |       d=2      |       d=3      |       d=4      |
> > | ----------- | ----------- |----------- | ----------- |
> > | Time (s)      | 0.75       |    35.61   |  2363    |
> > | Memory (Gb)   | 7.4        |10.6        | 214       |
> >
> >
> > Table 2. Training time, time per epoch, and memory usage of our DSM method for different problem dimensions.
> > |      |       d=2      |       d=3      |       d=4      |       d=5   |
> > | ----------- | ----------- |----------- | ----------- |  ----------- |
> > | Training time (min)      | 29.5       |    60.3   |  97.5    |  154.0 |
> > | Time per epoch (s/ep)      | 0.52       |    1.09   |  1.16    |  1.24 |
> > | Memory (Gb)   | 17.0      | 22.5      |  28.0    | 33.5 |
> >
> > Memory: In particular, DSM memory usage and time per epoch grow linearly in d (according to our theory and evident in our numerical results) in contrast to the C-N solver, whose memory usage grows exponentially since discretization matrices are of size  $N^d \times N^d$.
> > We cannot run the C-N method for d > 4 with the amount of memory available in our computer system. The results show that our method is far more memory efficient for larger d.
> >
> > Compute time: While our DSM total compute times (including training time) are longer than C-N compute times for small d, the trend as d increases suggests the computational efficiency of our DSM method scales much better with d than the C-N method. In this example, the potential function has computational complexity $O(d^2)$ as it has all pairwise interactions. It implies that the training time complexity is bounded to be quadratic in the best case for all algorithms. Computation complexity of other DSM loss terms that involve computationally heavy differential operators with autograd remains to be linear. We assume domination of the differential operators computations explains why time per epoch does not seem to grow quadratically.  These findings support the claim that our DSM method training time scaling is more favourable compared to scaling of the classical methods.
> >
> > > **Q2** *A grid-based discretization seems to be used in the paper, at least in Figure 1 d. Can the authors clarify how does the discretization enter their algorithm?*
> >
> > You’re right; our DSM method has discretization in time, but not in $X$. Figure 1a demonstrates a DSM training procedure: we run it for $\tau$ epochs, and for every epoch, we predict $X_i$ iteratively for every time step $i$ ($X_i$ corresponds to a particle position at time point $i \in 0, .., N$). Figures 1b and 1c show examples of sampled trajectories at early epochs and a final epoch. Figure 1d illustrates discretization for grid-based methods (for example, finite difference schemes or PINNs). We hope our explanation makes sense.
> >
> > We appreciate your feedback, and we are ready to address any further concerns or suggestions you would like to discuss.

---

> > ### Comment · Reviewer_APZm · 2023-11-18
> > **Comparison to time-dependent variational principle - SOTA system sizes**
> >
> > > Thank you for your suggestion! We are happy to cite and include more discussion of the two mentioned papers (Carleo et al., Schmitt et al.). We agree that the variational approach and its time-dependent variant are old and well-studied methods.
> >
> > You are welcome! The time-dependent variational principle is the most widely used algorithm to solve the time-dependent Schroedinger equation for many-body systems and complex, time-dependent neural network wave functions, thus not properly discussing it is indeed quite problematic.
> >
> > > Both Carleo et al. and Schmitt et al. are neural network quantum state methods that are applicable to discrete quantum systems, e.g. spin systems, whereas our method is for continuous quantum systems. We are not aware of a time-dependent variant of a continuous neural network quantum state.
> >
> > Notice that there are also applications of the same variational technique to continuous space for bosonic systems of up to around 100 particles https://journals.aps.org/prx/abstract/10.1103/PhysRevX.7.031026 . Conceptually, in fact one can use exactly the same approach used for all the other simulations on lattice systems, since it is possible to discretize the laplacian with finite differences if using a spatial grid. Even if spatial coordinates are discretized, the scaling to solve the time-dependent SE variationally is still polynomial, and does not increase exponentially with the number of particles. I understand the authors here work directly with functions that are continuos, introduce a novel and interesting framework with exact sampling, etc etc. Still, I wouldn't argue, in general, that works using a finite-difference (discrete basis) solver are not solving the original PDE problem... since most state-of-the art solvers are based on finite differences, in most cases.
> >
> > Let me also clarify that the wave functions employed in the cited paper are pre-neural networks, thus maybe a bit easier in terms of number of parameters to be optimized (maybe in the 100k parameters regime, or so). Still, in the present submission we are seeing simulations for interacting bosons up to only 2 particles (please correct me if I am mistaken), and others on the harmonic oscillator. In this very sense the examples shown here are a bit too simple to be competitive with existing methods doing 100-particle interacting boson dynamics with high accuracy.
> >
> > > In addition, unlike our approach, the neural network quantum states discussed in these two papers require expensive MCMC sampling in order to generate samples, which must be done for each training step and each time step in the time evolution.
> >
> > In principle, I agree with this statement (even though MCMC is typically efficient on these systems). Nonetheless, if the price to have exact sampling is a very complex algorithm that does not easily scale to large number of particles, this should be also factored in.

---

> > > ### Comment · Reviewer_APZm · 2023-11-18
> > > **Additional references**
> > >
> > > I incorrectly mentioned in the previous comment that no continuos-space time-dependent neural network was present in the literature (and evolved with the TDVP, to solve the SE).
> > >
> > > There are at least two other relevant references:
> > > https://journals.aps.org/prxquantum/abstract/10.1103/PRXQuantum.4.040302
> > > https://iopscience.iop.org/article/10.1088/2632-2153/aca317/pdf
> > >
> > > Notice that the last one also uses exact sampling (and it is in about 8 dimensions). The former is doing MCMC, and it's in d=64.

---

> ### Author Response · Authors · 2023-11-18
>
> Thank you for bringing up those references! They are indeed very relevant. We will make sure to include them in the relevant work section.
>
> >we are seeing simulations for interacting bosons up to only 2 particles (please correct me if I am mistaken),
>
> In our reply to your comment (2/2), and in the revision (supplementary material D.5.1), we shared results for five interacting bosons.  We have not encountered issues with five particles. Currently, we are only limited in GPU memory that is available to us. The memory grows linearly, as we show in the reply. Therefore, we do not foresee issues running it for even larger systems.
>
> >Still, I wouldn't argue, in general, that works using a finite-difference (discrete basis) solver are not solving the original PDE problem... since most state-of-the art solvers are based on finite differences, in most cases.
>
> We agree with that statement. There are nuances, though, that to avoid exponential scaling, they adapt it to sub-classes of the quantum problems, and generally, they do not scale polynomially.
>
> >Notice that there are also applications of the same variational technique to continuous space for bosonic systems of up to around 100 particles https://journals.aps.org/prx/abstract/10.1103/PhysRevX.7.031026 .
>
> In this work, the authors introduce the time-dependent variational Monte Carlo method for continuous-space Bose gases and adapt their method to work well, incorporating external prior information about the Bijl-Dingle-Jastrow-Feenberg expansion of the wave function and truncating it. As the authors mention, their approach is an approximation, and going beyond three terms in the expansion is computationally prohibitive. Our approach has no such limitations embedded, and is not limited to problems where such expansion is available. Thus, it aims to solve the problem without simplifying it.
>
> >There are at least two other relevant references: https://journals.aps.org/prxquantum/abstract/10.1103/PRXQuantum.4.040302 https://iopscience.iop.org/article/10.1088/2632-2153/aca317/pdf
>
> In the first one, the authors postulate the Hamiltonian that captures the relevant physics of superconducting Josephson junctions and explicitly incorporate external knowledge into their VMC approach to make it efficient and scalable. Such knowledge can also be included in our approach, but that is beyond the scope of the work.
>
> When it comes to the second one, we can cite the authors directly:  *"Additionally, we found it challenging to model distributions whose tail behaviour deviated from that of the latent space distribution... because its form is not known beforehand, one cannot expect to accurately model the distribution on the entire domain."*
>
> That implies that while those approaches can scale well for some systems, they still use external knowledge. While it is fascinating how well they work for these cases, it is not a correct comparison with our approach, as, in principle, it is possible to incorporate such external knowledge in our method and make it much more efficient for such specific problems. To take it to an extreme, we can derive an analytical solution for zero potential and singular initial conditions, but we can't derive an analytical solution for every possible problem. Likewise, when comparing VMC adapted to a particular problem, it needs a lot of external theoretical knowledge specific to the potential to make it scalable.
>
> Moreover, in the application examples, this work relies on a particular type of problem to obtain exact samples. In many examples, it is a stochastic differential equation, which exists again due to external knowledge. In general, for quantum mechanics, Nelsonian equations are only Markovian equations. So, from that perspective, we are similar to them by doing exact sampling in the most general sense for quantum mechanics.
>
> > Nonetheless, if the price to have exact sampling is a very complex algorithm that does not easily scale to large number of particles, this should be also factored in.
>
> Well, strictly speaking, the price of research hours of adopting VMC to a particular potential should also be factored in.
>
> It would've been more correct to compare our approach with non-specialized to particular classes of problem approaches. As the No-Free-Lunch theorem states, finding an approach that works better for some sub-class is always possible. Thus, we compare and claim everything compared to other methods that are not limited in such ways.
>
> We will include such discussion in the relevant works section. We are grateful for bringing these points up, as having such discussion will significantly improve our work!

---

> > ### Comment · Reviewer_APZm · 2023-11-19
> > **Further comments**
> >
> > > In our reply to your comment (2/2), and in the revision (supplementary material D.5.1)
> >
> > I seem to be unable to access a revised version at this stage
> >
> > > We agree with that statement. There are nuances, though, that to avoid exponential scaling, they adapt it to sub-classes of the quantum problems, and generally, they do not scale polynomially.
> >
> > The t-VMC / tDVP approaches I mentioned solve the time-dependent SE in the variational manifold in polynomial time, under a single assumption 1. the MCMC sampling from |Psi(R,t)|^2 is efficient . Otherwise, there is no notion of subclass of quantum problems, in the sense that, in fact, the tDVP can be actually applied to a broader class of quantum problems than your approach. Your approach is heavily specialized to the non-relativistic Schr. equation in real space. The tDVP has been used not only fo this family of hamiltonians but also for spin models, rotors, and much more. Also notice that also your approach is strictly speaking not guarantueed to be polynomially scaling, since you need to perform an optimization of a complex high-dimensional function. Thus, I wouldn't claim that tDVP approaches on discrete systems (as for example, discretizing the laplacian operator on a grid) are not polynomially scaling, whereas your approach is. In both cases, there are subtle nuances.
> >
> > > In this work, the authors introduce the time-dependent variational Monte Carlo method for continuous-space Bose gases and adapt their method to work well, incorporating external prior information about the Bijl-Dingle-Jastrow-Feenberg expansion of the wave function and truncating it. As the authors mention, their approach is an approximation, and going beyond three terms in the expansion is computationally prohibitive. Our approach has no such limitations embedded, and is not limited to problems where such expansion is available. Thus, it aims to solve the problem without simplifying it.
> >
> > > In the first one, the authors postulate the Hamiltonian that captures the relevant physics of superconducting Josephson junctions and explicitly incorporate external knowledge into their VMC approach to make it efficient and scalable. Such knowledge can also be included in our approach, but that is beyond the scope of the work.
> >
> > I am not sure what exactly you mean here. In both cases, they have just picked a specific hamiltonian to showcase the method, as much as you have picked the harmonic oscillator and gaussian-interacting particles. Their approach is generalizable very easily to your case, for example. There is no specific external knowledge but that of the Hamiltonian. In the second case, the ansatz they use is a neural network that can be used in many different problems. In the first case, the variational freedom is smaller, I agree, but the method would work also with a neural network wave function. Incidentally, the simple problems you are studying in your submission are such that they can be very effectively solved with the Bijl-Dingle-Jastrow-Feenberg variational form.
> >
> > Also notice that your approximation is in the variational form you use to encode the velocities u_theta v_theta. In this sense I am not sure what do you exactly mean that your approach has "no such limitations embedded". You are also limited by a variational choice.
> >
> > > Well, strictly speaking, the price of research hours of adopting VMC to a particular potential should also be factored in.
> > > It would've been more correct to compare our approach with non-specialized to particular classes of problem approaches. As the No-Free-Lunch theorem states, finding an approach that works better for some sub-class is always possible. Thus, we compare and claim everything compared to other methods that are not limited in such ways.
> >
> > Again, this is problematic, because you seem to believe that VMC approaches based on neural networks are highly specialized. They are not (e.g. FermiNet, that you cite as a prominent example of VMC approach for ground states in table 1. Not sure why, by the way, only FermiNet is mentioned in there and not, say, PauliNet or the many other neural-network based VMC methods... but ok).
> >
> > I do not mean to diminish the importance of your work, which is nice and novel, but if you wish to convince a broad community that your approach is competitive, understanding and comparing against existing state of the art is essential. In your submission, unfortunately, you are only comparing to exponentially scaling exact methods, not against other variational methods to solve the tD SE that have been used in the past 6/7 years to simulate up to 100 or more particles with high precision.

---

> ### Author Response · Authors · 2023-11-19
>
> >  Your approach is heavily specialized to the non-relativistic Schr. equation in real space.
>
> In fact, it is not. We limit the description to such case as the theoretical basis is already heavy enough. Spin extension of Nelsonian equations is described in Nelson E., “Quantum Fluctuations”, relativistic versions of the equations are described in Serva M., “ Relativistic stochastic processes associated to Klein-Gordon equation” and also in https://arxiv.org/pdf/2004.11983.pdf, and relativistic spin in https://iopscience.iop.org/article/10.1209/0295-5075/14/2/001/pdf. All Nelsonian equations are extendable into non-flat manifolds in a “plug-in” way without changing the theory (we again refer to Nelson’s work here); a slight modification from Itô integral to Stratonovich one in algorithm even seems to make it work with manifolds with non-trivial cohomology group (i.e., with irremovable gauge symmetry), for example in https://arxiv.org/pdf/quant-ph/0007015.pdf https://arxiv.org/pdf/2304.07524.pdf.
>
> That's all to highlight one point: we are not heavily specialized. Our approach is extendable to even Lorentzian manifolds; the only thing we need to “change” is to make the code more general by ‘upgrading’ the differential operators to coordinate-invariant.
>
> > under a single assumption 1. the MCMC sampling from |Psi(R,t)|^2 is efficient . Otherwise, there is no notion of subclass of quantum problems
>
> We want to highlight one of the most important theoretical results of our work (developed in Section F): our sampling bounds do not depend on such things as a uniform spectral gap that is inevitable for MCMC (https://arxiv.org/pdf/1112.1392.pdf), which is bound to be exponential in dimension without heavy-task specialization, e.g. https://www.jstor.org/stable/24520134.
>
> The very assumption that MCMC sampling is efficient implies heavy task-specific adaptation of VMC, and, in the general theoretical sense, it is bound to be exponential. On the contrary, our approach avoids such issues due to the fundamental properties of Nelsonian PDEs, as we show in Section F.
>
> > Also notice that your approximation is in the variational form you use to encode the velocities u_theta v_theta. In this sense I am not sure what do you exactly mean that your approach has "no such limitations embedded". You are also limited by a variational choice.
>
> There is indeed a choice of neural networks $u_\theta$, $v_\theta$. However, we enjoy the universal approximation property in our proof-of-concept work and do not adjust it to make it work. This is to make our point about fundamental aspects of the framework more straightforward. We leave making our approach much more efficient for each problem by finding task-specific neural architectures for future work.
>
> > Again, this is problematic, because you seem to believe that VMC approaches based on neural networks are highly specialized. They are not (e.g. FermiNet, that you cite as a prominent example of VMC approach for ground states in table 1. Not sure why, by the way, only FermiNet is mentioned in there and not, say, PauliNet or the many other neural-network based VMC methods... but ok).
>
> When comparing VMC with us, we need the following adjustments to make it fair:
> 1. From FermiNet, remove all pairwise interactions from model features or add them to our approach.
> 2. Either remove Slater-determinant-based architecture in FermiNet and replace it with our simple architecture or use it within our approach.
> 3. For all VMC approaches, replace task-dependent efficient MCMC samplers with some generic sampler, such as Langevin SDE. Their efficiency, in general, is only possible as a result of task adaptation, as all interaction potentials are non-convex. Therefore, it implies exponential dependence in the spectral gap (e.g., see Eq. 4.3 in https://arxiv.org/pdf/1702.03849.pdf).
> 4. …
>
> We could remove all external knowledge specific to the potential, or incorporate it problem-wise into our method. From a No-free-lunch perspective, we are always bound to fail compared to methods that rely on some external additional perspective. To make the comparison fair, we either need to level the field by removing all externalities from the corresponding methods or do the amount of research like in all referenced by your papers to integrate it into our approach for each problem individually.
>
> And by no means we want to diminish the importance of VMC! We agree that VMC is a powerful framework, and we must discuss it with citations of all the works you mentioned. We are very grateful for such a fruitful discussion and agree that the current work's limitations need to be discussed in Relevant Work and Limitations Section, which we will add in the revision.

---

> > ### Comment · Reviewer_APZm · 2023-11-20
> >
> > >That's all to highlight one point: we are not heavily specialized. Our approach is extendable to even Lorentzian manifolds; the only thing we need to “change” is to make the code more general by ‘upgrading’ the differential operators to coordinate-invariant.
> >
> > My point was that tDVP has been used both in continuous and discrete Hamiltonian operators. The approach you are showing here is specialized to (bosonic) systems in continuous space. For example, your approach, as it is, wouldn't work for fermions, simply because the phase is discontinuous and its gradient is intrinsically not well defined.
> > Again, I don't mind if your approach works only for bosons, it's all fine. All this line of comments was coming from the fact that you were mentioning that your approach is more general than tDVP...  which is clearly not the case.
> >
> > > All that, to highlight, that the very assumption that MCMC sampling is efficient implies heavy task-specific adaptation of VMC and in general theoretical sense it is bound to be exponential. On the contrary our approach, avoids such issues due to fundamental properties of Nelsonian PDEs, see Section F.
> >
> > Very simple MCMC schemes are effective in sampling physically relevant wave functions up to thousand or more particles. I agree that having an exact sampling is very nice. However, there exist very efficient MCMC schemes to sample from thousands of particles. This is not really the bottleneck of most variational approaches to quantum mechanics.
> >
> > > When comparing VMC with us, we need the following adjustments to make it fair:
> > From FermiNet, remove all pairwise interactions from model features or add them to our approach.
> > Either remove Slater-determinant-based architecture in FermiNet and replace it with our simple architecture or use it within our approach.
> > For all VMC approaches, replace task-dependent efficient MCMC samplers with some generic sampler, such as Langevin SDE. Their efficiency, in general, is only possible as a result of task adaptation, as all interaction potentials are non-convex. Therefore, it implies exponential dependence in the spectral gap (e.g., see Eq. 4.3 in https://arxiv.org/pdf/1702.03849.pdf).
> > …
> >
> > Let me stress this again once more. You cannot equate VMC with "physically inspired" functions. VMC (and t-VMC/t-DVP) are just frameworks to optimize a given variational state (that could even be exactly the same functions you are taking in your work) to either approximate the ground state (VMC) or the dynamics (t-VMC/t-DVP). The fact that some of the variational states used in the 1000s of works in the field require MCMC, is a detail. There have been also several other works using states that do not require MCMC, but that still use VMC or tVMC. Please, do not confuse the numerical framework with the variational form used in that framework.
> >
> > My main criticism here is that you are doing an application of machine learning to quantum physics, but your analysis is essentially limited to the description of FermiNet (and in some cases PauliNet), a very specific ansatz that is used to describe ground state properties of fermions with VMC. You do not discuss, or attempt to compare against, instead, the state of the art methods in performing dynamics with neural-network wave functions. These, as I mentioned, can treat systems significantly larger than what you have done here. For this reason, while I am "sold" on the theoretical framework you propose (which is significant, at least for bosons, since fermions are a no go), I am not at all convinced that your scheme is practical beyond 5/10 particles. Of course, I would be more than happy to be proven wrong.

---

> ### Author Response · Authors · 2023-11-20
>
> We are grateful for your thoughtful remarks,
>
> > as it is, wouldn't work for fermions, simply because the phase is discontinuous and its gradient is intrinsically not well defined.
> > at least for bosons, since fermions are a no go
>
> In fact, this is not correct, we refer to Dankel, T. G. Jr., "Mechanics on manifolds and the incorporation of spin into Nelson's stochastic mechanics", this work covers both half and integer spins.
>
> >You cannot equate VMC with "physically inspired" functions. VMC (and t-VMC/t-DVP) are just frameworks to optimize a given variational state (that could even be exactly the same functions you are taking in your work) to either approximate the ground state (VMC) or the dynamics (t-VMC/t-DVP). The fact that some of the variational states used in the 1000s of works in the field require MCMC, **is a detail**. There have been also several other works using states that do not require MCMC, but that still use VMC or tVMC. Please, do not confuse the numerical framework with the variational form used in that framework.
>
> This is not just "a detail", this is crucial and why we insist on the claim that it is not correct to compare our work with VMC in the context that we present in our work. This is the central piece of our work.
>
> As our approach proposes a theoretical mechanism, that we justified in our theory, which you agree with, is of theoretical importance, that treats the entire problem as a black-box and produces exact sampling. The only "black box" t-VMC implementation we are aware of is based on Metropolis-Hastings algorithm that is not nearly as efficient, as the samplers used in those methods you mention.
>
> In our work, we heavily emphasize on doing such "black box sampling under no assumptions". Thus, we compare with approaches that do comparatively the same. Based on a per-ansatz basis, we can select a more appropriate approximation class for the networks $u_{\theta}, v_{\theta}$, more appropriate integrator (which is analogue of "effective MCMC sampling"; yet we take the most basic integrator as proof of a concept). After doing so, we can compare with these approaches you mention that are able to sample for many particles. Though, this is entirely out of the scope of the current work.
>
> We also refer to t-VMC related work https://arxiv.org/pdf/2305.14294.pdf and cite the authors of this work: "First, we formalize the origin of the numerical challenges affecting tVMC by proving that they arise from the **Monte Carlo sampling**, which may **hide a bias** or an **exponential cost** when the wave function contains zeros, as is the case for many physically-relevant problems." and, for example, to highlight typical assumptions placed in VMC works and, we also cite authors of that work: "The inset highlights a power-law relation between the $N_s$ necessary to reconstruct the dynamics accurately and $\varepsilon$, proving that $N_s ∼ 2^N$" and, finally, "As normalisation of the state imposes $\varepsilon \propto 2^{−N}$ , the number of samples necessary to correctly compute the quantum geometric tensor and the variational forces will diverge as $N_s \propto 2^N$ , eliminating the advantage of stochastic sampling and rendering tVMC computationally ineffective.".
>
> That is all to highlight that very assumption "we have effective MCMC" already assumes "assume we can solve VMC efficiently". Thus, from a perspective of our work, this is the same as assuming "assume one can do sampling better than black box sampling, let's compare it with a black box sampling now".
>
> We are grateful for your reasonable and thoughtful remarks, and we do agree that there are many limitations of such of such kind as you mention, but Rome was not built in one day, like VMC became efficient for sub-problems, this was a collective scientific effort of dozens of research groups, and we do not expect to outperform all those research groups efforts in one work, while they had an opportunity to develop VMC for their problems and obtain efficient samplings under assumptions that we do not assume.
>
> We are hopeful to attract an interest of research groups that might attempt to adopt it to their problems and use more assumptions, available in their field, to significantly improve the work and make it scalable as much as they wish.
>
> Our work mainly of theoretical with the new theoretical insights on possibility of "blackbox" exact sampling
>
> We surely will include such discussion in the Relevant work and Limitation sections, which will improve of work as what you say is fair, and will strengthen our central message about our work as being fundamental for a new approaches that it might pre-seed which would become based on a mixture of our ideas and variety of tVMC ones, though, we cannot foresee it now, but I am sure that there are many research groups, for whom VMC is not efficient, and, thus, they might potentially instead extend our work and make it more effective with more efficient integrators than those we use for their case.

---

> ### Author Response · Authors · 2023-11-23
>
> Dear Reviewer APZm,
>
> We will be grateful if you can elaborate on why you lowered your score, given that we addressed your concerns, in particular:
> 1. We clarified the applicability of our approach to fermionic systems, which you claimed was a no-go for our approach.
> 2. We provided more experiments for the interacting systems in higher dimensions, but you doubted it could run in more than two dimensions.
> 3. We explained the differences in setups we consider in our work and the approaches you mentioned.
> 4. You agreed that our approach is novel and that our approach does exact sampling.
> 5. We agreed that t-VMC needs to be discussed in the Relevant Work and Limitation section.
>
> We are grateful for the fruitful discussion that we had. However, we are very confused with your decision to lower the score and not allow us to address your concerns. We would be very grateful if you could elaborate and explain your decision, as we value your feedback.

---

> > ### Comment · Reviewer_APZm · 2023-11-23
> >
> > My current assessment/score is based on the revision I have access to. This does not contain any discussion on the related methods and does not address my concerns on the comparison to the state of the art. Am I missing something crucial/ a more recent revision of the paper here? (For example you mentioned you improved Table 1 but I don't see how)
> >
> > > We clarified the applicability of our approach to fermionic systems, which you claimed was a no-go for our approach.
> >
> > I am not convinced about this point. The paper you linked shows the case of a single fermion, which does not have issues with the gradients of the wave function's phase being singular. For many interacting fermions the wave function changes sign when you exchange two coordinates, so the gradient of the phase is intrinsically ill defined, for this reason I am still not convinced that the scheme (at least in same form you proposed here, where gradients of the phase are needed) is just not applicable.
> >
> > > We provided more experiments for the interacting systems in higher dimensions, but you doubted that it could run in more than two dimensions.
> >
> > You ran experiments on dimensions and systems that are very far from the state of the art, especially for the very simple systems you are considering here. You mention that the RAM consumption explodes for more than a few particles, this is quite concerning for scalability, in my opinion.
> >
> > > We explained differences in setups that we consider in our work, as approaches you mentioned.
> >
> > I am sorry but I cannot see these changes in the available pdf. If I am missing a revision please let me know. Indeed in the latest reply you mentioned "We are also planning on adding more discussion of related works in the main part of our paper." so I had the impression you were not to implement these changes at this stage yet. I would be happy to revise the paper when these changes are made available.
> >
> > > You agreed that our approach is novel and that our approach does exact sampling.
> >
> > Indeed, I do agree about this. I have deep concerns about scalability and generalizability instead, as mentioned.
> >
> > Notice also that here we are not touching on another important issue, namely the integrator you use, which is only first order. Using higher-order integrators for a stochastic ODE is a notoriously painful task and I am not sure this is well described in the limitations of the method.
> >
> > In any case, I would be happy to raise my score if at least the presentation of the related methods and a bit of discussion on the limitations is improved. I just cannot see this in the current version of the manuscript.

---

> ### Author Response · Authors · 2023-11-23
>
> >I am not convinced about this point.
>
> There is no problem of considering the same on the manifold formed from the product of spin manifolds. We also refer to Section 21 "Spin" in Nelson E.,  "Quantum Fluctuations" where the author describes it in a general case.
>
> >My current assessment/score is based on the revision I have access to.
>
> We kindly ask you to look at the latest revision. And especially in Supplementary materials in the experimentation Section and in Table 1, as it is updated. We added dimension five interacting particles in the Supplementary materials.
>
> We have not added Related Work and Limitation changes yet, as we have limitation on the number of pages, but we will add them in the final revision. We will fully faithfully describe about limitations that you pointed out as it is reasonable.
>
> > You mention that the RAM consumption explodes for more than a few particles, this is quite concerning for scalability, in my opinion.
>
> We showed **the oppositive**, we showed that memory consumption does not explode and growth linearly, it exploded for C-N method, but not for ours. We refer you to the experimental secion in the revision or look at the reply to the Reviewer JVMX
>
> >I am sorry but I cannot see these changes in the available pdf. If I am missing a revision please let me know.
>
> Can you see the changes in "supplemenraty materials" file? They are not in main pdf, as main pdf has hard limit on the number of pages.
>
> >Notice also that here we are not touching on another important issue, namely the integrator you use, which is only first order. Using higher-order integrators for a stochastic ODE is a notoriously painful task and I am not sure this is well described in the limitations of the method.
>
> We discussed in the main text that we use the simplest, integrator choice is out of the scope, and we do not touch this topic, you are right.
>
> >In any case, I would be happy to raise my score if at least the presentation of the related methods and a bit of discussion on the limitations is improved.
>
> Given this limited time left before rubuttle end, we kindly ask you to raise this score, as in such limited time we won't be possibly able to fit into hard pages limit of the main pdf.
>
> We can however, say that we will do the following:
> 1. We will mention VMC in the Table 2 and describe that it is general framework that exists.
> 2. We will mention all works you mention and mention their impressive capabilities to run for 100 bosonic particles for their respective problem formulations.
> 3. We will mention limitation of our work like reliance on SDE integrators.
> 4. We will mention that our approach, while has favourable assymptotics, in current implementation we were able to show only five interacting particles.
>
> We hope that you can understand our current time limitation & pages constraint that exists before we can do camera-ready version. We hope that you can trust our word that all those remarks will be added before publishing.
>
> In the current revision, we have added in the supplementary experiments with more particles & showed that the memory does not explode.
>
> We hope for your understanding and trust and kindly ask to raise the score if this will be satisfactory for you.

---

> > ### Comment · Reviewer_APZm · 2023-11-23
> > **Revised Score**
> >
> > I have revised my score as a gesture of good will, despite the fact that the changes I asked for have not been implemented in the current revision. I trust these will be implemented in the camera ready version.
> >
> > > We will mention VMC in the Table 2 and describe that it is general framework that exists.
> >
> > OK, though the main point is mentioning t-VMC/ t-DVP approaches, since these are the main relevant technique to compare your work against.
> >
> > > We hope that you can understand our current time limitation & pages constraint that exists before we can do camera-ready version.
> >
> > I would suggest to find some space by removing, for example, the highly speculative discussion on BQP and similar. It is clear your approach (like anything else involving optimizing neural networks and non trivial losses) is theoretically at least in NP.
> > Including discussion of relevant and related techniques instead is much less speculative and factual, so it should find space in the main text.
> >
> >
> > Considering Fermions:
> >
> > > There is no problem of considering the same on the manifold formed from the product of spin manifolds. We also refer to Section 21 "Spin" in Nelson E., "Quantum Fluctuations" where the author describes it in a general case.
> >
> > No, I am not referring to spin in the sense of Section 21. I am referring to statistics, namely section 20 of the book you mention. In there, Nelson himself clearly writes: "Probabilistic techniques have not been applied in a natural way to the study of Fermi fields, either in imaginary or real time. My hunch is that no departure from the framework of stochastic mechanics is needed... "
> > So it's a "hunch" rather than a proof. As I was mentioning already, if you have a Fermionic state the issue you have to deal with is the antisymmetry, and it is not at all obvious that the diffusion process you are considering will live to explore more than the nodal surface imposed by the initial state, as it must do in the exact solution. In fact, I believe it is possible to show that it will not go beyond the initial nodal surface.

---

> ### Author Response · Authors · 2023-11-23
>
> Dear Reviewer,
>
> Given that there is one minute left, and we believe that there was some honest confusion about our method with C-N method, i.e., around the memory scaling of our method, which does not explode as you mentioned, and given our promise that we will include mentioned by your works in the reference and will discuss limitations of our work, which should address your concerns, we hope for your reconsideration of the score, given the novelty of our approach. It's potential importance for the field.
>
> We are happy to address any other concerns and would've changed the revision if we had enough time to reply.

---

### Official Review · Reviewer_oJtq · 2023-10-28

**Soundness:** 3 good
**Presentation:** 3 good
**Contribution:** 3 good
**Rating:** 8
**Confidence:** 3

**Summary:**

In this paper, the authors propose a stochastic formulation of the Schrödinger equation. The proposed formulation can be used to simulate quantum mechanics with high efficiency.
A key difference between this new formulation and the one proposed by Nelson is that the new method employs the gradient of the divergence operator to facilitate the neural computations.
They also prove theoretically that the loss function used to train neural networks is upper bounded by the L2 distance between the true process that samples from the quantum density and an approximate process which the neural network tries to predict.
Experimental results show that the proposed method is superior to the baseline method PINNs.

**Strengths:**

1. The new stochastic formulation of the Schrödinger equation provides an efficient way for quantum mechanics simulation by utilizing the power of neural computation.
2. Training loss of the neural networks for learning stochastic process is bounded with theoretical guarantees.
3. The O(Nd) computational complexity looks very promissing and opens the door for large-scale quantum simulation.

**Weaknesses:**

1. Seems the neural network employed in this study is a single layer neural network. I am wondering how a single layer neural network could learn dynamics of a complicated wave function. Also the illustrations in the experiment look pretty simple. The authors are encouraged to tackle more complicated cases using the proposed method.
2. The O(Nd) computational complexity need more elaboration. Is it the computational complexity of training the neural networks on a single trajectory? For learning a stochastic process, we may need to sample many trajectories  in order to learn hidden low dimensional representations of process. I am not sure whether it is fair comparisons with other methods as listed in the table 1.

**Questions:**

1. For training losses defined in eq. 11 to eq. 15, because they need integration operation, I am wondering in practice how the integration is done and what is the window length of the integration during the training process at each iteration/epoch.
2. In page 5, the authors mentioned for each iteration, they will sample new trajectories using eq. 7. How do we handle the cold start problem at the very beginning of the training process. I mean at the beginning of the training process, neural networks have not learned dynamics of the stochastic process. So the trajectories we sampled may be invalid or they will mis-guide the learning of the target stochastic process.

---

> ### Author Response · Authors · 2023-11-18
> **Authors reply 1/2**
>
> Thank you for your thoughtful and positive feedback! Let us address your questions:
> > **W1** *Seems the neural network employed in this study is a single layer neural network. I am wondering how a single layer neural network could learn dynamics of a complicated wave function. Also the illustrations in the experiment look pretty simple. The authors are encouraged to tackle more complicated cases using the proposed method.*
>
> Even in the “simple” settings considered in our paper, existing algorithms (e.g., finite-difference based or PINNs) encounter significant limitations and scaling issues both in terms of computation time and memory requirements. The core contribution of our submission is demonstrating an approach that sidesteps these challenges. The most interesting system to study, of course, is many-particle systems with interactions. We show that our method works for a simple interacting system (Sec. 5.4), and we use a more sophisticated network (for example, we add skip-connections) in this case. In Appendix D5, we provide more implementation details for this setting. Also, in our reply (2/2) to the reviewer JVMX, we show how our DSM approach scales with the problem dimension for the interacting system, given that potential can be computed linearly.
>
> Regarding the use of a single-layer neural network, we intentionally choose a simplified architecture to emphasize the foundational aspects of our framework. We aim to prove that our framework works, not due to some factors like a smart choice of architecture. As our research progresses, we are open to exploring more sophisticated task-specialized neural architectures and algorithmic enhancements to tackle more complex simulations. Finally, works like [1] show that even single-layered neural networks with non-linear activations, when used in Neural SDEs, are powerful universal approximators.
>
> [1] Veeravalli T. et al.,  A Constructive Approach to Function Realization by Neural Stochastic Differential Equations
>
> > **W2** *The O(Nd) computational complexity need more elaboration. Is it the computational complexity of training the neural networks on a single trajectory? For learning a stochastic process, we may need to sample many trajectories in order to learn hidden low dimensional representations of process. I am not sure whether it is fair comparisons with other methods as listed in the table 1.*
>
> Thank you for noticing that. This comparison is unfair for a numerical solver indeed, as it does not have a well-defined notion of one iteration. By an iteration, we mean one training epoch, so we report computational complexity per epoch. We are sorry about this confusion. We will clarify this by modifying the table as detailed below (in fact, it’s modified in the revised version). That said, our reported complexities are important for understanding scaling properties of different methods in higher-dimensional settings. Note that all complexities are given under the assumption that the computational complexity of the potential is $O(d)$. If it’s more than that, then complexities of all methods needs to be corrected by this higher complexity additively.
>
> To modify the table, we will add a column, “Overall complexity”, and we will place “N/A” in the “Iteration complexity” column for a numerical solver. In the “Overall complexity” of the numerical solver, assuming that we aim for a precision $\varepsilon = \mathcal{O}(\frac{1}{\sqrt{N}})$, we will place $\mathcal{O}(\frac{1}{\epsilon^{d+2}})$. In the “Iteration complexity” column, we will keep other methods as they are. Finally, in “Overall complexity”, we will specify that for all other methods existing theoretical results allow only to place lower bounds on complexity. In particular, we will multiply iterations complexity by a factor $\mathcal{O}(\mathrm{poly}(\frac{1}{\epsilon}))$, coming from a general non-convex stochastic gradient descent bound. Such scaling $\mathrm{poly}(\cdot)$ (but not exponential) is broadly generic for non-convex stochastic gradient convergence to the minima; and it is true for a broad class of non-convex stochastic gradient methods [1].

---

> > ### Author Response · Authors · 2023-11-18
> > **Authors reply 2/2**
> >
> > One of the constant multiplies to the asymptotic for our method is covered in Theorem 4.2 in Section 4. There are other factors that need to be accounted to estimate the overall convergence, e.g., the lowest eigenvalue of the neural tangent kernel that is architecture-dependent and might depend on the dimension exponentially in the worst case. For approaches like FermiNet convergence of MCMC sampler is important. It also might exhibit exponential convergence for some problems, but for some not. For PINN and FermiNet, an inverse of an eigenvalue of a neural tangent kernel of $\Delta \psi_{\theta}$ should appear in the constants. Since for all methods, including ours, explicit w.r.t. dimension bounds on the convergence are not derived yet, we will specify it in such a broadly generic form. This highlights the fact that, in some cases, deep learning achieves more favorable scaling. Establishing sub-classes of quantum systems with a non-exponential constant $C$ is out of the scope of the current work. We note that while our Theorem 4.2 establishes one of the factors of this constant to not scale exponentially for some sub-class of quantum mechanical problems, for the full picture, we need to account for a universal approximation error, stochastic gradient dynamics, and discretization errors. It requires new theoretical techniques to be developed.
> >
> > [1] Fehrman B. et al., “Convergence rates for the stochastic gradient descent method for non-convex objective functions”
> >
> > > **Q1** *For training losses defined in eq. 11 to eq. 15, because they need integration operation, I am wondering in practice how the integration is done and what is the window length of the integration during the training process at each iteration/epoch.*
> >
> > We consider the simplest integrator of SDE, the Euler-Maruyama scheme. We split time $(0, 1]$ uniformly in $N+1$ parts, where $N=1000$. So, $dt = 0.001$. (We mention these details in Appendix C). A higher-order integrator or an adaptive integrator could achieve the desired accuracy with a much lower $N$. We leave it for future work.
> >
> > > **Q2** *In page 5, the authors mentioned for each iteration, they will sample new trajectories using eq. 7. How do we handle the cold start problem at the very beginning of the training process. I mean at the beginning of the training process, neural networks have not learned dynamics of the stochastic process. So the trajectories we sampled may be invalid or they will mis-guide the learning of the target stochastic process.*
> >
> > We note that despite the sampling of imperfect neural SDE, our samples for X(0) are perfect. This means that the training dynamic is not misguided at initial timestamps. This fact is used to sharpen our bounds in Theorem 4.2. From the theorem, it can be seen that improvements even in ‘misguided’ regions actually improve the bound on the strong convergence, which is an interesting observation by itself. It indicates that learning is robust to misguiding. The theoretical mechanism of incremental accumulation of the errors, instead of blowing up, is explained in Lemma F.5. It is worth additional research on whether using a mechanism of adaptive/reweighting might improve the bounds and further accelerate the convergence, which is out of the score of the current work.
> >
> > We are grateful for your valuable feedback and are committed to addressing any additional concerns or suggestions you may bring to our attention.

---

### Official Review · Reviewer_iLX6 · 2023-10-30

**Soundness:** 4 excellent
**Presentation:** 4 excellent
**Contribution:** 4 excellent
**Rating:** 8
**Confidence:** 3

**Summary:**

The paper presents a neural-network-based solver for simulating the time evolution of quantum systems, called Deep Stochastic Mechanics (DSM). DSM is based on Nelson’s formulation of stochastic quantum mechanics where the quantum system evolution is characterized by a stochastic process. The authors captured the correspondence between Nelson’s formulation and a diffusion process, where a partial differentiation equation lies at the heart. Then they propose to approximate the solution to the PDE with neural networks. A training process employs errors of the PDE in the time-evolution and initial points as the loss functions to approximate the solution. Empirical studies of this method are also conducted on several typical quantum systems.

**Strengths:**

The paper’s approach to simulating quantum systems is very intriguing. It provides insights into the connection between quantum mechanics and diffusion processes and exploits it with neural networks which recently have shown advantages in dealing with diffusion processes. It ends in a stochastic NN-based simulator with low training cost and with theoretical guarantees on the quality of the solution. Since the training process is based on trajectories of the stochastic process, it is naturally adaptive to the structure of the solution and boosts the precision because of the focus on the solution space.

The experiments demonstrated the performance of DSM to surpass prior methods, tackling the most challenging problem in simulating quantum systems: the curse of dimensionality. It is suggested that NNs have the capability of exploiting the low latent dimension of the simulated quantum system, making use of their advantages in extracting low-dimensional features in high-dimensional data. This observation seems the most interesting part of this approach, which, to the best of my knowledge, is not exploited in other approaches. It is quite possible that a large number of quantum systems effectively have low latent dimensions, and DSM may be well-suited to provide sampling from these systems.

Although the materials may not be easily accessible to the general ML audience, the well-organized presentation is likely to deliver the core ideas of this paper to a broader community. The inspiring innovations of this paper make it worthy to be published in ICLR.

**Weaknesses:**

Although the experiments cover several interesting cases where DSM works well, the limitation of DSM is not fully discussed in the paper. I would like to understand what cases will fail DSM. For example, does it perform worse on systems with complicated interactions or large latent space? I believe such examples may better illustrate the limitations and the suitable scenarios of DSM.

**Questions:**

Additionally, I wonder how the performance of DSM through stochastic quantum mechanics compares to other (NN-based) PDE solvers via Shrodinger’s formulation. Is it possible to exploit the latent space within Shrodinger’s picture, either with NNs or not?

A minor point: in Section 5.1, it is claimed that "Table 2 shows ... and the training time for ...", while the training time is not found in Table 2. Where is it?

---

> ### Author Response · Authors · 2023-11-18
>
> Thank you for the time and effort devoted to our paper. Let us address the mentioned weaknesses and questions:
> > **W1** *Although the experiments cover several interesting cases where DSM works well, the limitation of DSM is not fully discussed in the paper. I would like to understand what cases will fail DSM. For example, does it perform worse on systems with complicated interactions or large latent space? I believe such examples may better illustrate the limitations and the suitable scenarios of DSM.*
>
> We introduce a novel methodology and explore its capabilities in various settings, including non-interacting and interacting systems. In general, we might expect DSM to encounter challenges in high-energy interactions and dynamics, as shown in constants in Theorem F.13. Additionally, issues may arise with singular potentials (for example, an atom model) due to numerical instabilities. However, similar challenges are faced by traditional methods, PINNs, and other approaches. It’s possible to use some tricks to sidestep it, for example, clipping gradients and approximating potential with globally smooth function, as explained in Appendix G.3. We also mention potential limitations in Section 6 and future work in Appendix I. For example, our method requires modification to take a spin into account. Our experiments consider bosons only while exploring fermionic systems, and addressing antisymmetry properties is important for future research.
>
> > **Q1** *Additionally, I wonder how the performance of DSM through stochastic quantum mechanics compares to other (NN-based) PDE solvers via Shrodinger’s formulation. Is it possible to exploit the latent space within Shrodinger’s picture, either with NNs or not?*
>
> Our approach is designed for the time-dependent Schrödinger equation, while many NN-based and not NN-based methods exist for the time-independent (stationary) equation. These methods usually look for the ground state solution, and some methods work reasonably well in high dimensions (for example, FermiNet). The most “apples-to-apples” comparison with our approach is PINNs. Its loss function uses the Schrödinger equation, and traditional numerical solvers also use the Schrödinger equation explicitly. We compare our method with these methods in our manuscript, and our method usually outperforms them in terms of accuracy and computational complexity.
>
> The problem of the direct solvers of the Shrödinger equation is that they might not provide robust convergence to the score function necessary for using ODE/SDE samplers from a density. However, there might be a way: firstly, apply DSM to obtain an approximate sampler from the wave function, and then apply a method like PINN where trajectories from DSM are used as a grid. An alternative is using Langevin-like ODE/SDE with a drift $(\mathcal{Re}+\mathcal{Im})\nabla log \psi_{\theta}(x, t)$ with fixed t where the sampler is converged well enough to the invariant measure, which in this case is $|\psi_{\theta}(x, t)|^2$. However, due to the non-log-concavity (multi-modality) of the wave function, this adds a computational overhead of ensuring that the sampler is converged to the invariant measure, which in that case will be $|\psi_{\theta}(\cdot, t)|$ (not even to the true $|\psi(x, t)|^2$). Such an approach would be dimensionally cursed by design, as even for a class of dissipative densities (strongly convex outside a ball), the bound of the sampler to the invariant measure is exponential in dimension [1, 2]. On the contrary, our approach relying on Nelsonian PDEs minimizes the path-wise sampling error $X(t)$ to $Y(t)$ directly, as shown in Theorem F.13 in the Appendix. This is a fundamental and unique property of the Nelsonian formulation, which we established in Section F. While it might not be applicable to a general problem of sampling from arbitrary density, it allows us to sidestep such issues.
>
> [1] Bardet J., et al., Functional inequalities for Gaussian convolutions of compactly supported measures: explicit bounds and dimension dependence
>
> [2] Raginsky M., Non-Convex Learning via Stochastic Gradient Langevin Dynamics: A Nonasymptotic Analysis https://arxiv.org/pdf/1702.03849.pdf
>
> >**Q2** *A minor point: in Section 5.1, it is claimed that "Table 2 shows ... and the training time for ...", while the training time is not found in Table 2. Where is it?*
>
> Thank you for pointing this out. We do not report training time here, it’s s typo. We added scaling experiments where we report training time in Appendix D.5.1 in our supplementary material.
>
> We appreciate your positive feedback, and we are happy to address any further concerns or suggestions you may have.

---

> ### Comment · Reviewer_iLX6 · 2023-11-21
>
> Thanks for the author response. I do not have further comments.

---

### Official Review · Reviewer_JVMX · 2023-10-30

**Soundness:** 3 good
**Presentation:** 3 good
**Contribution:** 3 good
**Rating:** 6
**Confidence:** 4

**Summary:**

The paper presents a new approach to simulating time evolution in quantum systems by learning the gradient of the modulus and phase of the wavefunction over time. This is done by using a loss function that ensures the networks parametrizing these gradients satisfy the Schroedinger equation. The main novelty is to evaluate this loss on a batch coming from the samples of a stochastic process that describes the evolution of the modulus squared of the wavefunction. Numerical experiments for low dimensions show a better behavior than PINN.
My score is between 6 and 8, leaning towards 6. If more evidence for favourable scaling wrt traditional methods is provided, I can increase my score.

**Strengths:**

- Novel and interesting approach to simulating time evolution in quantum mechanics using deep learning
- Favourable O(d) scaling compared to PINNs which requires O(d^2) to compute the Jacobian
- Evaluation of the loss on trajectories coming from the stochastic process governing the modulus squared of the wavefunction, compared to PINNs which require to specify collocation points.
- Experiments show better performance than PINNs

**Weaknesses:**

- The framework does not seem to allow estimation of non-diagonal observables, eg momentum
- While I understand the complexity argument for evaluating the loss in O(d), it is unclear to me whether this method can scale better than traditional approaches. In particular, experiment of figure 4 studies the harmonic oscillator for d=1..9. However, if I understand the setting, the Hamiltonian is separable in the dimensions and so the problem has a natural scaling linear in d. I am therefore unsure about the promise for a more favourable scaling than traditional methods.
- The use of stochastic formulation of quantum mechanics for quantum mechanics simulation is not necessarily novel, however I have not seen references to it. I am not an expert in this field, but one example I found is [Quantum Dynamics with Trajectories, Robert E. Wyatt].

Minor:
- below eq 5, one of the two u's should be v.

**Questions:**

- Can you benchmark the method against traditional solvers for an interacting problem in higher dimension and show a more favourable scaling?
- How would one estimate the expectation of a non-diagonal operator?

---

> ### Author Response · Authors · 2023-11-18
> **Authors reply 1/2**
>
> Thank you for your thoughtful feedback! Let us address your concerns:
> > **W1** *The framework does not seem to allow estimation of non-diagonal observables, eg momentum*
>
> Our framework allows estimation of the momentum in the following ways:
>
> 1. The first way is estimating $\mathbb{E} \mathbf{P}(t)$ as $ \mathbb{E} v(X(t), t) = \mathbb{E}(v(X(t), t) \pm u(X(t), t)$. “Quantum fluctuations” by Nelson proves that this expectation of the momentum is the expectation of the osmotic velocity. However, this approach works only for the first moment of the momentum; higher order and arbitrary expectations (i.e.,  $\mathbb{E} g(P(t))$ for arbitrary nonlinear g) can’t be obtained that way.
>
> 2. Another way works if a potential is quadratic (e.g., a harmonic oscillator). The Fourier transform allows re-writing the Shrödinger equation in momentum coordinates, which also has a quadratic potential and allows diffusion interpretation. Therefore, a diffusion can be obtained as $\mathrm{d}P(t) = (\nu u(P(t), t) + v(P(t), t))\mathrm{d}t + \sqrt{\frac{\nu \hbar}{2}}\mathrm{d}W^{P}(t)$. This allows us to estimate all momentum $f(P(t))$ for any non-linear $f$. This, however, implies that we can’t obtain an arbitrary nonlinear pointwise observable $g(X(t))$ at the same time.
>
> 3. Moreover, an arbitrary non-linear observable can be obtained by recovering $p(x, t)$, $S(x, t)$ from samples using learned $u(x, t), v(x, t)$. This procedure is described in Appendix G.6. However, it is as computationally expensive as classical methods.
>
> 4. Finally, in Appendix G.4, we describe how to possibly measure observables of a form $g(X(t’), X(t))$ for arbitrary nonlinear g and  $t’ < t$ by incorporating a measurement process. It requires applying our method at least twice (on an interval [0, t’], and then on [t’, t]). This requires further research on whether deep learning can effectively capture initial conditions at the moment of the measurement $t’$ as an input to neural networks to avoid applying our approach $B+1$ (see definition in Appendix G.4) times but only twice. We leave this direction for future work.
>
> We are ready to add more details regarding momentum estimation to our paper.
>
> > **W2** *While I understand the complexity argument for evaluating the loss in O(d), it is unclear to me whether this method can scale better than traditional approaches… experiment of figure 4 studies the harmonic oscillator for d=1..9. However, if I understand the setting, the Hamiltonian is separable in the dimensions and so the problem has a natural scaling linear in d. I am therefore unsure about the promise for a more favourable scaling than traditional methods.*
>
> Traditional methods would demand $O(N^d)$ computations/memory, while a PINN would require at least $O(d^2)$ computations/memory. It holds regardless of a problem they are applied to, given no external knowledge of the potential function properties. At the very least, our method does not introduce a computational overhead beyond what is needed to compute the harmonic potential (which is $O(d)$ computations in this non-interacting particles case). This advantageous behavior is observed without any reliance on prior knowledge about the separability of the Hamiltonian. Our method does not rely on a grid in X, offering a significant advantage in terms of computational efficiency. Additionally, compared to PINN, our DSM approach avoids computing the Laplacian, which ensures that no overhead is added in the asymptotics beyond what is needed to compute the potential itself. That establishes lower bounds on the computational complexity of our method; and this bound is sharp for this problem.
>
> To explore the dimension dependency fully, we need to account for factors that arise from the error of discretizing SDE, a universal approximation bound, and stochastic gradient descent dynamics that might also give additional dependencies on the dimension. Addressing these aspects requires further research beyond the scope of the current work. However, recent works on diffusion models established a few interesting results on the dimension-free universal approximation of particular classes of score functions with deep learning, which relates to our work since $u(x, t)$ is a score function [1].
>
> [1] Lee H. et al., Convergence of score-based generative modeling for general data distributions

---

> ### Author Response · Authors · 2023-11-18
> **Authors reply 2/2**
>
> > **W3** *The use of stochastic formulation of quantum mechanics for quantum mechanics simulation is not necessarily novel, however I have not seen references to it. I am not an expert in this field, but one example I found is [Quantum Dynamics with Trajectories, Robert E. Wyatt].*
>
> We thank you for pointing out this reference! In the revised version of our paper, we will incorporate a citation to Wyatt's paper to ensure proper acknowledgment of the related work. In fact, our work interpolates between the Bohmian picture, explored in Wyatt’s paper, and the Nelsonian interpretation through parameter $\nu$. Many results from our work can be seen as generalizations and further development of these ideas with deep learning. For example, the equidistribution principle (EP) from Bohmian interpretation is generalized to the Nelsonian picture in our Lemma F.6. Moreover, Wyatt’s work primarily focuses on reconstructing a density from sampled trajectories, but not on efficiently solving the equations guiding those trajectories. From that point of view, Wyatt’s work is complementary to ours, as it proposes how one can reconstruct the density once you have a sampler of trajectories. This extension might open a way for solving a problem of estimating all possible non-diagonal observables if the convergence of the reconstruction can be ensured up to high order derivatives in a suitable sense (e.g., in Sobolev space $W^{p}_{2}$).
>
> > minor typo below eq. 5
>
> It’s fixed; thank you!
>
> > **Q1** *Can you benchmark the method against traditional solvers for an interacting problem in higher dimension and show a more favourable scaling?*
>
> There are experiments in the original submission that do correspond to interacting particles. **For this rebuttal, we ran new experiments for larger d to show scaling properties more clearly.** The following Tables illustrate time and memory usage for the Crank–Nicolson (C-N) solver for d=2, 3, 4 and training time/time per epoch and memory usage of our DSM method for d=2, 3, 4, and 5. Also, Figure 11 in **the revised supplementary material D.5.1** shows predicted densities for different dimensions. Note that while we don’t have the baseline for d=5, we believe DSM predictions are still reasonable.
>
>
> Table 1. Time to get a solution and memory usage of the Crank-Nicolson method for different problem dimensions.
> |      |       d=2      |       d=3      |       d=4      |
> | ----------- | ----------- |----------- | ----------- |
> | Time (s)      | 0.75       |    35.61   |  2363    |
> | Memory (Gb)   | 7.4        |10.6        | 214       |
>
> Table 2. Training time, time per epoch, and memory usage of our DSM method for different problem dimensions.
> |      |       d=2      |       d=3      |       d=4      |       d=5   |
> | ----------- | ----------- |----------- | ----------- |  ----------- |
> | Training time (min)      | 29.5       |    60.3   |  97.5    |  154.0 |
> | Time per epoch (s/ep)      | 0.52       |    1.09   |  1.16    |  1.24 |
> | Memory (Gb)   | 17.0      | 22.5      |  28.0    | 33.5 |
>
> Memory: In particular, DSM memory usage and time per epoch grow linearly in d (according to our theory and evident in our numerical results) in contrast to the C-N solver, whose memory usage grows exponentially since discretization matrices are of size  $N^d \times N^d$.
> We cannot run the C-N method for d > 4 with the amount of memory available in our computer system. The results show that our method is far more memory efficient for larger d.
>
> Compute time: While our DSM total compute times (including training time) are longer than C-N compute times for small d, the trend as d increases suggests the computational efficiency of our DSM method scales much better with d than the C-N method. In this example, the potential function has computational complexity $O(d^2)$ as it has all pairwise interactions. It implies that the training time complexity is bounded to be quadratic in the best case for all algorithms. Computation complexity of other DSM loss terms that involve computationally heavy differential operators remains to be linear. We assume domination of the differential operators computations explains why time per epoch does not seem to grow quadratically. These findings support the claim that our DSM method training time scaling is more favorable than scaling of the classical methods.
>
> > **Q2** *How would one estimate the expectation of a non-diagonal operator?*
>
> We replied to this in **W1** and partially in **W3**.
>
> Thank you for your insightful feedback, and we look forward to addressing any further concerns or suggestions you may have.

---

> > ### Comment · Reviewer_JVMX · 2023-11-20
> >
> > Thank you for your response. I do not have further comments.

---

### Author Response · Authors · 2023-11-22
**General response**

Dear Reviewers,

We sincerely appreciate your helpful comments and suggestions. We also would like to thank you for noting the strengths of our paper, namely:

- **Novelty** of our framework to simulating time evolution in quantum mechanics using deep learning with exact sampling.
- **Theoretical justification** of the framework with explicit bounds on the sampling error with novel insights.
- **Favorable scaling** compared to classical methods and PINNs.
- **Empirical validation** of our DSM approach.

We had insightful discussions that helped us clarify the motivations and the advantages of our method. We have answered the individual questions in separate comments. In the revised version, we have incorporated the reviewers' suggestions by fixing typos, adding more clarifications of our method (updated Table 1), and more experimental results. In particular, we have added experimental results on studying scaling capabilities of our method for interacting particles in Appendix D.5.1. These results support our claim about our DSM method having favorable scaling in terms of training time and memory needs. We are also planning on adding more discussion of related works in the main part of our paper.

In closing, we thank the reviewers again for their time and attention. If you have further concerns, please let us know, and we will be happy to address them.

---

### Meta-Review · Area_Chair_GqvH · 2023-12-12

**Metareview:**

The authors present a method for numerical simulation of the time-dependent Schroedinger equation in (potentially) high dimensions using Nelsonian stochastic mechanics, a reformulation of quantum mechanics as a stochastic differential equation, and show how deep neural networks can be used to represent the solution of this dynamical system. The idea is quite original, and deserves to be developed further, but there are several issues with the paper as is.

The comparison against the prior literature is lacking. The authors almost exclusively focus on recent work using neural networks and ignore the vast swathes of the computational quantum mechanics literature that predates the recent interest in deep neural networks for computational quantum mechanics. This is not merely a matter of citing prior work (as the authors agreed to do during the discussion period) but of actually *comparing* against the prior work. I am glad to see that the authors extended their calculations of the interacting case from 2 particles up to 5 particles, but the only comparison against prior work is against the Crank-Nicholson method on a grid. A proper comparison against time-dependent VMC is necessary here, with or without a neural network ansatz.

I should note that, based on the related work the authors cite, it seems that they are arguing that this work should be judged only against other deep neural network based approaches like PINNs or the FermiNet. However, even within the domain of neural networks for VMC with electrons in real space, the FermiNet and PauliNet were not the first work in that domain - DeepWF was. The FermiNet and PauliNet were the first works *to show an improvement over non-neural approaches*. A similar demonstration would be needed here to merit publication (and I would have made a similar criticism of DeepWF had it been submitted to ICLR). A comparison of t-VMC and DSM on an interacting bosonic system seems like the minimum needed.

The authors also motivate the paper on the basis of its potential for use in many-electron calculations, but no consideration of fermions is given anywhere in the paper. I would either include a simple fermionic example like the electron gas or a quantum dot, or rewrite the introduction to focus more on general quantum simulation and less on electronic structure.

I am also not certain about the novelty of the main difference between their formulation and the "traditional" Nelsonian equations. The difference seems to come down entirely to a difference in how to express the Laplacian of the current velocity, and it seems like a trivial change to the operator based on the assumption that the current velocity is integrable, which the authors themselves state holds in general for Nelsonian time evolution but not the Schroedinger equation. So I am both confused as to how applicable it is, and skeptical about how big of an advance it is, although it does seem like a practical trick for speeding up calculations.

Overall, I will say that I liked the *idea* very much, and hope that the authors do develop it further. But it does take proper engagement with the prior work in the field, which goes far beyond recent work using neural networks, to show that this method is a practical advance in the state-of-the-art.

**Justification For Why Not Higher Score:**

I address this in the metareview.

**Justification For Why Not Lower Score:**

I address this in the metareview.

---

### Decision · Program_Chairs · 2024-01-16

Reject